# Dueling Bandits with Team Comparisons

**Lee Cohen**[*]
Tel Aviv University
leecohencs@gmail.com

**Ulrike Schmidt-Kraepelin**[*]
Technische Universität Berlin
u.schmidt-kraepelin
@tu-berlin.de

**Yishay Mansour**
Tel Aviv University and
Google Research
mansour.yishay@gmail.com

## Abstract

We introduce the *dueling teams problem*, a new online-learning setting in which the learner observes noisy comparisons of disjoint pairs of $k$-sized *teams* from a universe of $n$ players. The goal of the learner is to minimize the number of duels required to identify, with high probability, a *Condorcet winning team*, i.e., a team which wins against any other disjoint team (with probability at least $1/2$). Noisy comparisons are linked to a total order on the teams. We formalize our model by building upon the dueling bandits setting (Yue et al., 2012) and provide several algorithms, both for stochastic and deterministic settings. For the stochastic setting, we provide a reduction to the classical dueling bandits setting, yielding an algorithm that identifies a Condorcet winning team within $\mathcal{O}((n+k\log(k))\frac{\max(\log\log n,\log k)}{\Delta^2})$ duels, where $\Delta$ is a gap parameter. For deterministic feedback, we additionally present a gap-independent algorithm that identifies a Condorcet winning team within $\mathcal{O}(nk\log(k) + k^5)$ duels.

## 1 Introduction

Multi-arm bandits (MAB) is a classical model of decision making under uncertainty. In spite of the simplicity of the model, it already incorporates the essential tradeoff between exploration and exploitation. In MAB, the learner performs actions and can only observe rewards of the actions performed. One of the main tasks in MAB is *best arm identification*, where the goal is to identify a near-optimal action while minimizing the number of actions executed. The MAB model has numerous practical applications, including online advertising, recommendation systems, clinical trials, and more. (See Slivkins (2019); Lattimore & Szepesvári (2020) for more background).

One weakness of the MAB model is the assumption that real-valued rewards are always available. In many applications, it is more natural to compare two actions and observe which one of them is better rather than give every single action a numerical reward. For example, recommendation systems often suggest two items and obtain only their relative preference as feedback (e.g., by a click on one of them). This leads very naturally to the well-known model of dueling bandits (Yue et al., 2012), where the learner selects a pair of actions each time and observes the binary "winner" of a duel between the two. (See Busa-Fekete et al. (2018) for a survey on extensions of the dueling bandit model).

In this work we are interested in the case that the learner has to select two disjoint *teams* for a duel, which are $k$-sized subsets of the actions (which we call players). This appears naturally in sports or online games, where the goal is to pick one of the best teams from a set of players by observing the outcomes of matches (say, to be a school representative team, or to sponsor for tournaments). Examples include doubles tennis, basketball, and the online game League of Legends, where each match requires two disjoint teams of players to compete. Similar phenomena appear in working environments, where different R&D teams compete on implementing a project. Another example could be online advertisements where multiple products are bundled to a display ad and a customer

---

[*]These authors contributed equally to this work.

35th Conference on Neural Information Processing Systems (NeurIPS 2021).

can click on any of two presented bundles, e.g., some online games offer in-app bundle purchases, and the information regarding sales of different bundles can improve the bundles' composition.

Our basic model is the following. We have a universe of $n$ players, and at each iteration the learner selects two **disjoint** teams for a duel and observes the winner. For any two different teams, there exists an unknown stationary probability that determines the winner of a duel between them. The requirement that dueling teams need to be disjoint is in accordance with the situation in games, where a single person cannot play for both teams. The goal of the learner is to minimize the number of duels required to identify, with high probability, a *Condorcet winning team*, i.e., a team which wins against any other disjoint team (with a probability of at least $1/2$). We assume these probabilities are linked to a strict total order on all teams, which implies the existence of a Condorcet winning team, yet it is typically not unique. We make two minimal and natural assumptions on this total order on teams: that it is *consistent* to some total order among the players, and that team probabilistic comparisons hold *Strong Stochastic Transitivity*, a common assumption in dueling bandit settings.

Our *consistency* assumption implies that the best team is the team of top-$k$ players, and this also a Condorcet winning team. However, not all relations between players are deducible for the learner. In particular, even achieving accurate estimations of the latent winning probabilities between all disjoint teams might not suffice to separate the top-$k$ players from the rest. Consider for example an instance with four players $1 \succ 2 \succ 3 \succ 4$ where $k = 2$ and the total order among the teams is lexicographical, i.e., $12 \succ 13 \succ 14 \succ 23 \succ 24 \succ 34$. Assume that each of the three feasible duels is won by the team containing player 1 with a fixed probability $\gamma > 1/2$. Then, the learner has no chance of detecting the team $12$ as the top-$k$ team. However, any of the teams $12, 13$ and $14$ is a Condorcet winning team.

Our main target is to present algorithms for which the number of duels is bounded by a polynomial in the number of players $n$ and the team size $k$, although the number of teams is exponential in $k$, i.e., $\Omega((n/k)^k)$ and the number of valid duels is $\Omega(2^k(\frac{n}{2k})^{2k})$. Even if one were to accept an exponential number of arms, a direct reduction to the standard dueling bandits setting would not be feasible as not all pairs of teams are comparable in our model. In particular, duels of the form $(S \cup \{a\}, S \cup \{b\})$, which would yield a signal regarding the relation between players $a$ and $b$, are forbidden. The inherent difficulty of our endeavor comes from two limitations: (1) Not all the relations between two single players are deducible, (see example above), and (2) even for pairs of players with deducible relation, having $\Omega(2^k(\frac{n}{2k})^{2k})$ valid duels and the same amount of (latent) winning probabilities makes the task of deducing their relations hard.

We start by giving a full characterization of the *deducible* pairwise relations between players, namely relations that can be detected by an unlimited amount of duels. Our characterization implies that every deducible players relation has one of two types of *witnesses*, which are constant-size sets of duels that prove their relation. Once we find a witness for one pair of players, it can often be transformed to a witness for other pairs of players. Building upon this characterization, we introduce a parameter $\Delta_{a,b}$ which captures the distinguishability of any two players $a$ and $b$ and takes a value of $0$ whenever the pair is not deducible. Assuming $\Delta := \Delta_{k,k+1} > 0$, where $k$ and $k+1$ are $k^{\text{th}}$ and $(k+1)^{\text{th}}$ best players, we give a reduction to the classic dueling bandits problem. Combining this reduction with a high-probability top-$k$ identification algorithm for the dueling bandits setting (e.g., Mohajer et al. (2017); Ren et al. (2020)) yields a similar sample complexity upper bound, e.g., this implies a high-probability top-$k$ identification algorithm for dueling teams with $\mathcal{O}(\Delta^{-2}(n + k\log(k))\max(\log\log n, \log k))$ duels.

Interestingly, it turns out that the deterministic case, i.e., when winning probabilities are in $\{0, 1\}$, constitutes a challenging special case of our problem where $\Delta$ can be particularly small, or even $0$. To overcome this issue we design delicate algorithms which are independent of $\Delta$. On a high level, a preprocessing procedure first excludes as many bad players as possible. To do so, it runs a method for identifying pairwise relations between players which performs only a small number of duels, but has little control over the pair for which the relation is uncovered. For general total orders this implies an algorithm requiring $\mathcal{O}(nk\log(k) + 2^{\mathcal{O}(k)})$ duels. For the natural case of *additive linear* orders, we present a more elaborated approach for detecting a Condorcet winning team within the reduced instance, resulting in an algorithm that performs $\mathcal{O}(nk\log(k) + k^5)$ duels.

We introduce our model in Section 2, give a characterization of deducible relations in Section 3, and discuss the stochastic and deterministic setting in Sections 4 and 5, respectively. In Section 6 we provide a discussion including a lower bound and a regret bound. Algorithms and (full) proofs are relegated to Sections A-D in the appendix, and Section E characterizes additive linear total orders.

## 1.1 Related Work

**MAB best arm or subset identification** single arm identification was initiated in Even-Dar et al. (2006) and later studied in many works including Bubeck et al. (2011); Kaufmann et al. (2016); Chen et al. (2017). This setting was extended by Kalyanakrishnan & Stone (2010) for multiple arms identification (i.e., top $k$ arms), using a single arm samples. Other works that address the objective of top$-k$ identification include Chen et al. (2014); Zhou et al. (2014); Bubeck et al. (2013).

**Dueling bandits** The work of Yue et al. (2012) lay down the framework of non-parametric bandit feedback under total order among arms, strong stochastic transitivity, and stochastic triangle inequality assumptions and were followed by many subsequent works (For more, see a survey, Busa-Fekete et al. (2018).) In particular, some subsequent works target the task of identifying the top $k$ players in this setting Mohajer et al. (2017); Ren et al. (2020).

**Dueling bandits with sets of actions** One line of dueling bandits extension consider the case where the learner selects a subset of actions and observes the outcomes of all duels between all pairs of actions in the subset (Brost et al., 2016; Sui et al., 2017), or the winner of the subset Saha & Gopalan (2018); Ren et al. (2018). As a consequence, these settings give the learner strictly more information than the dueling bandits setting. In contrast, feedback in our setting reveals less information.

**MAB with multiple actions selection** There are works in which the learner selects a (sometimes fixed-sized) subset of actions at each iteration, and observes either all of the individual selected arms rewards (semi-bandit feedback) or an aggregated form of the rewards (full-bandit feedback), and the task is to detect to best arm or the top $k$. These include *combinatorial bandits* Cesa-Bianchi & Lugosi (2012), *top-k* Rejwan & Mansour (2020), *linear bandit and routing* Awerbuch & Kleinberg (2008), and more. The main difference between combinatorial bandits and our setting is the feedback.

**Comparison models** Noisy pairwise comparison models, especially for sorting and ranking, have a long history which dates backs to the 1950's (For more, see a survey, Pelc (2002).). Specifically, the mathematical problem Counterfeit coin was introduced in the form of a puzzle (Grossman, 1945): given a pile of 12 coins, determine which coins has a different weight (and therefore counterfeit) using balance scales while minimizing the number of measurements. The problem was followed by numerous generalizations (see Guy & Nowakowski (1995)). While this problem is restricted to coins with two different weights, our setting can be seen as a variant with multiple weights. Finally, a PAC related comparison-based model was studied in Balcan et al. (2016).

## 2 The Dueling Teams Problem

We formalize our problem as follows. Let $n, k \in \mathbb{N}$ with $1 \leq k \leq \frac{n}{2}$. We denote the set of players by $[n] := \{1, \ldots, n\}$ and call any set of $k$ distinct players a *team*. Moreover, we assume the existence of an underlying strict total order among all teams, and denote it by $\succ$. We also refer to $\succ$ as the ground truth order. In particular, for any two teams $A$ and $B$ either $A \succ B$ holds, in which case we say that $A$ is *better* than $B$, or vice versa, and this relation is transitive. Additionally, we require the total order among the teams to be consistent with a total order among players and formalize this in the *consistency* assumption at the end of this section.

In each round, the learner selects an ordered pair of two disjoint teams, $A$ and $B$ to perform a *duel*, and receives a noisy binary feedback about which team is better. Note that in contrast to the usual dueling bandits setting, our setting does not allow duels of the form $(A, A)$, as selecting teams with mutual players for a duel is not an option. We denote the (directly) *observable* part of $\succ$ by $\succ_{obs}$, i.e., $A \succ_{obs} B$ iff $A$ and $B$ are disjoint teams and $A \succ B$. Note that $\succ_{obs}$ is not transitive.

We write $A > B$ if team $A$ is the random winner of duel $(A, B)$. The probability $\Pr[A > B]$ is stationary and denoted by $P_{A,B} = \Pr[A > B]$. In each duel of team $A$ against team $B$ the outcome $A > B$ is sampled independently from a Bernoulli distribution with parameter $P_{A,B} = 1 - P_{B,A}$. We assume that the probabilistic comparisons are linked to the total order among the teams, i.e., $A \succ B$ implies $P_{A,B} > 1/2$, and that $P_{A,B}$ exists for every pair of teams (not only disjoint ones). In the deterministic setting, it holds that $P_{A,B} \in \{0, 1\}$ for any teams $A \neq B$. In other words, $A \succ_{obs} B$ iff the outcome of each duel $(A, B)$ is $A > B$, and for two disjoint teams $A$ and $B$ the learner can observe whether $A \succ_{obs} B$ or $B \succ_{obs} A$ by performing a single duel.

A team $A$ is a *Condorcet winning team*[2] if $A \succ_{obs} B$ for all teams $B$ such that $A \cap B = \emptyset$. From our assumption on $\succ$, there always exists a Condorcet winning team, but it is not necessarily unique. The learner's goal is to minimize the number of duels required to identify, with high probability in the stochastic setting and with probability 1 in the deterministic case, a Condorcet winning team.

In the following we formalize two more assumptions we impose on our model, the former affects the linking of the probabilities to the strict total order $\succ$, the latter restricts the total order $\succ$ itself.

**Strong stochastic transitivity (SST)** Similarly to the dueling bandits settings in Yue et al. (2012), we assume *strong stochastic transitivity*. Namely, for every triplet of different teams $A \succ B \succ C$ it holds that $P_{A,C} \geq \max\{P_{A,B}, P_{B,C}\}$.

**Consistency** We assume that the total order $\succ$ is consistent to a total order among single players. More precisely, we say that $\succ$ satisfies *consistency*[3] if for every two players $a, b \in [n]$ either of the following holds true:

(i) $S \cup \{a\} \succ S \cup \{b\}$ for all $S \subseteq [n] \setminus \{a, b\}, |S| = k - 1$.
(ii) $S \cup \{b\} \succ S \cup \{a\}$ for all $S \subseteq [n] \setminus \{a, b\}, |S| = k - 1$.

The consistency assumption lets us derive a relation among the single players, by defining $a \succ b$ iff $S \cup \{a\} \succ S \cup \{b\}$ holds for some $S$. By team relation transitivity, $\succ$ implies a total order on $[n]$. Whenever we write $a \succ b$ for some players $a, b \in [n]$ this is short-hand notation for $S \cup \{a\} \succ S \cup \{b\}$ for all subsets $S \subseteq [n] \setminus \{a, b\}$ of size $k - 1$. Though the consistency assumption might appear strong at first sight, dropping it would render the model intractable as the order among the teams would be independent of the players' identities. Namely, there is no linkage to the composition of the teams.[4]

For notational convenience, we assume without loss of generality that $1 \succ 2 \succ \cdots \succ n$ and write $A_m^*$ for the set of players containing the top $m$ players, i.e., $A_m^* = [m]$. In particular, the consistency assumption yields that $A_k^*$ is a Condorcet winning team. However, though the ground truth ranking induces a total order among the players, the learner might not be able to deduce the entire order. In the following we give a characterization of the *deducible* part of the ground truth order $\succ$.

# 3 Witnesses: A Characterization of Deducible Relations

In this section we provide a high level description of the complete characterization of all the pairwise relations between single players that can be deduced via team duels. We refer the reader to the full description of the characterization, which appears in Appendix A. Though single players relations cannot be sampled via feasible team duels directly as intersected teams cannot duel each other, we show a sufficient and necessary condition for deducible relations in the form of a constant number of winning probabilities of observable (feasible) duels. We refer to pairs of sets of players participating in such duels as *witnesses*.

We denote by $\mathbb{P}_{obs}$ the set of all tuples $(P', \succ')$, where each $P'$ is a team winning probability matrix that satisfy SST w.r.t. $\succ'$, which is a consistent strict total order on teams, and both $P'$ and $\succ'$ are compatible with the winning probabilities of observable duels and each other, i.e., $\{P'_{A,B} = P_{A,B} \mid A \text{ and } B \text{ are disjoint teams}\}$ and $P'_{A,B} = 1 - P'_{B,A} > 1/2$ iff $A \succ' B$. We remark that it follows directly from the definition of $\mathbb{P}_{obs}$ that $(P, \succ) \in \mathbb{P}_{obs}$, where $P$ is the ground truth winning probability matrix and $\succ$ the ground truth total order.

We denote by $\mathcal{C}_{obs}$ the set of strict total orders $\succ'$ for which there exists a tuple $(P', \succ') \in \mathbb{P}_{obs}$. Intuitively, a total order $\succ'$ is in $\mathcal{C}_{obs}$ if there exists an instance $(P', \succ')$ that has $\succ'$ as its ground truth total order, follows our assumptions and is compatible with the observable part of $(P, \succ)$. Lastly, we define $A \succ^* B$ if and only if $A \succ' B$ for all $\succ' \in \mathcal{C}_{obs}$, where $A$ and $B$ are not necessarily disjoint. We refer to $\succ^*$ as the *deducible* relation. For single player relations, we define $a \succ^* b$ if and only if

---

[2]The name is motivated by the fact that such a team is a weak Condorcet winner for the relation $\succ_{obs}$.

[3]*Consistency* is also reminiscent of *separable* preferences from economic theory (Breton & Sen, 1999).

[4]For example, the following instance would be possible: Players $n - k, ..., n$ are the $k$ 'intuitively worst players' (i.e., they lose in almost all duels) but together they form the best team (and a unique Condorcet winning team), and players $1, ..., k$ are the 'intuitively best players' and form the second best team in the linear order among teams. In such a model any algorithm would need to iterate over all teams. In other words, the problem becomes a dueling bandits problem with $\Omega((n/k)^k)$ arms.

there exists $S \subseteq [n] \setminus \{a,b\}$ such that $S \cup \{a\} \succ' S \cup \{b\}$ for all $\succ' \in \mathcal{C}_{obs}$. We stress that we only use $\mathbb{P}_{obs}$ and $\mathcal{C}_{obs}$ for analysis and never compute them.

Next, we define two sets of *potential witnesses* that have a simple structure and, in some cases, allow us to deduce single players relation: (1) A *potential subsets witnesses* set, denoted by $\mathcal{S}_{a,b}$, that contains all pairs $(S, S')$ such that $S$ and $S'$ are disjoint subsets of $[n] \setminus \{a,b\}$ and both are of size $k-1$, and (2) A *potential subset-team witnesses* set, denoted by $\mathcal{T}_{a,b}$, that contains all pairs $(S,T)$ where $S$ and $T$ are disjoint subsets of $[n] \setminus \{a,b\}$, such that $S$ is of size $k-1$ and $T$ is of size $k$ (and is therefore a team). Below, we define under which conditions a potential witness is indeed a *witness*.

**Definition 3.1.** *An element* $(S, S') \in \mathcal{S}_{a,b}$ *is a* subsets witness *for* $a \succ b$ *if* $P_{S \cup \{a\}, S' \cup \{b\}} > P_{S \cup \{b\}, S' \cup \{a\}}$. *An element* $(S, T) \in \mathcal{T}_{a,b}$ *is a* subset-team witness *for* $a \succ b$ *if* $P_{S \cup \{a\}, T} > P_{S \cup \{b\}, T}$.

We capture the set of the elements of $\mathcal{S}_{a,b}$ that are subsets witnesses for $a \succ b$ by $\mathcal{S}^*_{a,b}$ and analogously, $\mathcal{T}^*_{a,b} = \{(S,T) \in \mathcal{T}_{a,b} \mid (S,T)$ is a subset-team witness for $a \succ b\}$. It might be the case that $\mathcal{S}^*_{a,b} \cup \mathcal{T}^*_{a,b}$ is empty, in particular this holds when $b \succ a$. It is also possible that both $\mathcal{S}^*_{a,b} \cup \mathcal{T}^*_{a,b}$ and $\mathcal{S}^*_{b,a} \cup \mathcal{T}^*_{b,a}$ are empty, in which case we will show that the relation between players in $a$ and $b$ cannot be deduced. The following theorem implies that the other direction is also true.

**Theorem 3.2.** *Let* $a, b \in [n]$. *Then,* $a \succ^* b$ *if and only if* $\mathcal{S}^*_{a,b} \cup \mathcal{T}^*_{a,b} \neq \emptyset$.

*Proof sketch.* Assume that $\mathcal{S}^*_{a,b} \cup \mathcal{T}^*_{a,b} \neq \emptyset$. We show that $a \succ^* b$ by using SST, the fact that $\succ$ is a consistent strict total order, and an exhaustive case analysis. For the sake of illustration we present only one case here, namely, that $(S, S') \in \mathcal{S}^*_{a,b}$ and that both (1) $S \cup \{a\} \succ S' \cup \{b\}$ and (2) $S \cup \{b\} \succ S' \cup \{a\}$ hold. Assume for contradiction that $a \succ^* b$ does not hold. It thus follows that there exists an order, $\succ' \in \mathcal{C}_{obs}$ for which $b \succ' a$ holds. Let $P'_{A,B}$ be the corresponding winning probabilities. Then, using consistency of $\succ'$ and (1) respectively, we get $S \cup \{b\} \succ' S \cup \{a\} \succ' S' \cup \{b\}$ and from SST $P'_{S \cup \{b\}, S' \cup \{b\}} \geq P'_{S \cup \{a\}, S' \cup \{b\}} > 1/2$. In addition, applying consistency again, it follows that $S \cup \{b\} \succ S' \cup \{b\} \succ S' \cup \{a\}$. Applying SST once more we get $P'_{S \cup \{b\}, S' \cup \{a\}} \geq P'_{S \cup \{b\}, S' \cup \{b\}} \geq P'_{S \cup \{a\}, S' \cup \{b\}}$, a contradiction to $(S, S') \in \mathcal{S}^*_{a,b}$ (since this implies $P_{S \cup \{a\}, S' \cup \{b\}} > P_{S \cup \{b\}, S' \cup \{a\}}$).

For the other direction we start by defining $\mathcal{D}_a$ as the set of observable duels $(A, B)$ such that $a \in A$. Moreover, we define a permutation $\pi$ on the set of teams, which simply exchanges the players $a$ and $b$ when present. We then show that $a \succ b$ implies $P_{A,B} \geq P_{\pi(A),\pi(B)}$ for all $(A, B) \in \mathcal{D}_a$. Moreover, we show that $a \succ^* b$ implies that there exists $(A, B) \in \mathcal{D}_a$ with $P_{A,B} > P_{\pi(A),\pi(B)}$ as follows. Assume not. Then we show that the relation $\succ'$ defined by $A \succ' B$ iff $\pi(A) \succ' \pi(B)$ is included in $C_{obs}$. However, $a \succ^* b$ implies that for any $S \subseteq [n] \setminus \{a,b\}$ of size $k-1$ it holds that $S \cup \{a\} \succ^* S \cup \{b\}$ which implies $(i)$ $S \cup \{a\} \succ S \cup \{b\}$ as well as $(ii)$ $S \cup \{a\} \succ' S \cup \{b\}$. Applying the definitions of $\succ'$ and $\pi$, statement $(ii)$ implies $S \cup \{b\} = \pi(S \cup \{a\}) \succ \pi(S \cup \{b\}) = S \cup \{a\}$ and hence yields a contradiction to $(i)$. Finally, take some $(A, B) \in \mathcal{D}_a$ with $P_{A,B} > P_{\pi(A),\pi(B)}$. If $b \in B$, then $(A \setminus \{a\}, B \setminus \{b\}) \in \mathcal{S}^*_{a,b}$, otherwise $(A \setminus \{a\}, B) \in \mathcal{T}^*_{a,b}$. $\square$

For the sake of brevity, we introduce the set $\mathcal{X}_{a,b}$ which combines the pairs from $\mathcal{S}_{a,b}$ and $\mathcal{T}_{a,b}$ into a set of triples. Formally, $\mathcal{X}_{a,b} = \{(S, S', T) \mid (S, S') \in \mathcal{S}_{a,b}, (S, T) \in \mathcal{T}_{a,b}\}$. We say that $(S, S', T)$ is a witness for $a \succ b$ if $(S, S') \in \mathcal{S}^*_{a,b}$ or $(S, T) \in \mathcal{T}^*_{a,b}$, and denote $(S, S', T) \in \mathcal{X}^*_{a,b}$.

# 4  Stochastic Setting

In this section we focus on algorithms identifying, with high probability, the top-$k$ team, which is in particular a Condorcet winning team. The main idea is to utilize the results from the previous section to reduce the dueling teams setting to the classic dueling bandits setting (Yue et al. (2012)). To this end we will introduce our *gap parameter*, $\Delta$, which intuitively captures how easy it is to prove the relationship between the top $k$ and the top $(k+1)$ player. We start by defining, for any element $(S, S', T) \in \mathcal{X}_{a,b}$, a random variable $X_{a,b}(S, S', T)$ that combines the outcomes of four duels that helps determining whether $(S, S', T)$ is a witness for $a \succ^* b$. Formally,

$$X_{a,b}(S, S', T) = \big(\mathbb{1}[S \cup \{a\} > S' \cup \{b\}] - \mathbb{1}[S \cup \{b\} > S' \cup \{a\}]$$
$$+ \mathbb{1}[S \cup \{a\} > T] - \mathbb{1}[S \cup \{b\} > T]\big)/4.$$

Observe that $X_{a,b}(S, S', T)$ can take values from $\{-1/2, -1/4, 0, 1/4, 1/2\}$, thus $\mathbb{E}[X_{a,b}(S, S', T)] \in [-1/2, 1/2]$. Moreover, we have the following properties:

1. For every $(S, S', T) \in \mathcal{X}_{a,b}$, it holds that $\mathbb{E}[X_{a,b}(S, S', T)] > 0 \iff (S, S', T) \in \mathcal{X}_{a,b}^*$.

2. If $\mathbb{E}[X_{a,b}(S, S', T)] = 0$ for every $(S, S', T) \in \mathcal{X}_{a,b}$, then Theorem 3.2 implies that the pairwise relation between players $a, b$ cannot be deduced.

Building upon the random variables $X_{a,b}(S, S', T)$, which are defined for a fix pair of players, $a, b$, and for each element in $\mathcal{X}_{a,b}$, we define a single random variable $X_{a,b}$ by picking a random triplet $(S, S', T) \in \mathcal{X}_{a,b}$ and returning a realization of $X_{a,b}(S, S', T)$. For convenience, whenever we write $\mathbb{E}[X_{a,b}]$ we mean $\mathbb{E}_{(S,S',T) \sim \mathcal{X}_{a,b}}[X_{a,b}]$. Using the probabilistic method, we obtain the following theorem, which then brings us to the definition of a gap parameter for our problem.

**Theorem 4.1.** *For every two players $a, b \in [n]$ it holds that $a \succ^* b$ if and only if $\mathbb{E}[X_{a,b}] > 0$.*

**Gap parameter** We define our gap parameter by $\Delta := \mathbb{E}[X_{k,k+1}]$. Another interpretation for $\Delta$ is that $\Delta = 1/2(\alpha + \beta - 1)$, where $\alpha \in [0, 1]$ is the probability that the top $k$ player wins against the top $k + 1$ player, when the two players are placed with a random potential subsets witness from $\mathcal{S}_{k,k+1}$ (i.e., disjoint random subsets of $k - 1$ players) and $\beta \in [0, 1]$ captures the difference in winning probabilities when $k$ and $k + 1$ are placed with a random potential subset-team witness from $\mathcal{T}_{k,k+1}$ (i.e., two disjoint random subsets of sizes $k - 1$ and $k$). Within Appendix B we show that for $k = 1$, this gap is at least of the same order as the gap parameter for the classical dueling bandit setting.

In the following we show that our gap parameter does not just help us to distinguish between the top $k$ and $(k + 1)$ players, but also allows us to distinguish other players in $A_k^*$ and players from $[n] \setminus A_k^*$. To this end, we show in Lemma 4.2 that the expectations $\mathbb{E}[X_{a,b}]$ satisfy strong stochastic transitivity w.r.t. the ground truth total order on players. We note that most elements $(S, S', T) \in \mathcal{X}_{a,b}$ hold $\mathbb{E}[X_{a,c}(\pi(S), \pi(S'), \pi(T))] \geq \mathbb{E}[X_{a,b}(S, S', T)]$ (and analogously for $X_{b,c}$), where $\pi$ is a permutation exchanging players $b$ and $c$, but, surprisingly, this is not true in general. By carefully constructing a charging scheme, we manage to show that this holds in expectation over all elements of $\mathcal{X}_{a,b}$, and derive strong stochastic transitivity for the distinguishabilities of players.

**Lemma 4.2.** *For a triplet of players $a \succ b \succ c$ it holds that*

$$\mathbb{E}[X_{a,c}] \geq \max\{\mathbb{E}[X_{a,b}], \mathbb{E}[X_{b,c}]\}.$$

This also yields the following theorem, which paves the way for our reduction in what follows.

**Theorem 4.3.** *For any $a, b \in [n]$ with $a \in A_k^*, b \notin A_k^*$ it holds that $\mathbb{E}[X_{a,b}] \geq \mathbb{E}[X_{k,k+1}] = \Delta$. Thus, if $\Delta > 0$ and for a team $A$ it holds that $\mathbb{E}[X_{a,b}] \geq \Delta$ for all $a \in A, b \in [n] \setminus A$, then $A = A^*$.*

**The reduction** We now outline the gap-dependent algorithm. The results we have derived in Section 3 will allow us to deduce, with high probability, whether a distinguishability of a given pair of players is at least $\Delta$, and if so determine which is the better player. Intuitively, this is done by performing $\mathcal{O}(\frac{1}{\Delta^2})$ team duels. We use $\mathbb{E}[X_{a,b}]$ as a proxy for the distinguishability between two single players, $a, b$, taking advantage of the fact that if their relation is deducible, then $\mathbb{E}[X_{a,b}] \neq 0$ and in this case $\mathbb{E}[X_{a,b}] > 0$ iff $a \succ b$. Similar to the dueling bandits setting, even though $|\mathbb{E}[X_{a,b}]| < \Delta$ for some pairs of players, identifying $A_k^*$ with high probability is possible.

Since we cannot directly sample $X_{a,b}$, we will instead sample uniformly at random a triplet of sets, $(S, S', T)$ from $\mathcal{X}_{a,b}$. Using $(S, S') (\in \mathcal{S}_{a,b})$ and $(S, T) (\in \mathcal{T}_{a,b})$, we can then perform all the duels required for an unbiased sample of $X_{a,b}(S, S', T)$, which is by itself a sampling of $X_{a,b}$. Given any dueling teams instance, we define a dueling bandits instance as follows: for every two players $a, b \in [n]$, we define the probability that $a$ wins in a (singles) duel against $b$ as

$$P_{a,b} = 1/2 + \mathbb{E}[X_{a,b}]. \tag{1}$$

Clearly, $1 - P_{a,b} = P_{b,a}$, $P_{a,b} \in [0, 1]$ and $P_{a,b} > 1/2$ implies $a \succ b$. In addition, Theorem 4.1 implies that $a$ is better than $b$ in this dueling bandits instance iff $a \succ^* b$. So whenever a dueling bandits algorithm is asking for a duel query, $(a, b)$, we can make an independent sample of $X_{a,b}$ by randomly drawing a triplet $(S, S', T) \in \mathcal{X}_{a,b}$ and returning a random sampling of $X_{a,b}(S, S', T) + 1/2$. In cases where the realization of $X_{a,b}(S, S', T) + 1/2$ is in $\{1/4, 1/2, 3/4\}$, we assign $a$ as the duel winner if the result of flipping a coin with bias $X_{a,b}(S, S', T) + 1/2$ is 1. We formalize this idea in the sub-procedure *singlesDuel* (in the appendix), that simulates a duel for classical dueling bandits

settings using team duels. Notice that, by Lemma 4.2, the probabilities $P_{a,b}$ defined in (1) satisfy SST with respect to the total order among the players induced by the ground truth order $\succ$. In addition, the feedback of each single player duel we perform is time-invariant, thus all the non-parametric assumptions for dueling bandits settings apply here. The reduction allows us to identify the top-$k$ players using any dueling bandit algorithm with the same goal that works for total order on arms that satisfy SST, and a gap between the top $k$ and $(k+1)$ arms as assumptions. We formalize this below.

**Theorem 4.4.** *Given any dueling teams instance with $n$ and $k$ (namely, $P_{A,B}$ for every two teams that hold strict total order, SST, and consistency), we have that the dueling bandit instance defined by (1) satisfies SST with respect to the ground truth order among players $\succ$ and for any two players $a \succ b$ it holds that $P_{a,b} \geq 1/2$. Moreover, $P_{k,k+1} = 1/2 + \Delta$.*

Using the above theorem we can use any dueling bandit algorithm for top-$k$ identification to solve our problem. Mohajer et al. (2017) provide an algorithm that returns the top-$k$ players with probability exceeding $1 - (\log n)^{-c_0}$ with sample complexity at most $c_1 \Delta_{k,k+1}^{-2}(n + k \log k) \max (\log \log n, \log k)$ in expectation, where $c_0$ and $c_1$ are universal positive constants and $\Delta_{k,k+1}$ is the distinguishability between the $k$ and the $(k+1)$ best players (see Algorithm 2 and Theorem 1 in Mohajer et al. (2017)).

Ren et al. (2020) show an algorithm that returns the top-$k$ players with probability at least $1 - \delta$ with sample complexity $\mathcal{O}(\sum_{i \in [n]}(\Delta_i^{-2}(\log(n/\delta) + \log \log \Delta_i^{-1})))$, where $\Delta_i = \mathbb{1}_{i \succ k+1} \cdot \Delta_{i,k+1} + \mathbb{1}_{k \succ i} \cdot \Delta_{k,i}$ and $k, k+1$ are the top $k$ and the top $(k+1)$ players, respectively (see Algorithm 5 and Theorem 8 in Ren et al. (2020))[5]. These algorithms, together with Theorem 4.1 allow us to derive the following theorem.

**Theorem 4.5.** *There exists an algorithm that returns $A_k^*$ with probability exceeding $1 - (\log n)^{-c_0}$ with sample complexity at most $c_1(n + k \log k)\frac{\max (\log \log n, \log k)}{\Delta^2}$ in expectation, where $c_0$ and $c_1$ are universal positive constants. In addition, there exists an algorithm that returns $A_k^*$ with probability at least $1 - \delta$ with sample complexity $\mathcal{O}(\sum_{i \in [n]}(\Delta_i^{-2}(\log(n/\delta) + \log \log \Delta_i^{-1})))$, where $\Delta_i = \mathbb{1}_{i \succ k+1} \cdot \mathbb{E}[X_{i,k+1}] + \mathbb{1}_{k \succ i} \cdot \mathbb{E}[X_{k,i}]$ and $i$ denotes the top $i$ player, thus $\Delta_i \geq \Delta$ for all $i \in [n]$.*

## 5 Deterministic Setting

In Section 4 we showed the existence of algorithms that identify the top-$k$ team with a number of duels that depends on $\Delta$. But what if $\Delta$ is very small or even 0? One reason for that can be that all relevant probabilities (e.g., $P_{\{k\} \cup S,T}$, $P_{\{k+1\} \cup S,T}$) are very close to $1/2$ for all $(S, S', T) \in \mathcal{X}_{k,k+1}^*$. Then, a high number of duels to identify $A_k^*$ is unavoidable and giving upper bounds in dependence of a gap parameter very much resembles the literature on dueling bandits, where a gap between the top $k$ and $k+1$ players is often a parameter of the sample complexity (e.g., Mohajer et al. (2017)). Another reason for $\Delta$ to be small is that the number of witnesses for $k \succ k+1$ is small. We overcome this issue by designing $\Delta$-independent algorithms, assuming deterministic feedback, i.e., $P_{A,B} \in \{0, 1\}$ for all pairs of teams $(A, B)$. In this setting, $(S, T) \in \mathcal{T}_{a,b}^*$ if and only if $S \cup \{a\} \succ_{obs} T \succ_{obs} S \cup \{b\}$, and $(S, S') \in \mathcal{S}_{a,b}^*$ if and only if $S \cup \{a\} \succ_{obs} S' \cup \{b\}$ and $S' \cup \{a\} \succ_{obs} S \cup \{b\}$. In the appendix (Section C) we show that our results extend to a slightly stochastic environment.

The limitation of the set of witnesses makes the task of identifying a Condorcet winning team in the deterministic setting surprisingly hard. For general total orders, a crucial difficulty lies in proving that a given team is indeed Condorcet winning. Nevertheless, we are able to get the following result:

**Theorem 5.1.** *For deterministic feedback, there exists an algorithm that performs $\mathcal{O}(kn \log(k) + k^2 \log(k)2^{5k})$ duels and outputs a Condorcet winning team.*

For the natural special case of *additive total orders* we obtain a significantly better upper bound. A total order $\succ$ is *additive total*, if there exist values for the players denoted by $v(x), x \in [n]$ such that $A \succ B$ iff $\sum_{a \in A} v(a) > \sum_{b \in B} v(b)$. In Section E of the appendix we give a sufficient and necessary condition for a total order to be additive. For additive total orders we present an algorithm that identifies a Condorcet winning team after polynomial many duels and also outputs a proof.

**Theorem 5.2.** *For deterministic feedback and additive total orders, there exists an algorithm that finds a Condorcet winning team within $\mathcal{O}(kn \log(k) + k^5)$ duels.*

---

[5]We remark that Ren et al. (2020) also assume Stochastic triangle inequality which we do not, however it is only used to derive a lower bound.

Both algorithms rely on the same preprocessing procedure called *ReducePlayers* which reduces the number of players from $n$ to $\mathcal{O}(k)$. At the heart of this procedure is a subroutine called *Uncover*. After describing *Uncover* and *ReducePlayers*, we prove Theorem 5.1. Towards proving Theorem 5.2, we introduce two more subroutines, namely *NewCut* and *Compare*, which are crucial for identifying and proving a Condorcet winning team within the smaller instance. Finally, Algorithm *CondorcetWinning* combines all components and proves Theorem 5.2. While all algorithms are formalized in Appendix C, we give sketches thereof and theorems stating their input and output below.

**The Uncover Subroutine**   Given two disjoint teams $A \succ B$, the *Uncover* subroutine finds a pair of players $a \in A$ and $b \in B$ and a subsets witness for their relation, i.e., an element from $\mathcal{S}^*_{a,b}$. To understand the idea of the subroutine, consider some arbitrary ordering of the elements in $A$ and $B$, respectively, i.e., $A = \{a_1, \dots, a_k\}$ and $B = \{b_1, \dots, b_k\}$. Then, iteratively exchange the elements $a_1$ and $b_1$, $a_2$ and $b_2$, resulting in sets $A_0 = A, B_0 = B, A_1 = \{b_1, a_2, \dots, a_k\}, B_1 = \{a_1, b_2, \dots, b_k\}, A_2 = \{b_1, b_2, a_3, \dots, a_k\}$, and so on. Since $A_0 \succ B_0$ but $A_0 = B_k \succ A_k = B_0$ holds, there needs to be some earliest point in time $i \le k$ for which $B_i \succ A_i$ is true. This implies $a_i \succ b_i$ as $(\{a_1, \dots, a_{i-1}, b_{i+1}, \dots, b_k\}, \{b_1, \dots, b_{i-1}, a_{i+1}, \dots, a_k\})$ is a subsets witness.

While the above sketched subroutine is simple, it performs $k$ duels in the worst case. We refine this idea by a binary search approach, decreasing the number of duels to $\log(k)$.

**Lemma 5.3.** *Let $A$ and $B$ be two disjoint teams with $A \succ B$. After performing $\mathcal{O}(\log(k))$ duels,* Uncover *returns $(a, b)$ with $a \in A$, $b \in B$ and $(S, S') \in \mathcal{S}^*_{a,b}$, and thus $a \succ b$.*

We remark that Lemma C.2 in the appendix is a slightly stronger version of the above lemma which allows us to partition $A$ and $B$ into two subsets each, $A = A^{(1)} \cup A^{(2)}$ and $B = B^{(1)} \cup B^{(2)}$. Under some circumstances, we can then guarantee that *Uncover* reveals the pairwise comparison between two players $a \succ b$, where $a$ is from $A^{(1)}$ and $b$ is from $B^{(1)}$.

**Reducing the Number of Players to $\mathcal{O}(k)$**   The fact that we can eliminate some players from $[n]$ and still find (and prove) a Condorcet winning team is due to the following observation.

**Observation 5.4.** *Let $R \subseteq [n]$ such that $A^*_{2k} \subseteq R$. Let $\hat{A} \subseteq R$ be a team such that $\hat{A} \succ A$ for all teams $A \subseteq R \setminus \hat{A}$. Then, $\hat{A}$ is a Condorcet winning team.*

The procedure *ReducePlayers* reduces the set of players $[n]$ to some subset $R \subseteq [n]$ guaranteeing that $A^*_{2k} \subseteq R$ and $|R| < 6k$. The algorithm maintains a dominance graph $D = (V, E)$ on the set of players. More precisely, the nodes of $D$ are the players, i.e., $V = [n]$, and there exists an arc from node $a$ to node $b$ if the algorithm has proven that $a \succ b$. The set $V_{<2k}$ is the subset of the players having an indegree smaller than $2k$ in $D$. The high level idea of the algorithm is the following: It starts with the empty dominance graph $D = ([n], \emptyset)$. Then, it iteratively identifies pairwise relations of the players with help of *Uncover* and adds the respective arcs to the graph. By adding more and more arcs to $D$, the set of nodes $V_{<2k}$ shrinks while $A^*_{2k} \subseteq V_{<2k}$ is always guaranteed. At some point, the algorithm cannot identify any more pairwise relations and returns $V_{<2k}$. How does the algorithm identify pairwise relations? At any point it tries to find a matching between $2k$ players, say $\{(a_1, b_1), \dots, (a_k, b_k)\}$ with the constraint that, for all $i \in [k]$, none of the arcs $(a_i, b_i)$ or $(b_i, a_i)$ is present within the graph $D$ yet, and then applies *Uncover* on the sets $A = \{a_1, \dots, a_k\}$ and $B = \{b_1, \dots, b_k\}$. In the proof of Lemma 5.5 we show that, as long as $|V_{<2k}| \ge 6k - 1$, the algorithm can find such a matching. With the help of Lemma 5.5, we then prove Theorem 5.1.

**Lemma 5.5.** *Given the set of players $[n]$,* ReducePlayers *returns $R \subseteq [n]$ with $|R| \le 6k - 2$ and $A^*_{2k} \subseteq R$.* ReducePlayers *performs $\mathcal{O}(nk \log(k))$ duels and runs in time $\mathcal{O}(n^2 k^2)$.*

*Proof sketch (of Theorem 5.1).* Let $D$ be the dominance graph at the end of *ReducePlayers*. Then, the learner selects a $k$-sized subset of $V_{<2k}$, call it $\hat{A}$, with the property that there is no arc from any node in $V_{<2k} \setminus \hat{A}$ towards some node in $\hat{A}$. Then, the learner tests $\hat{A}$ against all possible teams containing players from $V_{<2k} \setminus \hat{A}$, which are $\mathcal{O}(2^{5k})$ many. If $\hat{A}$ wins all of these duels, then $\hat{A}$ is a Condorcet winning team by Observation 5.4. However, if there exists $A \succ \hat{A}$, then, by the choice of $\hat{A}$, there does not exist any arc from $A$ towards $\hat{A}$. Hence, by calling the subroutine *Uncover* for two arbitrary orderings of $A = \{a_1, \dots, a_k\}$ and $\hat{A} = \{\hat{a}_1, \dots, \hat{a}_k\}$, the learner will identify one additional arc. This procedure can be repeated $\mathcal{O}(k^2)$ times and thus shows Theorem 5.1. □

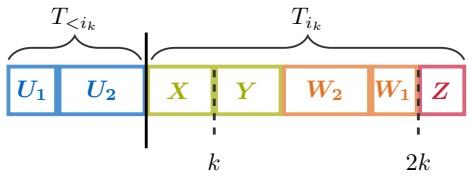 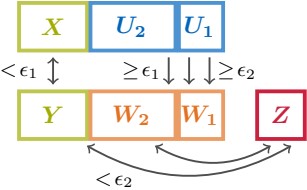

Figure 1: Illustration of the proof technique of algorithm *CondorcetWinning1*. In the left illustration, the solid black line indicates that all players left to it were proven to be better than all players right to it. The dashed line marked with "$k$" indicates that the sets to its left contain $k$ players in total. However, this line does not indicate proven relations, i.e., players from $X$ are not necessarily better than players from $Y$. The right figure illustrates the proof for $X \cup U_1 \cup U_2$ being Condorcet winning.

**Subroutines NewCut and Compare** The *NewCut* subroutine takes as input a subset of the players $R \subseteq [n]$, a pair $a, b \in R$, and a witness proving that $a \succ b$, i.e., $(S, T') \in \mathcal{S}^*_{a,b} \cup \mathcal{T}^*_{a,b}$. That means, $T'$ can be either of size $k - 1$ or $k$, and $S$ and $T'$ are not required to be subsets of $R$. The subroutine outputs a partition of $R$ into two non-empty sets $U$ and $L$ with $U \triangleright L$, which is short-hand notation for $u \succ \ell$ for any $u \in U$ and $\ell \in L$. The subroutine starts by initiating the set $U = \{a\}$ and redefines $R = R \setminus \{a, b\}$. At all times, $U$ contains only players $u$ for which the algorithm has found a witness for $u \succ b$. These witnesses are stored in a list $\mathcal{W}$, and it is checked whether they can be modified to become witnesses for $x \succ b$ for any other element in $x \in R$. This modification is done by applying permutations on the set of subsets of the players, similarly as in the proof of Theorem 3.2 and Lemma 4.2. If the algorithm finds a witness for $x \succ b$, then $x$ is added to $U$ and removed from $R$. The new witness is also stored in $\mathcal{W}$. This process ends when either $R$ is empty or all witnesses in $\mathcal{W}$ have been checked. At this point it holds that $U \triangleright R \cup \{b\}$, and the algorithm returns $(U, L := R \cup \{b\})$.

**Lemma 5.6.** *Let $R \subseteq [n]$, $a, b \in R$ and $(S, T') \in \mathcal{S}^*_{a,b} \cup \mathcal{T}^*_{a,b}$. Then, $\mathrm{NewCut}(R, (a, b), (S, T'))$ returns a partition of $R$ into $U$ and $L$ such that $U \triangleright L$, $a \in U$ and $b \in L$. The number of duels performed by* NewCut *and its running time can be bounded by $\mathcal{O}(|R|^2)$.*

From now on we assume additive linear orders. The *compare* subroutine is crucial for obtaining upper bounds for differences between values of players' subsets. It is used in the following situation. Let $(a, b)$ be a pair of players and $(S, S') \in \mathcal{S}^*_{a,b}$ be a witness for $a \succ b$. Then, it can be easily shown that $v(a) - v(b) > |v(S) - v(S')|$. We will be interested in the question whether a similar relation holds for two subsets of $S$ and $S'$, namely, $C \subseteq S$ and $D \subseteq S'$ of equal size. The *compare* subroutine checks whether such a relation holds by performing two additional duels. If it returns *True*, then $v(a) - v(b) > |v(C) - v(D)|$. Otherwise, there can be found a pair $c \in C$ and $d \in D$ and a witness for their relation by one call to the *Uncover* subroutine. This observation is formalized below.

**Lemma 5.7.** *Let $a \succ b$ be two players, $(S, S') \in \mathcal{S}^*_{a,b}$ and $C \subseteq S, D \subseteq S'$ with $|C| = |D|$. If* Compare$((a, b), (S, S'), (C, D))$ *returns* True, *then $v(a) - v(b) > |v(C) - v(D)|$. Otherwise, one call to* Uncover *returns $c \in C$ and $d \in D$ together with a witness for their relation.*

**Algorithm CondorcetWinning** The algorithm maintains a partition of the players into a weak ordering, i.e., $\mathcal{T} = \{T_1, \ldots, T_\ell\}$ with $T_1 \triangleright T_2 \triangleright \cdots \triangleright T_\ell$. We introduce the short-hand notation $T_{\leq j} = \bigcup_{m \in [j]} T_m$ and $T_{<j} = \bigcup_{m \in [j-1]} T_m$. After the application of the preprocessing procedure *ReducePlayers*, this partition consists of one set, namely $\mathcal{T} = \{T_1\}$, where $|T_1| \in \mathcal{O}(k)$ and $A^*_{2k} \subseteq T_1$. At any point in the execution of the algorithm, we are especially interested in two indices, namely $i_k \in [\ell]$ such that $|T_{<i_k}| < k < |T_{\leq i_k}|$ and similarly $i_{2k} \in [\ell]$ such that $|T_{<i_{2k}}| < 2k < |T_{\leq i_{2k}}|$. [6] Observe that all players from $T_{<i_k}$ are guaranteed to be among the top-$k$ players. On the other hand, among the players from $T_{i_k}$ some belong to $A^*_k$ and others do not. The main idea of the algorithm is then the following: Take a prefix of $\mathcal{T}$ of size $k$, i.e., this team contains the set of players $T_{<i_k}$ and is a subset of the players in $T_{\leq i_k}$, and either prove that this prefix is a Condorcet Winning team, or refine the partition $\mathcal{T}$ and repeat the process. The refinement is done by splitting one element of $\mathcal{T}$, say $T_i$, into two non-empty sets, $T_i^1 \triangleright T_i^2$, and re-indexing the sets within $\mathcal{T}$. This is done with the help of *Uncover* and *NewCut*. Clearly, this increases the number of sets in the partition $\mathcal{T}$ by one.

---

[6]In case one of these indices does not exist, it implies that we have either identified the set $A^*_k$ or $A^*_{2k}$. In the first case we have found a Condorcet winning team and in the second case Observation 5.4 implies that we can find one by performing one additional duel. For the sake of brevity we disregard this case from now on.

We provide two different algorithms, namely *CondorcetWinning1* for the case $i_k = i_{2k}$ and *CondorcetWinning2* when $i_k \neq i_{2k}$. Unsurprisingly, the latter case requires a strictly less sophisticated approach, which is why we focus on *CondorcetWinning1* in the following.

The algorithm starts by partitioning the set $T_{<i_k}$ into two sets $U_1$ and $U_2$, where $U_1$ is a prefix of $T_{<i_k}$ of size $|T_{\leq i_k}| - 2k$. It partitions the set $T_{i_k}$ into five sets $X, Y, W_1, W_2$, and $Z$. In particular it is known that $(U_1 \cup U_2) \triangleright (X \cup Y \cup W_1 \cup W_2 \cup Z)$ but no relation among any pair in $T_{i_k}$ is known. Regarding the sizes of the sets it holds that $|U_i| = |W_i|$ for $i \in \{1, 2\}$, $|X| = |Y| = k - |U_1| - |U_2|$ and $|U_1| = |Z|$. The main aim of the algorithm will be to define $0 < \epsilon_1 < \epsilon_2$ and prove that the following statements are true:

(i) $|v(X) - v(Y)| < \epsilon_1$

(ii) $|v(a) - v(b)| < \epsilon_2$ for all $a \in Y \cup W_1 \cup W_2$ and $b \in Z$, and

(iii) there exist $u_1, \ldots, u_{|Z|+1} \in U_1 \cup U_2$ as well as $w_1, \ldots, w_{|Z|+1} \in W_1 \cup W_2$ such that

    (a) $v(u_1) - v(w_1) \geq \epsilon_1$ and

    (b) $v(u_i) - v(w_i) \geq \epsilon_2$ for all $i \in \{2, \ldots, |Z|+1\}$.

With these statements we show that $U_1 \cup U_2 \cup X$ is a Condorcet winning team. More precisely, we show $v(U_1 \cup U_2 \cup X) - v(W_1 \cup W_2 \cup Y) > |Z| \cdot \epsilon_2$ and $v(W_1 \cup W_2 \cup Y) - v(B^*) > -|Z| \cdot \epsilon_2$, where $B^*$ is the best response[7] towards $U_1 \cup U_2 \cup X$. See Figure 1 for an illustration of the argument.

It remains to sketch how the algorithm defines $\epsilon_1, \epsilon_2$ and proves $(i) - (iii)$. For simplicity assume $U_1 \triangleright U_2$. The algorithm then attempts to do the following steps: (1) Find a witness for players $\bar{u} \in U_2$ and $\bar{w} \in W_2$, using *Uncover*. (2) Use *Compare*, to prove that $|v(X) - v(Y)| < v(\bar{u}) - v(\bar{w})$ and $|v(a) - v(b)| < v(\bar{u}) - v(\bar{w})$ holds for all players $a \in W_1 \cup W_2 \cup Y$ and $b \in Z$. (3) Repeat step (2) by replacing $\bar{w}$ with any player of $W_1$. If one of the steps (1)-(3) fails, we show that the partition $\mathscr{T}$ can be refined. Otherwise, we show that $(i) - (iii)$ hold for $\epsilon_1 = v(\bar{u}) - v(w_1^*)$ and $\epsilon_2 = v(\bar{u}) - v(w_2^*)$, where $w_1^*$ and $w_2^*$ are the best and second best players from $W_1 \cup \{\bar{w}\}$, respectively. The following Lemma concludes the proof sketch of Theorem 5.2.

**Lemma 5.8.** *For every instance with $\mathcal{O}(k)$ players, after performing $\mathcal{O}(k^5)$ many duels,* CondorcetWinning1 *has identified a Condorcet winning team.* CondorcetWinning2 *identifies a Condorcet winning team after $\mathcal{O}(k^2 \log(k))$ duels.*

# 6 Extensions and Discussion

Below, we discuss several implications of our results as well as directions for future work.

**Regret bound** In this paper we provided algorithms to identify, with high probability, a Condorcet winning team. Another common performance metrics for the dueling bandits setting is (e.g., weak) regret. We define a regret in our setting w.r.t. $A_k^*$. This is reasonable since SST implies that $P_{A_k^*, B} \geq P_{\bar{A}, B}$ for every Condorcet winning team $\bar{A}$ and any team $B$. Then, for time horizon $T$ we denote by $(A_t, B_t)$ the selected duel at time $t$ and define the regret to be

$$R_T = \sum_{t=1}^{T} \min \{P_{A_k^*, A_t} - 1/2, P_{A_k^*, B_t} - 1/2\}.$$

This definition is based on weak regret for dueling bandits, as defined in Yue et al. (2012). Within Appendix D, we derive a regret bound of $R_T = \mathcal{O}(n(\Delta^{-2}(\log(T) + \log\log \Delta^{-1}))$.

**Checking Condorcet winners beyond additive linear orders** The question how many duels are necessary to prove or disprove that a given team is Condorcet winning remains open for total orders that are not additive linear, even if $n \in \mathcal{O}(k)$. Nevertheless, our algorithm from Theorem 5.1 shows that a polynomial upper bound for this number would imply the existence of an algorithm with a polynomial number of duels. More formally: Let $q$ be the number of duels required to check whether a given team is a Condorcet winning team within an instance with $\mathcal{O}(k)$ players. Then, there exists an algorithm that identifies a Condorcet winning team within $\mathcal{O}(kn \log(k) + k^2 log(k)q)$ duels.

**Lower bounds** For both settings, we can show a lower bound of $n - 2k$ duels in order to identify a Condorcet winning team. We refer to Appendix D for a formal statement and a proof.

---

[7]We call $B^*$ a best response towards $U_1 \cup U_2 \cup X$, if $B^*$ contains the best $k$ players from $[n] \setminus (U_1 \cup U_2 \cup X)$.

# 7 Acknowledgments

This project has received funding from the European Research Council (ERC) under the European Union's Horizon 2020 research and innovation program (grant agreement No. 882396), the Israel Science Foundation (grant number 993/17), the Yandex Initiative for Machine Learning at Tel Aviv University, the Deutsche Forschungsgemeinschaft under grant BR 4744/2-1, and the Ariane de Rothschild Women Doctoral Program.

This paper is dedicated to Hunter, a dear friend who passed away May 31, 2021.

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
