# Appendix

## A   Extended Version and Proofs of Section 3

Within the main text, we covered two different types of witnesses for single players relations. In this section, we show that whenever a relation between single players can be proven from observable duels in our setting, there exists at least one type of witness for it. For the convince of the reader, we recall the definitions mentioned in the main text in a comprehensive manner, provide more explanations and some examples.

**Possible witnesses**   For two players $a$ and $b$ we define $\mathcal{S}_{a,b}$ as the set of pairs of disjoint $k-1$ sized subsets of players from $[n] \setminus \{a, b\}$, i.e.,

$$\mathcal{S}_{a,b} = \{(S, S') \mid S, S' \subseteq [n] \setminus \{a, b\}, S \cap S' = \emptyset, |S| = |S'| = k - 1\},$$

and $\mathcal{T}_{a,b}$ as the set of disjoint $k-1$ sized subset $S$ and a team $T$ pair from $[n] \setminus \{a, b\}$, i.e.,

$$\mathcal{T}_{a,b} = \{(S, T) \mid S, T \subseteq [n] \setminus \{a, b\}, S \cap T = \emptyset, |S| = k - 1, |T| = k\}.$$

**Definition A.1** (Witnesses and witnesses sets). *A witness for $a \succ b$ is one of the following types: (i)* Subsets*: A pair of disjoint subsets $(S, S') \in \mathcal{S}_{a,b}$ such that*

$$P_{\{a\} \cup S, \{b\} \cup S'} > P_{\{b\} \cup S \succ \{a\} \cup S'}.$$

*We denote the set of all* subsets *witnesses for $a \succ b$ by $\mathcal{S}_{a,b}^*$.*
*(ii)* Subset-Team*: $(S, T) \in \mathcal{T}_{a,b}$, such that*

$$P_{\{a\} \cup S, T} > P_{\{b\} \cup S, T}.$$

*We denote the set of all* subset-team *witnesses for $a \succ b$ by $\mathcal{T}_{a,b}^*$.*

In case we find a witness, we can use it to infer over players relations as follows.

**Lemma A.2.** *If $\mathcal{S}_{a,b}^* \cup \mathcal{T}_{a,b}^* \neq \emptyset$, then $a \succ b$.*

*Proof.* First, consider the existence of $(S, S') \in \mathcal{S}_{a,b}^*$.
Hence

$$(*) \ P_{S \cup \{a\}, S' \cup \{b\}} > P_{S \cup \{b\}, S' \cup \{a\}}$$

Assume for contradiction that $b \succ a$. Consistency implies $S \cup \{b\} \succ S \cup \{a\}$ and $S' \cup \{b\} \succ S' \cup \{a\}$.

Adding up the two implications from the witness definition and SST, we have

$$P_{S \cup \{a\}, S' \cup \{b\}} >_{(*)} P_{S \cup \{b\}, S' \cup \{a\}} >_{b \succ a} P_{S \cup \{a\}, S' \cup \{a\}} >_{b \succ a} P_{S \cup \{a\}, S' \cup \{b\}},$$

Which is a contradiction.

Now, consider the existence of $(S, T) \in \mathcal{T}_{a,b}^*$. We have that

$$(**) \ P_{S \cup \{a\}, T} > P_{S \cup \{b\}, T}$$

Assume for contradiction that $b \succ a$. Consistency implies $S \cup \{b\} \succ S \cup \{a\}$.

$$P_{S \cup \{b\}, T} >_{b \succ a} P_{S \cup \{a\}, T} >_{(**)} P_{S \cup \{b\}, T},$$

Which is a contradiction. $\qquad\square$

Note that while the above lemma implies a sufficient condition for $a \succ b$, there is no guarantee that for every $a \succ b$ there exists a witness that proves it, in the form of disjoint subsets. For example, consider a lexicographical order among teams with $n = 4$, $k = 2$ (i.e., $\{1, 2\} \succ \{1, 3\} \succ \{1, 4\} \succ \{2, 3\} \succ \{2, 4\} \succ \{3, 4\}$) with uniform noise, e.g. when $P_{A,B} = 0.6$ for all teams $A \succ B$. It follows from consistency and $\{1, 2\} \succ \{2, 3\}$ that $1 \succ 3$, but there is no witness for that. Moreover, even if we execute each of the three feasible duels enough to estimate correctly that $P_{12,34} = P_{13,24} = P_{14,23} = 0.6$ there is no way to distinguish between the second and third best players. In what follows we formalize this intuition, showing that if a players relation is provable then one of the aforementioned witnesses types exists for it.

Next, we recall the Observable relation and the set $\mathcal{C}_{obs}$.

**Observable relation**   Let $\succ_{obs}$ denote the relation between every two disjoint teams, i.e.,

$$A \succ_{obs} B \iff A \succ B, \ |A| = |B| = k, \ A \cap B = \emptyset, \ A, B \subseteq [n].$$

Namely the relation $\succ_{obs}$ is deducible from valid duels [8].

In what follows, we elaborate more on the definition of $\mathcal{C}_{obs}$ by defining first a set for Compatible winning probabilities.

**Compatible winning probabilities**   Let $\mathbb{P}_{obs}$ be the set of all tuples $(P', \succ')$, where $P'$ are the winning probability matrices for teams, i.e., $P' = (P'_{A,B})_{|A|=|B|=k, A,B \in [n]} \in [0,1]^{\binom{n}{k}} \times [0,1]^{\binom{n}{k}}$, and $\succ'$ is a consistent total order on the teams such that:

1. For every pair of disjoint teams $(A, B)$ the winning probability matrix $P'$ has the same winning probability as the ground truth $P$, i.e., $A \cap B = \emptyset$ implies $P'_{A,B} = P_{A,B}$.

2. It holds that $P'_{A,B} = 1/2$ iff $A = B$.

3. $P'_{A,B} > 1/2$ if and only if $A \succ' B$.

4. $P'$ satisfies SST w.r.t. $\succ'$.

Namely, $\mathbb{P}_{obs}$ contains all tuples $(P', \succ')$ that are compatible to our assumptions and do not contradict the winning probabilities the learner can observe.

**Compatible relations**   Let $\mathcal{C}_{obs}$ be the set of all total orders $\succ'$ for which there exists $(P', \succ') \in \mathbb{P}_{obs}$. Notice that by the definition of $\mathbb{P}_{obs}$, we know that $\succ'$ satisfy consistency and in particular it holds that $A \succ' B$ for every disjoint teams $(A, B)$ with $A \succ_{obs} B$. Namely, $\mathcal{C}_{obs}$ is the sets of all possible total orders that could explain the results of the observable duels.

We remark that it follows directly from the definition of $\mathbb{P}_{obs}$ that $(P, \succ) \in \mathbb{P}_{obs}$, where $P$ is the ground truth winning probability matrix and $\succ$ the ground truth total order. Because of this, it also holds that $\succ$ is in $\mathcal{C}_{obs}$. To illustrate that $\succ$ is typically not the only total order in $\mathcal{C}_{obs}$, we provide the following example.

**Example A.3.** *For $n = 5, k = 2$, consider the lexicographic order , i.e., $\{1,2\} \succ \{1,3\} \succ \{1,4\} \succ \{1,5\} \succ \{2,3\} \succ \{2,4\} \succ \{2,5\} \succ \{3,4\} \succ \{3,5\} \succ \{4,5\}$ and assume $P_{A,B} = 0.6$ iff $A \succ B$ (equivalently $P_{A,B} = 0.4$ iff $B \succ A$). Then, we have that*

$$A \succ_{obs} B \iff \begin{cases} 1 \in A, \text{ or} \\ 1 \notin A \cup B, 2 \in A. \end{cases}$$

*While $\succ \in \mathcal{C}_{obs}$, there are other consistent total orders in $\mathcal{C}_{obs}$, such as $\{1,2\} \succ' \{1,5\} \succ' \{1,4\} \succ' \{1,3\} \succ' \{2,5\} \succ' \{2,4\} \succ' \{2,3\} \succ' \{5,4\} \succ' \{5,3\} \succ' \{4,3\}$ (the order $\succ'$ is obtained by swapping players 3 and 5 in $\succ$). Similarly, the probability matrices $P_{A,B} = 0.6$ for all $A \succ B$, (the ground truth), but $P^1_{A,B} = 0.7 \ \forall \ A \succ B$ and $P^2_{A,B} = 0.6 \ \forall \ A \succ' B$ are also in $\mathbb{P}$.*

We now recall the definition of the deducible relation, $\succ^*$ for both teams and single players, where the latter definition is a combination of the former and single players consistency.

The intuition behind these definitions is that a relation can be deducible (proven) by team duels if any "reasonable" total order that could possibly be the ground order agree on this relation. We stress that both $\mathbb{P}_{obs}$ and $\mathcal{C}_{obs}$ are strictly for analysis, as we do not need to explicitly calculate them.

In what follows, we formalize the notion of the *deducible* part of the ground truth $\succ$, i.e., the relations of $\succ$ that can be deduced by the learner, if it were given infinite access to duels.

**Definition A.4.** *Team $A$ is* deducibly better *than a different team $B$, denoted by $A \succ^* B$, if $A \succ' B$ for all $\succ' \in \mathcal{C}_{obs}$.*

**Definition A.5.** *Player $a$ is* deducibly better *than player $b$, denoted by $a \succ^* b$, if there exists a subset $S \subseteq [n] \setminus \{a,b\}, |S| = k - 1$ such that $\{a\} \cup S \succ' \{b\} \cup S$ for all $\succ' \in \mathcal{C}_{obs}$.*

---

[8]Notice that technically, $\succ_{obs}$ is not defined on pairs of different teams which are not disjoint, and therefore not even a partial order on teams (e.g., we have that $\{a,b\} \succ_{obs} \{c,d\} \succ_{obs} \{a,e\}$ but $\{a,b\} \not\succ_{obs} \{a,e\}$ as they share a player and the duel $(\{a,b\}, \{a,e\})$ is not observable.).

We continue with an example for relations that $\succ^*$ must satisfy. Suppose the learner has observed that $\{a,c\} \succ_{obs} \{b,d\} \succ_{obs} \{a,e\} \succ_{obs} \{c,d\}$. Since all the relations $\succ \in \mathcal{C}_{obs}$ satisfy transitivity, it follows that $\{b,d\} \succ^* \{c,d\}$, $\{a,c\} \succ^* \{a,e\}$, and $\{a,c\} \succ^* \{c,d\}$. As each $\succ \mathcal{C}_{obs}$ also satisfies single players consistency, we deduce $b \succ^* c$, $c \succ^* e$ and $a \succ^* d$, respectively. Applying single players consistency again, we can get, for example, $\{a,b\} \succ^* \{a,c\} \succ^* \{a,e\} \succ^* \{d,e\}$ (using $b \succ^* c$, $c \succ^* e$ and $a \succ^* d$, respectively). Intuitively, what we will show in Theorem 3.2 is that for every pair of players that one is provably better than the another there exists a witness for it, thus there is a short proof with which the learner can verify their relation with $O(1)$ queries in the deterministic case. Before we start proving the Theorem 3.2 we prove the following helpful lemma.

**Lemma A.6.** *Let $\succ \in \mathcal{C}_{obs}$ and $P$ be a corresponding probability matrix satisfying SST.*

*Let $a, b \in [n]$ with $a \succ b$. Then, the following holds true:*

1. *Let $(S, S') \in \mathcal{S}_{a,b}$, then $P_{\{a\}\cup S, \{b\}\cup S'} \geq P_{\{b\}\cup S, \{a\}\cup S'}$.*

2. *Let $(S, T) \in \mathcal{T}_{a,b}$, then $P_{\{a\}\cup S, T} \geq P_{\{b\}\cup S, T}$.*

*Proof.* 1. We start by proving that for every $(S, S') \in \mathcal{S}_{a,b}$ it holds that

$$P_{\{a\}\cup S, \{b\}\cup S'} \geq P_{\{b\}\cup S, \{a\}\cup S'}$$

by exhaustion.

(a) If $S \cup \{a\} \succ S' \cup \{b\}$ and $S' \cup \{a\} \succ S \cup \{b\}$ then it follows that $1/2 < P_{\{a\}\cup S, \{b\}\cup S'}, P_{\{a\}\cup S', \{b\}\cup S}$ and therefore

$$P_{\{a\}\cup S, \{b\}\cup S'} > 1/2 > 1 - P_{\{a\}\cup S', \{b\}\cup S} = P_{\{b\}\cup S, \{a\}\cup S'}.$$

(b) If (a) does not hold, then it follows that either of the following holds true:

(i) $\{b\}\cup S \succ \{a\}\cup S'$ (and $\{a\}\cup S \succ \{b\}\cup S'$ as $b \not\succ a$).
From single players consistency of $\succ$ we have that

$$\{a\}\cup S \succ \{b\}\cup S \succ \{a\}\cup S' \succ \{b\}\cup S'$$

Applying SST, we have that

$$P_{\{a\}\cup S, \{b\}\cup S'} \geq P_{\{a\}\cup S, \{a\}\cup S'} \geq P_{\{b\}\cup S, \{a\}\cup S'}.$$

(ii) $\{b\}\cup S' \succ \{a\}\cup S$ (and $\{a\}\cup S' \succ \{b\}\cup S$ as $b \not\succ a$).
From consistency, we have that

$$\{a\}\cup S' \succ \{b\}\cup S' \succ \{a\}\cup S \succ \{b\}\cup S$$

Applying SST, we have that

$$P_{\{a\}\cup S', \{b\}\cup S} \geq P_{\{a\}\cup S', \{a\}\cup S} \geq P_{\{b\}\cup S', \{a\}\cup S}.$$

Therefore

$$1 - P_{\{b\}\cup S', \{a\}\cup S} \geq 1 - P_{\{a\}\cup S', \{b\}\cup S}.$$

Applying $P_{A,B} = 1 - P_{B,A}$ for every $A, B \in [n]$,

$$P_{\{a\}\cup S, \{b\}\cup S'} \geq P_{\{b\}\cup S, \{a\}\cup S'}.$$

(iii) The case that $\{b\}\cup S' \succ \{a\}\cup S$ and $\{b\}\cup S' \succ \{a\}\cup S$ cannot hold as it would imply $b \succ a$ which is a contradiction to $a \succ b$, as $\succ$ being a consistent total order yields a total order on players.

2. Strict total order on teams together with consistency implies that either of the following holds: (a) $\{a\}\cup S \succ \{b\}\cup S \succ T$, (b)$\{a\}\cup S \succ T \succ \{b\}\cup S$, or (c) $T \succ \{a\}\cup S \succ \{b\}\cup S$. Applying SST on (a) and (c) proves the claim, and if (b) holds we have

$$P_{\{a\}\cup S, T} > 1/2 > P_{\{b\}\cup S, T}.$$

$\square$

We note that the left to right direction in the following sentence is very similar to Lemma A.2 and their proofs are equivalent, however for completeness we provide a full proof here as well.

**Theorem 3.2.** *Let* $a, b \in [n]$. *Then,* $a \succ^* b$ *if and only if* $\mathcal{S}^*_{a,b} \cup \mathcal{T}^*_{a,b} \neq \emptyset$.

*Proof.* We start with the direction from right to left, i.e., $\mathcal{S}^*_{a,b} \cup \mathcal{T}^*_{a,b} \neq \emptyset$ implies $a \succ^* b$.

First, consider $(S, S') \in \mathcal{S}^*_{a,b}$ and assume for contradiction that $a \succ^* b$ does not hold. That is, there exists $\succ' \in \mathcal{C}_{obs}$ and $P' \in \mathbb{P}_{obs}$ such that $b \succ' a$, and $P'$ is a corresponding winning probability matrix.

By Lemma A.6 and the definition of $\mathbb{P}_{obs}$ it follows that

$$P_{S\cup\{b\},S'\cup\{a\}} = P'_{S\cup\{b\},S'\cup\{a\}} \geq P'_{S\cup\{a\},S'\cup\{b\}} = P_{S\cup\{a\},S'\cup\{b\}}$$

holds, as the teams are disjoint. This is a contradiction to $(S, S') \in \mathcal{S}^*_{a,b}$.

Similarly, let $(S, T) \in \mathcal{T}^*_{a,b}$ and assume for contradiction that $a \succ^* b$ does not hold. That is, there exists $\succ' \in \mathcal{C}_{obs}$ and $P' \in \mathbb{P}_{obs}$ such that $b \succ' a$, and $P'$ is a corresponding winning probability matrix. By Lemma A.6 and the definition of $\mathbb{P}_{obs}$ it follows that

$$P_{S\cup\{b\},T} = P'_{S\cup\{b\},T} \geq P'_{S\cup\{a\},T} = P_{S\cup\{a\},T}$$

holds, as the teams are disjoint. This is a contradiction to $(S, T) \in \mathcal{T}^*_{a,b}$.

We turn to the direction from left to right, i.e. that $a \succ^* b$ yields $S^*_{a,b} \cup \mathcal{T}^*_{a,b} \neq \emptyset$. We start by defining $\mathcal{D}_a$ as the set of observable duels $(A, B)$ such that $a \in A$. Moreover, we define a permutation $\pi$ on the set of players, which simply exchanges the players $a$ and $b$ when present. More precisely,

$$\pi(S) = \begin{cases} S \setminus \{a\} \cup \{b\} & \text{if } a \in S, b \notin S \\ S \setminus \{b\} \cup \{a\} & \text{if } b \in S, a \notin S \\ S & \text{else.} \end{cases}$$

We claim that $a \succ^* b$ implies

$$P_{A,B} \geq P_{\pi(A),\pi(B)} \text{ for all } (A, B) \in \mathcal{D}_a \tag{2}$$

($P$ is the ground truth winning probability matrix). To see why, we first define

$$\mathcal{D}^1_a = \{(A, B) \in \mathcal{D}_a \mid b \in A\}$$
$$\mathcal{D}^2_a = \{(A, B) \in \mathcal{D}_a \mid b \in B\}$$
$$\mathcal{D}^3_a = \{(A, B) \in \mathcal{D}_a \mid b \notin A \cup B\}.$$

Notice that

$$\mathcal{D}_a = \mathcal{D}^1_a \cup \mathcal{D}^2_a \cup \mathcal{D}^3_a \tag{3}$$

When $(A, B) \in \mathcal{D}^1_a$, then $(\pi(A), \pi(B)) = (A, B)$ and $P_{A,B} = P_{\pi(A),\pi(B)}$.

When $(A, B) \in \mathcal{D}^2_a$, then $(A \setminus \{a\}, B \setminus \{b\}) \in \mathcal{S}_{a,b}$, and $P_{A,B} \geq P_{A\setminus\{a\}\cup\{b\},B\setminus\{b\}\cup\{a\}} = P_{\pi(A),\pi(B)}$ follows from Lemma A.6.

Similarly, when $(A, B) \in \mathcal{D}^3_a$ then $(A \setminus \{a\}, B) \in \mathcal{T}_{a,b}$ and $P_{A,B} \geq P_{A\setminus\{a\}\cup\{b\},B} = P_{\pi(A),\pi(B)}$ follows from Lemma A.6.

We will now show that $a \succ^* b$ implies the existence of $(A, B) \in \mathcal{D}_a$ with $P_{A,B} > P_{\pi(A),\pi(B)}$.

Assume not. Then in particular from (2) we have that $P_{A,B} = P_{\pi(A),\pi(B)}$ holds for all $(A, B) \in \mathcal{D}_a$.

**Claim.** *Let* $\succ'$ *be the relation defined by* $A \succ' B$ *iff* $\pi(A) \succ \pi(B)$ *with the corresponding winning probabilities defined by* $P'_{A,B} = P_{\pi(A),\pi(B)}$. *If* $P_{A,B} = P_{\pi(A),\pi(B)}$ *for every* $(A, B) \in \mathcal{D}_a$ *then* $P' \in \mathbb{P}_{obs}$ *and thus* $\succ' \in \mathcal{C}_{obs}$.

*Proof.* Observe that $P_{A,B} = P'_{A,B}$ for all disjoint teams $A$ and $B$ follows by definition. In addition, since $\pi$ is invertible and involuntary, for every team $A$ there exists a team $A_\pi$ such that $\pi(A_\pi) = A$ hence $P_{A,A} = P_{A_\pi,A_\pi} = 1/2$. It remains to show that (1) Every pair of different teams $A, B$ holds $P'_{A,B} > 1/2$ iff $A \succ' B$, (2) that $\succ'$ is a total ordering satisfying single players consistency, and (3) that $P'$ satisfy SST w.r.t. $\succ'$.

(1) Let $A, B$ be two different teams. It follows by the assumption over $P$ that $P'_{A,B} = P_{\pi(A),\pi(B)} > 1/2$, iff $\pi(A) \succ \pi(B)$, which holds iff $A \succ' B$ by definition.

(2) We now show that $\succ'$ is a strict total order. From it's definition we have that $\succ'$ is irreflexive. We also have that $\succ'$ is connected (and therefore strict) as $\pi$ is invertible and involutory, and every pair of different teams $A, B$ holds either $A \succ' B$ (if $\pi^{-1}(A) = \pi(A) \succ \pi(B) = \pi^{-1}(B)$) or $B \succ' A$ (if $\pi^{-1}(B) = \pi(B) \succ \pi(A) = \pi^{-1}(A)$), but not both. For transitivity, Consider a triplet of different teams, $A, B, C$ such that $A \succ' B \succ' C$ (and therefore $\pi^{-1}(A) = \pi(A) \succ \pi^{-1}(B) = \pi(B) \succ \pi^{-1}(C) = \pi(C)$). From transitivity of $\succ$, we get $\pi^{-1}(A) = \pi(A) \succ \pi(C) = \pi^{-1}(C)$ which implies $A \succ' C$. Hence the relation $\succ'$ is a strict total order.

We continue by showing that $\succ'$ satisfies single players consistency.
Let $x, y \in [n]$ be a pair of players and $S \in [n] \setminus \{x, y\}$ be a set of players such that $x \cup S \succ' y \cup S$. We will show that $\{x\} \cup S' \succ' \{y\} \cup S'$ for all $S' \in [n] \setminus \{x, y\}$.

Since $\pi$ is invertible, we know that there exist players $x_\pi = \pi(x)$ and $y_\pi = \pi(y)$, and a set, $S_\pi = \pi(S) \in [n] \setminus \{x_\pi, y_\pi\}$, such that

$$\{x\} \cup S = \pi^{-1}(\{x_\pi\} \cup S_\pi)$$

and

$$\{y\} \cup S = \pi^{-1}(\{y_\pi\} \cup S_\pi).$$

From the definition of $\succ'$, we get

$$\{x_\pi\} \cup S_\pi \succ \{y_\pi\} \cup S_\pi.$$

Therefore from the consistency of $\succ$ every $S'_\pi \in [n] \setminus \{x_\pi, y_\pi\}$ holds $\{x_\pi\} \cup S'_\pi \succ \{y_\pi\} \cup S'_\pi$ hence by definition $\{x\} \cup S' \succ' \{y\} \cup S'$.

(3) We now show that $P'$ satisfy SST w.r.t. $\succ'$. Let $A \succ' B \succ' C$. From the definition of $\succ'$ we have that $\pi^{-1}(A) = \pi(A) \succ \pi^{-1}(B) = \pi(B) \succ \pi^{-1}(C) = \pi(C)$. As $P$ satisfy SST w.r.t. $\succ$,

$$P_{\pi(A),\pi(C)} \geq \max\{P_{\pi(A),\pi(B)}, P_{\pi(B),\pi(C)}\}$$

Once again from the definition of $\succ'$,

$$P'_{A,C} \geq \max\{P'_{A,B}, P'_{B,C}\},$$

Which means that $P'$ satisfy SST w.r.t. $\succ'$ by definition. □

Now, observe that, together with the above claim, $a \succ^* b$ imply that for any $S \subseteq [n] \setminus \{a, b\}$ of size $k - 1$ it holds that $S \cup \{a\} \succ^* S \cup \{b\}$ which implies ($i$) $S \cup \{a\} \succ S \cup \{b\}$ as well as ($ii$) $S \cup \{a\} \succ' S \cup \{b\}$, as both $\succ$ and $\succ'$ are in $\mathcal{C}_{obs}$. Applying the definitions of $\succ'$ and $\pi$, statement ($ii$) implies $\pi^{-1}(S \cup \{a\}) = \pi(S \cup \{a\}) \succ \pi(S \cup \{b\}) = \pi^{-1}(S \cup \{b\})$ which is equivalent to $S \cup \{b\} \succ S \cup \{a\}$ and hence yields a contradiction to ($i$).

We therefore deduce the existence of $(A, B) \in \mathcal{D}_a$ such that $P_{A,B} > P_{\pi(A),\pi(B)}$. From (3), either $(A, B) \in \mathcal{D}_a^2$, thus $(A \setminus \{a\}, B \setminus \{b\}) \in \mathcal{S}_{a,b}$, and $P_{A,B} > P_{A \setminus \{a\} \cup \{b\}, B \setminus \{b\} \cup \{a\}} = P_{\pi(A),\pi(B)}$ yields $(A \setminus \{a\}, B \setminus \{b\}) \in \mathcal{S}_{a,b}^*$, or $(A, B) \in \mathcal{D}_a^3$, thus $(A \setminus \{a\}, B) \in \mathcal{T}_{a,b}$ and $P_{A,B} > P_{A \setminus \{a\} \cup \{b\}, B} = P_{\pi(A),\pi(B)}$ implies $(A \setminus \{a\}, B) \in \mathcal{T}_{a,b}^*$ (As $(A, B) \in \mathcal{D}_a^1$, implies $P_{\pi(A),\pi(B)} = P_{A,B} > P_{A,B}$ which is a contradiction.). Overall, $\mathcal{S}_{a,b}^* \cup \mathcal{T}_{a,b}^* \neq \emptyset$. □

# B   Extended Version and Proofs of Section 4

We start by splitting the definition of $X_{a,b}(S, S', T)$ into two random variables, according to the two types of witnesses we introduced in the previous section. This will simplify the proof of Lemma 4.2.

For $(S, S') \in \mathcal{S}_{a,b}$ we introduce a random variable $Z_{a,b}(S, S')$ that combines the outcomes of the two duels obtained from the potential subsets witness $(S, S')$, namely $(S \cup \{a\}, S' \cup \{b\})$ and $(S' \cup \{a\}, S \cup \{b\})$ and similarly, a random variable $Y_{a,b}(S, T)$ that combines the outcomes of the two duels obtained by subset-team witness, $(S \cup \{a\}, T)$ and $(T, S \cup \{b\})$.

**Definition B.1.** *For $a, b \in [n], a \neq b$, $(S, S') \in \mathcal{S}_{a,b}$ and $(S, T) \in \mathcal{T}_{a,b}$,*

$$Z_{a,b}(S, S') = \frac{\mathbb{1}[(S \cup \{a\} > S' \cup \{b\})]}{2} + \frac{\mathbb{1}[(S' \cup \{a\} > S \cup \{b\})]}{2},$$

$$Y_{a,b}(S, T) = \frac{\mathbb{1}[\{a\} \cup S > T]}{2} + \frac{\mathbb{1}[T > \{b\} \cup S]}{2}.$$

We note that both $Z_{a,b}(S, S')$ and $Y_{a,b}(S, T)$ can take values in $\{0, 1/2, 1\}$.

The random variables $Z_{a,b}$ and $Y_{a,b}$ are the outcomes of picking random pairs, $(S, S') \in \mathcal{T}_{a,b}$ or $(S, T) \in \mathcal{S}_{a,b}$ and returning $Z_{a,b}(S, S')$ and $Y_{a,b}(S, T)$, respectively. Observe that

$$\mathbb{E}[Z_{a,b}] = \sum_{(S,S') \in \mathcal{S}_{a,b}} \frac{\mathbb{E}[Z_{a,b}(S, S')]}{|\mathcal{S}_{a,b}|} = \sum_{(S,S') \in \mathcal{S}_{a,b}} \frac{P_{\{a\} \cup S, \{b\} \cup S'} + P_{\{a\} \cup S, \{b\} \cup S'}}{2|\mathcal{S}_{a,b}|},$$

$$\mathbb{E}[Y_{a,b}] = \sum_{(S,T) \in \mathcal{T}_{a,b}} \frac{\mathbb{E}[Y_{a,b}(S, T)]}{|\mathcal{T}_{a,b}|} = \sum_{(S,T) \in \mathcal{T}_{a,b}} \frac{P_{\{a\} \cup S, T} + P_{T, \{b\} \cup S'}}{2|\mathcal{T}_{a,b}|},$$

Where the expectation $\mathbb{E}[Z_{a,b}]$ is taken over all elements of $\mathcal{S}_{a,b}$ and the expectation $\mathbb{E}[Y_{a,b}]$ is taken over all elements $\mathcal{T}_{a,b}$.

The following lemma apply for every $a \succ b$, even if $a \not\succ^* b$. We prove Lemma using SST and consistency.

**Lemma B.2.** *Let $a, b \in [n]$ be any two players such that $a \succ b$. Then,*

*(1) For every $(S, S') \in \mathcal{S}_{a,b}$ it holds that $\mathbb{E}[Z_{a,b}(S, S')] \geq 1/2$.*

*(2) For every $(S, T) \in \mathcal{T}_{a,b}$ it holds that $\mathbb{E}[Y_{a,b}(S, T)] \geq 1/2$.*

*Proof.* (1) Let $(S, S') \in \mathcal{S}_{a,b}$ and $a \succ b$. Then,

$$\mathbb{E}[Z_{a,b}(S, S')] = \frac{P_{\{a\} \cup S, \{b\} \cup S'} + P_{\{a\} \cup S', \{b\} \cup S}}{2} \geq \frac{1}{2}$$

$$\iff P_{\{a\} \cup S, \{b\} \cup S'} + P_{\{a\} \cup S', \{b\} \cup S} \geq 1$$

$$\iff P_{\{a\} \cup S, \{b\} \cup S'} \geq 1 - P_{\{a\} \cup S', \{b\} \cup S}$$

$$\iff P_{\{a\} \cup S, \{b\} \cup S'} \geq P_{\{b\} \cup S, \{a\} \cup S'},$$

which holds according to Lemma A.6.

(2) Let $(S, T) \in \mathcal{T}_{a,b}$ and $a \succ b$. From Lemma A.6 we have that

$$P_{\{a\} \cup S, T} \geq P_{\{b\} \cup S, T},$$

which is equivalent to

$$P_{\{a\} \cup S, T} \geq P_{\{b\} \cup S, T} = 1 - P_{T, \{b\} \cup S}$$

and therefore

$$2\mathbb{E}[Y_{a,b}(S, T)] \geq 1.$$

Hence, $\mathbb{E}[Y_{a,b}(S, T)] \geq 1/2$. $\qquad\square$

**Corollary B.3.** *For players $a, b \in [n]$ such that $a \succ b$ then $\mathbb{E}[Z_{a,b}], \mathbb{E}[Y_{a,b}] \geq 1/2$.*

For the definition of $X_{a,b}(S, S', T)$ we refer to the main part of our paper.

In similar fashion to the definitions of $\mathcal{S}_{a,b}$, $\mathcal{S}_{a,b}^*$ and $Z_{a,b}$ w.r.t. $Z(S, S')$, we defined

$$\mathcal{X}_{a,b} = \{(S, S', T) | (S, S') \in \mathcal{S}_{a,b}, (S, T) \in \mathcal{T}_{a,b}\},$$

and the random variable $X_{a,b}$ to be the outcome of picking a random triplet, $(S, S', T) \in \mathcal{X}_{a,b}$ and returning $X_{a,b}(S, S', T)$.

The set $\mathcal{X}_{a,b}^*$ contains all triplets $(S, S', T) \in \mathcal{X}_{a,b}$ such that either $(S, S') \in \mathcal{S}_{a,b}^*$ or $(S, T) \in \mathcal{T}_{a,b}^*$. Note that the support of each $X_{a,b}(S, S', T)$ is included $\{-1/2, -1/4, 0, 1/4, 1/2\}$ and that $\mathbb{E}[X_{a,b}] \in [-1/2, 1/2]$.

In the following we show how $X_{a,b}(S, S', T)$ can be expressed by $Z_{a,b}(S, S')$ and $Y_{a,b}(S, T)$, namely

**Observation B.4.** *For every two players $a, b \in [n]$, we have that $\mathbb{E}[X_{a,b}] = \frac{1}{2}(\mathbb{E}[Z_{a,b}] + \mathbb{E}[Y_{a,b}]) - \frac{1}{2}$.*

*Proof.*

$$
\begin{aligned}
X_{a,b}(S, S', T) &= \frac{\mathbb{1}[S \cup \{a\} > S' \cup \{b\}] - \mathbb{1}[S \cup \{b\} > S' \cup \{a\}]}{4} \\
&+ \frac{\mathbb{1}[S \cup \{a\} > T] - \mathbb{1}[S \cup \{b\} > T]}{4} \\
&= \frac{\mathbb{1}[S \cup \{a\} > S' \cup \{b\}] + \mathbb{1}[S' \cup \{a\} > S \cup \{b\}] - 1}{4} \\
&+ \frac{\mathbb{1}[S \cup \{a\} > T] + \mathbb{1}[T > S \cup \{b\}] - 1}{4} \\
&= \frac{Z_{a,b}(S, S') + Y_{a,b}(S, T) - 1}{2}.
\end{aligned}
$$

As a direct result of the linearity of expectation, we have that

$$
\begin{aligned}
\mathbb{E}_{(S,S',T) \sim \mathcal{X}_{a,b}}[X_{a,b}] &= \frac{1}{|\mathcal{X}_{a,b}|} \sum_{(S,S',T) \in \mathcal{X}_{a,b}} \mathbb{E}[X_{a,b}(S, S', T)] \\
&= \frac{1}{|\mathcal{X}_{a,b}|} \sum_{(S,S',T) \in \mathcal{X}_{a,b}} \frac{\mathbb{E}[Z_{a,b}(S, S')]}{2} + \frac{1}{|\mathcal{X}_{a,b}|} \sum_{(S,S',T) \in \mathcal{X}_{a,b}} \frac{\mathbb{E}[Y_{a,b}(S, T)]}{2} - \frac{1}{2} \\
&= \frac{1}{|\mathcal{X}_{a,b}|} \sum_{(S,S') \in \mathcal{S}_{a,b}} \binom{n-k-1}{k} \frac{\mathbb{E}[Z_{a,b}(S, S')]}{2} \\
&+ \frac{1}{|\mathcal{X}_{a,b}|} \sum_{(S,T) \in \mathcal{T}_{a,b}} \binom{n-k-1}{k-1} \frac{\mathbb{E}[Y_{a,b}(S, T)]}{2} - \frac{1}{2} \\
&= \frac{1}{|\mathcal{S}_{a,b}|} \sum_{(S,S') \in \mathcal{S}_{a,b}} \frac{\mathbb{E}[Z_{a,b}(S, S')]}{2} + \frac{1}{|\mathcal{T}_{a,b}|} \sum_{(S,T) \in \mathcal{T}_{a,b}} \frac{\mathbb{E}[Y_{a,b}(S, T)]}{2} - \frac{1}{2} \\
&= \frac{\mathbb{E}_{(S,S') \sim \mathcal{S}_{a,b}}[Z_{a,b}]}{2} + \frac{\mathbb{E}_{(S,T) \sim \mathcal{T}_{a,b}}[Y_{a,b}]}{2} - \frac{1}{2}.
\end{aligned}
$$

$\square$

For the next Theorem's proof we rely on Theorem 3.2, Corollary B.3 in one direction, and show the other using the probabilistic method.

**Theorem 4.1.** *For every two players $a, b \in [n]$ it holds that $a \succ^* b$ if and only if $\mathbb{E}[X_{a,b}] > 0$.*

*Proof.* We will show that for players $a, b \in [n]$ it holds that $a \succ^* b$ iff one of the following holds:
(1) $\mathbb{E}[Z_{a,b}] > 1/2$, or
(2) $\mathbb{E}[Y_{a,b}] > 1/2$.
This is equivalent to $\mathbb{E}[X_{a,b}] > 0$ according to the definition of $X_{a,b}$ and Corollary B.3.
($\Rightarrow$) If $a \succ^* b$ then from Theorem 3.2 we know that one of the following holds:

1. There exists a subsets, witness $(S, S') \in \mathcal{S}_{a,b}$ for $a \succ b$. So by definition $\mathbb{E}[Z_{a,b}(S, S')] > 1/2$, and combined with Lemma B.2 we have $\mathbb{E}[Z_{a,b}] > 1/2$.

2. There exists a subset-team witness $(S, T) \in \mathcal{T}_{a,b}$ for $a \succ b$. Thus $\mathbb{E}[Y_{a,b}(S, T)] > 1/2$, hence Lemma B.2 implies that $\mathbb{E}[Y_{a,b}] > 1/2$.

($\Leftarrow$) If (1) holds, the probabilistic method implies the existence of $(S, S') \in \mathcal{S}_{a,b}$ such that $\mathbb{E}[Z_{a,b}(S, S')] > 1/2$ which means that $(S, S')$ is a witness for $a \succ b$, hence, $a \succ^* b$ by Theorem 3.2. If (2) holds, the probabilistic method implies that there exists $(S, T) \in \mathcal{T}_{a,b}$ such that $\mathbb{E}[Y_{a,b}(S, T)] > 1/2$ which means that $(S, T)$ is a witness for $a \succ b$, hence, $a \succ^* b$ by Theorem 3.2.

Thus according to the definition of $X_{a,b}$ the theorem holds. $\qquad\square$

**Gap parameter** Recall that we defined our gap parameter by $\Delta = \mathbb{E}[X_{k,k+1}]$. In the following we show that our gap parameter does not just help us to distinguish between the top $k$ and the top $k + 1$ players, but also between other players in $A_k^*$ and players from $[n] \setminus A_k^*$. To this end, we show in Lemma 4.2 that strong stochastic transitivity holds for $\mathbb{E}[X_{a,b}]$. For most elements $(S, S', T) \in \mathcal{X}_{a,b}$ it holds that $\mathbb{E}[X_{a,c}(\pi(S), \pi(S'), \pi(T))] \geq \mathbb{E}[X_{a,b}(S, S', T)]$ (and analogously for $X_{b,c}$), where $\pi$ is a permutation exchanging $b$ and $c$, but, surprisingly, this is not true in general. By constructing a charging scheme, we can still show that this holds in expectation over all elements of $\mathcal{X}_{a,b}$, and derive a strong stochastic transitivity for distinguishabilities w.r.t. the total order $\succ$ on the players.

The proof of the following lemma also shows that from every $a \succ b$ witness $(S, S', T) \in \mathcal{X}_{a,b}^*$, and for any player $c$ such that $b \succ c$ we can create a $a \succ c$- witness. Similarly, from every $b \succ c$ witness $(S, S', T) \in \mathcal{X}_{b,c}^*$, and for any player $a$ such that $a \succ b$ we can create a $a \succ c$- witness.

**Lemma 4.2.** *For a triplet of players $a \succ b \succ c$ it holds that*

$$\mathbb{E}[X_{a,c}] \geq \max\{\mathbb{E}[X_{a,b}], \mathbb{E}[X_{b,c}]\}.$$

*Proof.* In the following we show that $\mathbb{E}[X_{a,c}] \geq \mathbb{E}[X_{a,b}]$. The proof that $\mathbb{E}[X_{a,c}] \geq \mathbb{E}[X_{b,c}]$ works completely analogously and is therefore omitted. Let $\pi$ be the function exchanging $b$ and $c$, i.e.

$$\pi(S) = \begin{cases} S \setminus \{c\} \cup \{b\} & \text{if } c \in S, b \notin S \\ S \setminus \{b\} \cup \{c\} & \text{if } b \in S, c \notin S \\ S & \text{else.} \end{cases}$$

Then, we define the function $f : \mathcal{X}_{a,b} \to \mathcal{X}_{a,c}$ by $f(S, S', T) = (\pi(S), \pi(S'), \pi(T))$. Observe that, for this application of $\pi$, the second case within the definition of $\pi$ never occurs, as none of the sets $S, S', T$ contains $b$ when $(S, S', T) \in \mathcal{X}_{a,b}$. It will we helpful to partition $\mathcal{X}_{a,b}$ in the following way.

$$\begin{aligned} \mathcal{X}_{a,b}^1 &= \{(S, S', T) \in \mathcal{X}_{a,b} \mid c \notin S \cup S' \cup T\} \\ \mathcal{X}_{a,b}^2 &= \{(S, S', T) \in \mathcal{X}_{a,b} \mid c \in S\} \\ \mathcal{X}_{a,b}^3 &= \{(S, S', T) \in \mathcal{X}_{a,b} \mid c \in S' \setminus T\} \\ \mathcal{X}_{a,b}^4 &= \{(S, S', T) \in \mathcal{X}_{a,b} \mid c \in T \setminus S'\} \\ \mathcal{X}_{a,b}^5 &= \{(S, S', T) \in \mathcal{X}_{a,b} \mid c \in T \cap S'\}. \end{aligned}$$

Then we can also define $\mathcal{X}_{a,c}^i = \{f(S, S', T) \mid (S, S', T) \in \mathcal{X}_{a,b}^i\}$ for all $i \in \{1, \ldots, 5\}$. Observe that $\{\mathcal{X}_{a,c}^i \mid i \in \{1, \ldots, 5\}\}$ is also a partition of $\mathcal{X}_{a,c}$.

We will start by proving that for every $(S, S', T) \in \mathcal{X}_{a,b}^1 \cup \mathcal{X}_{a,b}^2 \cup \mathcal{X}_{a,b}^3 \cup \mathcal{X}_{a,b}^4 \cup \mathcal{X}_{a,b}^5$

$$\mathbb{E}[Z_{a,c}(f(S, S'))] \geq \mathbb{E}[Z_{a,b}(S, S')] \tag{4}$$

and for all $(S, S', T) \in \mathcal{X}_{a,b}^1 \cup \mathcal{X}_{a,b}^2 \cup \mathcal{X}_{a,b}^3$

$$\mathbb{E}[Y_{a,c}(f(S, T))] \geq \mathbb{E}[Y_{a,b}(S, T)] \tag{5}$$

by exhaustion.

(i) Let $(S, S', T) \in \mathcal{X}_{a,b}^1$. We get that $f(S, S', T) = (S, S', T)$ and both

$$\begin{aligned} \mathbb{E}[Z_{a,c}(S, S')] &= \frac{P_{\{a\} \cup S, \{c\} \cup S'} + P_{\{a\} \cup S', \{c\} \cup S}}{2} \\ &\geq \frac{P_{\{a\} \cup S, \{b\} \cup S'} + P_{\{a\} \cup S', \{b\} \cup S}}{2} = \mathbb{E}[Z_{a,b}(S, S')], \end{aligned}$$

$$\mathbb{E}[Y_{a,c}(S,T)] = \frac{P_{\{a\}\cup S,T} + P_{T,\{c\}\cup S}}{2}$$

$$\geq \frac{P_{\{a\}\cup S,T} + P_{T,\{b\}\cup S}}{2} = \mathbb{E}[Y_{a,b}(S,T)]$$

follow from consistency and SST.

(ii) Let $(S, S', T) \in \mathcal{X}_{a,b}^2$. Then, $f(S, S', T) = (S \setminus \{c\} \cup \{b\}, S, T)$ and both

$$\mathbb{E}[Z_{a,c}(S \setminus \{c\} \cup \{b\}, S')] = \frac{P_{\{a\}\cup S\setminus\{c\}\cup\{b\},\{c\}\cup S'} + P_{\{a\}\cup S',\{c\}\cup S\setminus\{c\}\cup\{b\}}}{2}$$

$$\geq \frac{P_{\{a\}\cup S,\{b\}\cup S'} + P_{\{a\}\cup S',\{b\}\cup S}}{2}$$

$$= \mathbb{E}[Z_{a,b}(S,S')]$$

$$\mathbb{E}[Y_{a,c}(S \setminus \{c\} \cup \{b\}, T)]) = \frac{P_{\{a\}\cup S\setminus\{c\}\cup\{b\},T} + P_{T,\{c\}\cup S}}{2}$$

$$\geq \frac{P_{\{a\}\cup S,T} + P_{T,\{b\}\cup S}}{2} = \mathbb{E}[Y_{a,b}(S,T)]$$

follow as $\{c\} \cup S \setminus \{c\} \cup \{b\} = S \cup \{b\}$ and from consistency and SST yield the rest.

(iii) Let $(S, S', T) \in \mathcal{X}_{a,b}^3$. Then, $f(S, S', T) = (S, S' \setminus \{c\} \cup \{b\}, T)$ and

$$\mathbb{E}[Z_{a,c}(S, S' \setminus \{c\} \cup \{b\}))] = \frac{P_{\{a\}\cup S,\{c\}\cup S'\setminus\{c\}\cup\{b\}} + P_{\{a\}\cup S'\setminus\{c\}\cup\{b\},\{c\}\cup S}}{2}$$

$$\geq \frac{P_{\{a\}\cup S,\{b\}\cup S'} + P_{\{a\}\cup S',\{b\}\cup S}}{2} = \mathbb{E}[Z_{a,b}(S,S')]$$

follows as $\{c\} \cup S' \setminus \{c\} \cup \{b\} = S' \cup \{b\}$ and consistency and SST yield the rest. In addition, we already showed that in this case thus $\mathbb{E}[Y_{a,c}(S,T)] \geq \mathbb{E}[Y_{a,b}(S,T)]$ (due to the same reason as in (i)).

(iv) Let $(S, S', T) \in \mathcal{X}_{a,b}^4$. Then, $f(S, S', T) = (S, S', T \setminus \{c\} \cup \{b\})$. Observe that we have already shown that $\mathbb{E}[Z_{a,c}(S,S')] \geq \mathbb{E}[Z_{a,b}(S,S')]$ in this case (due to the same reason as (i)).

(v) Let $(S, S', T) \in \mathcal{X}_{a,b}^5$. Then, $f(S, S', T) = (S, S' \setminus \{c\} \cup \{b\}, T \setminus \{c\} \cup \{b\})$. Observe that we have already shown that $\mathbb{E}[Z_{a,c}(S, S' \setminus \{c\} \cup \{b\})] \geq \mathbb{E}[Z_{a,b}(S,S')]$ in this case (due to the same reason as (iii)).

This concludes the proof of equations (4) and (5). In particular, from (ii) and (iii) it directly follows that

$$\sum_{(S,S',T)\in\mathcal{X}_{a,c}^i} \mathbb{E}[X(S,S',T)] = \sum_{(S,S',T)\in\mathcal{X}_{a,b}^i} \mathbb{E}[X(f(S,S',T))] \geq \sum_{(S,S',T)\in\mathcal{X}_{a,b}^i} \mathbb{E}[X(S,S',T)]$$

(6)

holds for $i \in \{2, 3\}$.

We will continue the proof by showing that, for every $(S, T) \in \mathcal{S}_{a,b}$ with $c \in T$, it holds that

$$\mathbb{E}[Z_{a,c}(S, T \setminus \{c\})] + \mathbb{E}[Y_{a,c}(S, T \setminus \{c\} \cup \{b\})] \geq \mathbb{E}[Z_{a,b}(S, T \setminus \{c\})] + \mathbb{E}[Y_{a,b}(S,T)]. \quad (7)$$

This will then be helpful to conclude the proof.

To this end, observe that

$$\mathbb{E}[Z_{a,c}(S, T \setminus \{c\})] + \mathbb{E}[Y_{a,c}(S, T \setminus \{c\} \cup \{b\})])$$

$$= P_{S\cup\{a\},T} + P_{T\setminus\{c\}\cup\{a\},S\cup\{c\}} + P_{S\cup\{a\},T\setminus\{c\}\cup\{b\}} + P_{T\setminus\{c\}\cup\{b\},S\cup\{c\}}$$

$$= P_{S\cup\{a\},T\setminus\{c\}\cup\{b\}} + P_{T\setminus\{c\}\cup\{a\},S\cup\{c\}} + P_{S\cup\{a\},T} + P_{T\setminus\{c\}\cup\{b\},S\cup\{c\}}$$

$$\geq P_{S\cup\{a\},T\setminus\{c\}\cup\{b\}} + P_{T\setminus\{c\}\cup\{a\},S\cup\{b\}} + P_{S\cup\{a\},T} + P_{T,S\cup\{b\}}$$

$$= \mathbb{E}[Z_{a,b}(S, T \setminus \{c\})] + \mathbb{E}[Y_{a,b}(S, T)],$$

which follows by consistency and SST. This will now be helpful to establish a charging scheme. Namely, we are first going to show that

$$\sum_{(S,S',T)\in\mathcal{X}_{a,c}^4} \mathbb{E}[X_{a,c}(S, S', T)] = \sum_{(S,S',T)\in\mathcal{X}_{a,b}^4} \mathbb{E}[X_{a,c}(f(S, S', T))] \geq \sum_{(S,S',T)\in\mathcal{X}_{a,b}^4} \mathbb{E}[X_{a,b}(S, S', T)].$$

(8)

This is true since

$$\sum_{(S,S',T)\in\mathcal{X}_{a,c}^4} 2\mathbb{E}[X_{a,c}(S, S', T)] + |\mathcal{X}_{a,c}|$$

$$\sum_{(S,S',T)\in\mathcal{X}_{a,b}^4} 2\mathbb{E}[X_{a,c}(S, S', T \setminus \{c\} \cup \{b\})] + |\mathcal{X}_{a,c}|$$

$$= \sum_{(S,S',T)\in\mathcal{X}_{a,b}^4} \left(\mathbb{E}[Z_{a,c}(S, S')] + \mathbb{E}[Y_{a,c}(S, T \setminus \{c\} \cup \{b\})]\right)$$

$$= \binom{n-k-2}{k-1}\left(\sum_{(S,S')\in\mathcal{S}_{a,b}\cap\mathcal{S}_{a,c}} \mathbb{E}[Z_{a,c}(S, S')] + \sum_{(S,T)\in\mathcal{T}_{a,b}|c\in T} \mathbb{E}[Y_{a,c}(S, T \setminus \{c\} \cup \{b\})]\right)$$

$$= \binom{n-k-2}{k-1}\left(\sum_{(S,S')\in\mathcal{S}_{a,b}\cap\mathcal{S}_{a,c}} \mathbb{E}[Z_{a,c}(S, S')] + \sum_{(S,S')\in\mathcal{S}_{a,b}\cap\mathcal{S}_{a,c}} \mathbb{E}[Y_{a,c}(S, S' \cup \{b\})]\right)$$

$$= \binom{n-k-2}{k-1}\left(\sum_{(S,S')\in\mathcal{S}_{a,b}\cap\mathcal{S}_{a,c}} \mathbb{E}[Z_{a,c}(S, S')] + \mathbb{E}[Y_{a,c}(S, S' \cup \{b\})]\right)$$

$$\geq \binom{n-k-2}{k-1}\left(\sum_{(S,S')\in\mathcal{S}_{a,b}\cap\mathcal{S}_{a,c}} \mathbb{E}[Z_{a,b}(S, S')] + \mathbb{E}[Y_{a,b}(S, S' \cup \{c\})]\right)$$

$$= \binom{n-k-2}{k-1}\left(\sum_{(S,S')\in\mathcal{S}_{a,b}\cap\mathcal{S}_{a,c}} \mathbb{E}[Z_{a,b}(S, S')] + \sum_{(S,S')\in\mathcal{S}_{a,b}\cap\mathcal{S}_{a,c}} \mathbb{E}[Y_{a,b}(S, S' \cup \{c\})]\right)$$

$$= \binom{n-k-2}{k-1}\left(\sum_{(S,S')\in\mathcal{S}_{a,b}\cap\mathcal{S}_{a,c}} \mathbb{E}[Z_{a,b}(S, S')] + \sum_{(S,T)\in\mathcal{T}_{a,b}|c\in T} \mathbb{E}[Y_{a,b}(S, T)]\right)$$

$$= \sum_{(S,S',T)\in\mathcal{X}_{a,b}^4} \left(\mathbb{E}[Z_{a,b}(S, S')] + \mathbb{E}[Y_{a,b}(S, T)]\right)$$

$$\sum_{(S,S',T)\in\mathcal{X}_{a,b}^4} 2\mathbb{E}[X_{a,b}(S, S', T)] + |\mathcal{X}_{a,b}|,$$

where the inequality follows by equation (7). This completes the proof of (8).

Next, we are going to show that a similar bound holds when we sum over elements in $\mathcal{X}_{a,b}^1 \cup \mathcal{X}_{a,b}^5$. More precisely, we are going to show that

$$\sum_{(S,S',T)\in\mathcal{X}_{a,c}^1\cup\mathcal{X}_{a,c}^5} \mathbb{E}[X_{a,c}(S, S', T)] = \sum_{(S,S',T)\in\mathcal{X}_{a,b}^1\cup\mathcal{X}_{a,b}^5} \mathbb{E}[X_{a,c}(f(S, S', T))]$$

$$\geq \sum_{(S,S',T)\in\mathcal{X}_{a,b}^1\cup\mathcal{X}_{a,b}^5} \mathbb{E}[X_{a,b}(S, S', T)].$$

(9)

To this end, observe that

$$\sum_{(S,S',T)\in\mathcal{X}_{a,c}^1} 2\mathbb{E}[X_{a,c}(S, S', T)] + \sum_{(S,S',T)\in\mathcal{X}_{a,c}^5} 2\mathbb{E}[X_{a,c}(S, S', T)] + |\mathcal{X}_{a,c}^1| + |\mathcal{X}_{a,c}^5|$$

$$= \sum_{(S,S',T)\in\mathcal{X}^1_{a,b}} 2\mathbb{E}[X_{a,c}(S,S',T)] + \sum_{(S,S',T)\in\mathcal{X}^5_{a,b}} 2\mathbb{E}[X_{a,c}(S,S'\setminus\{c\}\cup\{b\},T\setminus\{c\}\cup\{b\})]$$

$$+ |\mathcal{X}^1_{a,b}| + |\mathcal{X}^5_{a,b}|$$

$$= \binom{n-k-2}{k} \sum_{(S,S')\in\mathcal{S}_{a,b}\cap\mathcal{S}_{a,c}} \mathbb{E}[Z_{a,c}(S,S')] + \binom{n-k-3}{k-1} \sum_{(S,T)\in\mathcal{T}_{a,b}\cap\mathcal{T}_{a,c}} \mathbb{E}[Y_{a,c}(S,T)]$$

$$+ \binom{n-k-2}{k-1} \sum_{(S,S')\in\mathcal{S}_{a,b}|c\in S'} \mathbb{E}[Z_{a,c}(S,S'\setminus\{c\}\cup\{b\})]$$

$$+ \binom{n-k-2}{k-2} \sum_{(S,T)\in\mathcal{T}_{a,b}|c\in T} \mathbb{E}[Y_{a,c}(S,T\setminus\{c\}\cup\{b\})]$$

$$= \binom{n-k-2}{k} \sum_{(S,S')\in\mathcal{S}_{a,b}\cap\mathcal{S}_{a,c}} \mathbb{E}[Z_{a,c}(S,S')]$$

$$+ [\dots] + \binom{n-k-2}{k-2} \sum_{(S,T)\in\mathcal{T}_{a,b}|c\in T} \mathbb{E}[Y_{a,c}(S,T\setminus\{c\}\cup\{b\})]$$

$$= \left(\binom{n-k-2}{k} - \binom{n-k-2}{k-2}\right) \sum_{(S,S')\in\mathcal{S}_{a,b}\cap\mathcal{S}_{a,c}} \mathbb{E}[Z_{a,c}(S,S')] + [\dots]$$

$$+ \binom{n-k-2}{k-2} \sum_{(S,S')\in\mathcal{S}_{a,b}\cap\mathcal{S}_{a,c}} \mathbb{E}[Z_{a,c}(S,S')] + \mathbb{E}[Y_{a,c}(S,S'\cup\{b\})]$$

$$\geq \left(\binom{n-k-2}{k} - \binom{n-k-2}{k-2}\right) \sum_{(S,S')\in\mathcal{S}_{a,b}\cap\mathcal{S}_{a,c}} \mathbb{E}[Z_{a,b}(S,S')] + [\dots]$$

$$+ \binom{n-k-2}{k-2} \sum_{(S,S')\in\mathcal{S}_{a,b}\cap\mathcal{S}_{a,c}} \mathbb{E}[Z_{a,b}(S,S')] + \mathbb{E}[Y_{a,b}(S,S'\cup\{c\})]$$

$$= \binom{n-k-2}{k} \sum_{(S,S')\in\mathcal{S}_{a,b}\cap\mathcal{S}_{a,c}} \mathbb{E}[Z_{a,b}(S,S')] + [\dots]$$

$$+ \binom{n-k-2}{k-2} \sum_{(S,S')\in\mathcal{S}_{a,b}\cap\mathcal{S}_{a,c}} \mathbb{E}[Y_{a,b}(S,S'\cup\{c\})]$$

$$= \binom{n-k-2}{k} \sum_{(S,S')\in\mathcal{S}_{a,b}\cap\mathcal{S}_{a,c}} \mathbb{E}[Z_{a,b}(S,S')] + [\dots]$$

$$+ \binom{n-k-2}{k-2} \sum_{(S,S')\mathcal{T}_{a,b}|c\in T} \mathbb{E}[Y_{a,b}(S,T)]$$

$$= \binom{n-k-2}{k} \sum_{(S,S')\in\mathcal{S}_{a,b}\cap\mathcal{S}_{a,c}} \mathbb{E}[Z_{a,b}(S,S')] + \binom{n-k-3}{k-1} \sum_{(S,T)\in\mathcal{T}_{a,b}\cap\mathcal{T}_{a,c}} \mathbb{E}[Y_{a,c}(S,T)]$$

$$+ \binom{n-k-2}{k-1} \sum_{(S,S')\in\mathcal{S}_{a,b}|c\in S'} \mathbb{E}[Z_{a,c}(S,S'\setminus\{c\}\cup\{b\})]$$

$$+ \binom{n-k-2}{k-2} \sum_{(S,T)\in\mathcal{T}_{a,b}|c\in T} \mathbb{E}[Y_{a,b}(S,T)]$$

$$\geq \binom{n-k-2}{k} \sum_{(S,S')\in\mathcal{S}_{a,b}\cap\mathcal{S}_{a,c}} \mathbb{E}[Z_{a,b}(S,S')] + \binom{n-k-3}{k-1} \sum_{(S,T)\in\mathcal{T}_{a,b}\cap\mathcal{T}_{a,c}} \mathbb{E}[Y_{a,b}(S,T)]$$

$$+ \binom{n-k-2}{k-1} \sum_{(S,S')\in\mathcal{S}_{a,b}|c\in S'} \mathbb{E}[Z_{a,b}(S,S')] + \binom{n-k-2}{k-2} \sum_{(S,T)\in\mathcal{T}_{a,b}|c\in T} \mathbb{E}[Y_{a,b}(S,T)]$$

$$= \sum_{(S,S',T)\in\mathcal{X}_{a,b}^1} 2\mathbb{E}[X_{a,b}(S,S',T)] + \sum_{(S,S',T)\in\mathcal{X}_{a,b}^5} 2\mathbb{E}[X_{a,b}(S,S',T)] + |\mathcal{X}_{a,c}^1| + |\mathcal{X}_{a,c}^5|,$$

where the first inequality follows by equation (7) and (4) and the second inequality follows from equation (4) and (5). The dots ([...]) stands for

$$\binom{n-k-3}{k-1} \sum_{(S,T)\in\mathcal{T}_{a,b}\cap\mathcal{T}_{a,c}} \mathbb{E}[Y_{a,c}(S,T)] + \binom{n-k-2}{k-1} \sum_{(S,S')\in\mathcal{S}_{a,b}|c\in S'} \mathbb{E}[Z_{a,c}(S, S'\backslash\{c\}\cup\{b\})],$$

which is a part of the expression that it is omitted during the calculations for the sake of brevity. Summarizing, we get that

$$\mathbb{E}[X_{a,c}] = \frac{\sum_{(S,S',T)\in\mathcal{X}_{a,c}} \mathbb{E}[X_{a,c}(S,S',T)]}{|\mathcal{X}_{a,c}|} = \frac{\sum_{i=1}^5 \sum_{(S,S',T)\in\mathcal{X}_{a,c}^i} \mathbb{E}[X_{a,c}(S,S',T)]}{|\mathcal{X}_{a,c}|}$$

$$\geq \frac{\sum_{i=1}^5 \sum_{(S,S',T)\in\mathcal{X}_{a,b}^i} \mathbb{E}[X_{a,b}(S,S',T)]}{|\mathcal{X}_{a,c}|} = \mathbb{E}[X_{a,b}],$$

where the inequality follows from equations (6), (8), and (9). The last inequality follows from $|\mathcal{X}_{a,b}| = |\mathcal{X}_{a,c}|$. $\qquad\square$

**Connection to the dueling bandit gap parameter**  In what follows, we will show that for $k = 1$, our gap ($\Delta$) is at least of the same order as the usual gap parameter for dueling bandit setting (will be denoted by $\Delta^{bandit}$), namely that $\Delta = \Omega(\Delta^{bandit})$.

Consider $k = 1$ (where each team contains a single player). In this case, an instance of our problem becomes a dueling bandit instance. In addition, consider the goal of identifying the best arm (here the best team is simply the best arm), and denote

$$\Delta^{bandit} = P_{1,2}^{bandit} - 1/2,$$

Where $\Delta^{bandit}$ is the common gap definition in dueling bandit.

The characterization for deducible relations from Section 3 applies in particular for $k = 1$, and we have that $\mathcal{S}_{1,2} = \{\emptyset\}$ and $\mathcal{T}_{1,2} = \{(\emptyset, i)\}_{i\notin\{1,2\}}$.

Using the definition of $\Delta$ and $X_{1,2}(S,S',T)$, we get

$$\Delta = \mathbb{E}[X_{1,2}] = \frac{1}{4}(\mathbb{E}[P_{1,2}] - \mathbb{E}[P_{2,1}] + \mathbb{E}_{i\notin 1,2}[P_{1,i}] - \mathbb{E}_{i\notin 1,2}[P_{2,i}]) =$$

$$= \frac{1}{4}(2P_{1,2} - 1 + \mathbb{E}_{i\notin 1,2}[P_{1,i} - P_{2,i}]) \geq \frac{1}{2}\left(P_{1,2} - \frac{1}{2}\right) = \frac{1}{2}\Delta^{bandit} = \Omega(\Delta^{bandit}).$$

The inequality is due to SST as $1 \succ 2 \succ i$ for all $i \notin \{1,2\}$ implies $P_{1,i} \geq P_{2,i}$.

**The reduction**  We close this section by giving the two subroutines mentioned within the reduction to the classic dueling bandits setting.

---

**Algorithm 1** singlesDuel: simulation of a duel between single players

---
**Input:** Players $a, b \in [n]$
**Output:** $w \in \{0,1\}$ such that $w = 1$ if $a$ won and $w = 0$ if $b$ won.
Pick $(S, S', T) \in \mathcal{X}_{a,b}$ randomly
$z \leftarrow (\mathbb{1}[\{a\}\cup S > \{b\}\cup S'] + \mathbb{1}[\{a\}\cup S' > \{b\}\cup S])/2$
$y \leftarrow (\mathbb{1}[\{a\}\cup S > T] + \mathbb{1}[T > \{b\}\cup S])/2$
$x \leftarrow (z + y - 1)/2$
**return** sample of a biased coin with bias $1/2 + x$

---

# C   Algorithms and Proofs of Section 5

**Uncover Subroutine**   As sketched within the main part of our paper, we refine the idea of the *Uncover* subroutine by a binary search approach. Moreover, we add the option to input a refinement of $A$ and $B$, namely $A = A^{(1)} \cup A^{(2)}$, $B = B^{(1)} \cup B^{(2)}$, guaranteeing that the uncovered relation is between a pair of players from $A^{(1)}$ and $B^{(1)}$, while $A^{(2)}$ and $B^{(2)}$ are contained in one of the sets of the witness each. For that to work, we require that

(a) $|A^{(1)}| + |A^{(2)}| = k$,

(b) $|A^{(i)}| = |B^{(i)}|$ for $i \in \{1, 2\}$,

(c) $A^{(1)} \cup A^{(2)} \succ B^{(1)} \cup B^{(2)}$, and

(d) $A^{(1)} \cup B^{(2)} \succ B^{(1)} \cup A^{(2)}$.

Observe that for any four sets satisfying $(a)$ and $(b)$ one of the four sets wins in both duels. By enforcing $(c)$ and $(d)$ we fix wlog that this set is $A^{(1)}$. Let us assume that the sets $A^{(1)}$ and $B^{(1)}$ are ordered, meaning that $A^{(1)} = \{a_1, \ldots, a_{|A^{(1)}|}\}$ and $B^{(1)} = \{b_1, \ldots, b_{|A^{(1)}|}\}$. We also introduce the shorthand notation $A_{\ell:r}$ for $\{a_\ell, \ldots, a_r\}$ and respectively $B_{\ell:r}$ for $\{b_\ell, \ldots, b_r\}$ for any $\ell, r \in [|A^{(1)}|]$. The subroutine is formalized in Algorithm 2.

---

**Algorithm 2** Uncover Subroutine

---

**Input:**   four disjoint sets, $A^{(1)}, B^{(1)}, A^{(2)}, B^{(2)}$ with $|A^{(1)}| = |B^{(1)}|, |A^{(2)}| = |B^{(2)}|, |A^{(1)}| + |A^{(2)}| = k$, $A^{(1)} \cup A^{(2)} \succ B^{(1)} \cup B^{(2)}$, and $A^{(1)} \cup B^{(2)} \succ B^{(1)} \cup A^{(2)}$

**Output:**   $a \in A^{(1)}, b \in B^{(1)}, (S, S') \in \mathcal{S}_{a,b}^*$ with $(C \subseteq S$ and $D \subseteq S')$ or $(D \subseteq S$ and $C \subseteq S')$

Set $S \leftarrow A^{(1)} \cup A^{(2)}, T \leftarrow B^{(1)} \cup B^{(2)}, \ell \leftarrow 1, r \leftarrow |A^{(1)}|$
**while** $\ell < r$ **do**
  $i \leftarrow \lfloor \frac{\ell+r}{2} \rfloor$
  $S \leftarrow S - A_{i+1:r} \cup B_{i+1:r}$
  $T \leftarrow T - B_{i+1:r} \cup A_{i+1:r}$
  **if** $S \succ T$ **then**
    $r \leftarrow i$
  **else**
    $\ell \leftarrow i + 1$
    swap $S$ and $T$
  **end if**
**end while**
**return** $(a_\ell, b_\ell)$, and $(S \setminus \{a_\ell\}, T \setminus \{b_\ell\})$

---

In order to show that the algorithm is well-defined and works correctly, the following Lemma will be helpful.

**Lemma C.1.** *In subroutine Uncover (Algorithm 2), at the end of every while loop, it holds that, (i) $\ell, r \in \mathbb{N}$ with $\ell \leq r$, (ii) $A_{\ell:r} \subseteq S$, $B_{\ell:r} \subseteq T$, (iii) $S \succ T$, and (iv) $T \setminus B_{\ell:r} \cup A_{\ell:r} \succ S \setminus A_{\ell:r} \cup B_{\ell:r}$, (v) exactly one of $S$ and $T$ contains $A^{(2)}$, the other set contains $B^{(2)}$.*

*Proof.*   We prove all statements via one joint induction over the iterations of the while loop. All statements are clearly true at the beginning of the first while loop. Now, consider any iteration in which the four statements are true at the beginning of the while loop. It suffices to show that they are still true after resetting $S$, $T$, $\ell$, and $r$. For clarity, we refer to the modified variables of the teams just before the if condition as $S'$, $T'$ and after the if condition as $S''$, $T''$. Similarly, $\ell'$, and $r'$ are the values of the indices after the if condition. In the following, we show that the four conditions still hold for $S''$, $T''$, $\ell'$, and $r'$.

**Case 1:** $S' \succ T'$. Then, $S'' = S', T'' = T', \ell' = \ell, r' = i$. The condition of the while loop, $\ell < r$, clearly implies that $\ell' = \ell \leq \lfloor \frac{\ell+r}{2} \rfloor = i = r'$. Moreover, by construction $A_{\ell:i} = A_{\ell':r'} \subseteq S''$ and

$B_{\ell:i} = B_{\ell':r'} \subseteq T''$ and hence condition $(ii)$ is satisfied. Condition $(iii)$, i.e., $S'' \succ T''$ is satisfied by the case condition. For condition $(iv)$ let us rewrite the induction hypothesis for condition $(iv)$ as

$$T - B_{\ell:r} \cup (A_{\ell:i} \cup A_{i+1:r}) \succ S - A_{\ell:r} \cup (B_{\ell:i} \cup B_{i+1:r}).$$

Observe that $T - B_{\ell:r} \cup A_{i+1:r} = T' - B_{\ell:i}$ and $S - A_{\ell:r} \cup B_{i+1:r} = S' - A_{\ell:i}$. Hence, the above expression can be rewritten as

$$T' - B_{\ell:i} \cup A_{\ell:i} \succ S' - A_{\ell:i} \cup B_{\ell:i}.$$

Plugging in $T' = T''$, $S' = S''$, $\ell = \ell'$ and $i = r'$ yields condition $(iv)$ for the updated variables. Lastly, condition $(v)$ is satisfied directly by applying the induction hypothesis.

**Case 2:** $T' \succ S'$. Then, $S'' = T'$, $T'' = S'$, $\ell' = i + 1$, $r' = r$. For condition $(ii)$, observe that $\ell, r \in \mathbb{N}$ with $\ell < r$ clearly implies that $\ell' = i + 1 = \lfloor \frac{\ell+r}{2} \rfloor + 1 \le \lfloor \frac{2r-1}{2} \rfloor + 1 \le r = r'$. Moreover, by construction $A_{i+1:r} \subseteq T' = S''$ and $B_{i+1:r} \subseteq S' = T''$ and hence $(ii)$ is satisfied. Condition $(iii)$, i.e., $S'' = T' \succ S' = T''$, is satisfied by the case condition. For condition $(iv)$, let us rewrite the induction hypothesis for condition $(iii)$ as

$$S - A_{\ell:r} \cup (A_{\ell:i} \cup A_{i+1:r}) \succ T - B_{\ell:r} \cup (B_{\ell:i} \cup B_{i+1:r}).$$

Observe that $S - A_{\ell:r} \cup A_{\ell:i} = S' - B_{i+1:r}$ and $T - B_{\ell:r} \cup B_{\ell:i} = T' - A_{i+1:r}$. Hence, the above expression can be rewritten to

$$S' - B_{i+1:r} \cup A_{i+1:r} \succ T' - A_{i+1:r} \cup B_{i+1:r}.$$

Inserting $S' = T''$, $T' = S''$, $i + 1 = \ell'$ and $r = r'$ yields condition $(iv)$ for the updated variables. Lastly, condition $(v)$ is satisfied directly by applying the induction hypothesis. $\qquad\square$

With the help of Lemma C.1 it is easy to see that the algorithm is well-defined, more precisely, that the constructed tuple $(S, T)$ forms a feasible duel within every iteration of the while loop. It remains to show that the algorithm works correctly and its running time is bounded by $\mathcal{O}(log(|A^{(1)}|))$.

**Lemma C.2.** *Let $A^{(1)}, A^{(2)}, B^{(1)}, B^{(2)}$ be sets satisfying conditions $(a)$ to $(d)$. After performing $\mathcal{O}(\log(|A^{(1)}|))$ duels,* Uncover *returns $(a, b)$ with $a \in A^{(1)}$, $b \in B^{(1)}$ and $(S, S') \in \mathcal{S}_{a,b}^*$ with either $A^{(2)} \subseteq S$ and $B^{(2)} \subseteq S'$ or $B^{(2)} \subseteq S$ and $A^{(2)} \subseteq S'$.*

*Proof.* By Lemma C.1, the termination of the algorithm implies that $\ell = r$. By statement $(ii)$ from Lemma C.1 we get that $a_\ell \in S$ and $b_\ell \in T$ holds. Moreover, conditions $(iii)$ and $(iv)$ can be rewritten as

$$(S \setminus \{a_\ell\}) \cup \{a_\ell\} \succ (T \setminus \{b_\ell\}) \cup \{b_\ell\} \tag{10}$$

and

$$(T \setminus \{b_\ell\}) \cup \{a_\ell\} \succ (T \setminus \{a_\ell\}) \cup \{b_\ell\}, \tag{11}$$

respectively. Clearly, this implies that $(S \setminus \{a_\ell\}, T \setminus \{b_\ell\}) \in \mathcal{S}_{a_\ell, b_\ell}^*$ and hence $a_\ell \succ b_\ell$.

It is easy to see that the number of iterations of the while loop is upper bounded by the height of a balanced binary tree on $|A^{(1)}|$ elements, i.e., $\mathcal{O}(log(|A^{(1)}|))$. Since every iteration induces exactly one query, this also bounds the total number of queries. Moreover, by condition $(v)$ we have that one of $A^{(2)}$ is included in $S$ or $T$ and $B^{(2)}$ in the other one. This concludes the proof. $\qquad\square$

Clearly, Lemma C.2 directly implies Lemma 5.3. For this, simply call *Uncover* with $A^{(2)} = B^{(2)} = \emptyset$.

**Lemma 5.3.** *Let $A$ and $B$ be two disjoint teams with $A \succ B$. After performing $\mathcal{O}(\log(k))$ duels,* Uncover *returns $(a, b)$ with $a \in A$, $b \in B$ and $(S, S') \in \mathcal{S}_{a,b}^*$, and thus $a \succ b$.*

**Reducing the Number of Players to** $\mathcal{O}(k)$ Before formalizing the pre-processing procedure *ReducePlayers* in Algorithm 3, recall that algorithm maintains a dominance graph $D = (V, E)$ on the set of players. More precisely, the nodes of $D$ are the players, i.e., $V = [n]$, and there exists an arc from node $a$ to node $b$ if the algorithm has proven that $a \succ b$. The set $V_{<2k}$ is the subset of the players having an indegree smaller than $2k$ in $D$.

Additionally, we define a second graph $G_{<2k}$ as follows: The set of nodes of $G_{<2k}$ equals $V_{<2k}$ and there exists an (undirected) edge between two nodes $a, b \in V_{<2k}$ if and only if neither of the arcs $(a, b)$ or $(b, a)$ is present within the graph $D$. The algorithm now searches for a matching of size $k$ within the graph $G_{<2k}$ by calling the subroutine *GreedyMatching*, formalized in Algorithm 4. Let $\{(a_1, b_1), \ldots, (a_k, b_k)\}$ be such a matching. In particular, this implies that the algorithm has not identified any of the relations between $a_i$ and $b_i$ yet. Hence, when calling *uncover* for the (ordered) sets $A = \{a_1, \ldots, a_k\}$ and $B = \{b_1, \ldots, b_k\}$ (after possibly swapping $A$ and $B$), the algorithm learns about one additional pairwise relation, say $a_i \succ b_i$ and add the arc $(a_i, b_i)$ to the graph $D$. Then, the algorithm also updates $D$ to its transitive closure. The algorithm ends when it cannot find a matching of size $k$ in $G_{<2k}$ anymore. We formalize the idea within Algorithm 3.

---

**Algorithm 3** ReducePlayers

> **Input:** a set of players $[n]$
> **Output:** a set $S$ with $|S| \le 6k-2$ s.t. $A_{2k}^* \subseteq S$
> **while** $|GreedyMatching(G_{<2k})| = k$ **do**
>     Let $\{\{a_1, b_1\}, \ldots, \{a_k, b_k\}\}$ be Greedy Matching
>     Set $A = \{a_1, \ldots, a_k\}, B = \{b_1, \ldots, b_k\}$
>     $(a, b) \leftarrow uncover(A, B)$
>     Add $(a, b)$ to $D$, $D \leftarrow transitiveClosure(D)$
>     Update $V_{<2k}$ and $G_{<2k}$
> **end while**
> **return** $V_{<2k}$

**Algorithm 4** Subroutine GreedyMatching

> **Input:** an undirected Graph $G = (V, E)$
> **Output:** a matching of size at most $k$
> $M \leftarrow \emptyset$
> **while** $|M| < k$ and $E \ne \emptyset$ **do**
>     Pick arbitrary edge $(u, v)$ from $E$
>     Delete all edges incident to $u$ and $v$ from $E$
> **end while**
> **return** $M$

---

**Lemma 5.5.** *Given the set of players* $[n]$, ReducePlayers *returns* $R \subseteq [n]$ *with* $|R| \le 6k - 2$ *and* $A_{2k}^* \subseteq R$. ReducePlayers *performs* $\mathcal{O}(nk \log(k))$ *duels and runs in time* $\mathcal{O}(n^2 k^2)$.

*Proof.* Let $R$ be the set returned by *ReducePlayers*. We start by proving that $A_{2k}^* \subseteq R$. Every player not included in $R$ has at least $2k$ ingoing arcs in $D$. In other words, there exist $2k$ players which dominate it. Hence, such a player is not included in $A_{2k}^*$.

We turn to prove that $|R| \le 6k - 2$: Any *independent set* within the graph $G_{<2k}$ contains less than $2k + 1$ nodes. An independent set within $G_{<2k}$ is a subset of the nodes $T \subseteq V_{<2k}$ such that no two nodes of $T$ are connected by an edge. Now, assume for contradiction that there exists an independent set $T \subseteq V_{<2k}$ within the graph $G_{<2k}$ with $|T| = 2k + 1$. Consider the subgraph of $D$ induced by the set $T$, i.e., $D[T] = (T, \{(a, b) \in E \mid a, b \in T\})$. Since $T$ is an independent set within $G_{<2k}$, we know that $D[T]$ is a tournament graph, i.e., a directed graph in which any two nodes are connected by exactly one directed arc. Moreover, since $D[T]$ is transitive (since $\succ$ and hence $D$ is transitive), there exists exactly one node within $T$ with an indegree of $2k$ within the graph $D$. This is a contradiction to $T \subseteq V_{<2k}$.

This observation is now helpful to conclude the first part of the proof. Assume for contradiction that $|V_{<2k}| \ge 6k - 1$. Then the following greedy procedure lets us construct a matching of size $2k$ within the graph $G_{<2k}$. This yields a contradiction to the termination of the while loop, since every maximal matching, and in particular, a matching of size smaller than $k$ returned by *GreedyMatching*, is a $1/2$-approximation of a matching with maximum cardinality. Hence, the existence of a matching with $2k$ edges yields a contradiction to the fact that *GreedyMatching* did not find a matching of size $k$. We start by defining $T = V_{<2k}$ and $M = \emptyset$. Since $|T| > 2k$, $T$ is not an independent set and there exists an edge between some two nodes in $T$. Now, pick any such edge, say $\{a, b\}$, and add it to $M$ and remove $a$ and $b$ from $T$. After $i$ rounds of this procedure, $|M| = i$ and $|T| = 2k + 2(2k - i) - 1$. We can repeat this procedure for $2k$ rounds and have found a matching of size $2k$, a contradiction.

We now turn to prove the number of duels performed by the algorithm. In every step of the while loop, the algorithm adds one arc which was not existent before to the graph $D$. Moreover, since any

selected matching never includes an edge with one of its endpoints having an indegree larger than $2k - 1$, no node has an indegree higher than $2k$ after the termination of the algorithm. We can then upper bound the number of arcs within $D$ by $2kn$.

This is also a bound for the number of iterations of the while loop. Within each iteration of the while loop the algorithm needs to make one query in order to identify the winning team and in addition it calls the subroutine *uncover*. As argued within the proof of Lemma 5.5, the *uncover* subroutine induces additional $\mathcal{O}(log(k))$ queries per while loop. Summarizing, this implies that the algorithm requires $\mathcal{O}(nk \log(k))$ queries in total.

As for the running time, we have already argued that the while loop does at most $\mathcal{O}(nk)$ iterations. Within the while loop the algorithm needs to run *GreedyMatching* for finding a matching of size $k$ within $G_{<2k}$ and run the *uncover* subroutine. While the latter step requires a running time of $\mathcal{O}(\log(k))$ as argued within Lemma 5.5, *GreedyMatching* for selecting a matching of size $k$ can be implemented in $\mathcal{O}(nk)$. In total, we get a running time of $\mathcal{O}(n^2k^2)$. □

**Subroutines NewCut and Compare**  In Algorithm 5 we formalize the subroutine *NewCut*, which takes as input a subset of the players $R \subseteq [n]$, a pair of players $a, b \in X$ and a witness $(S, T') \in \mathcal{S}^*_{a,b} \cup \mathcal{T}^*_{a,b}$ and outputs a partition of $R$ into $U$ and $L$ such that $U \rhd L$ holds. We denote by $\pi_{xy}$ the permutation on subsets that exchange players $x$ and $y$. More precisely,

$$\pi_{xy}(A) = \begin{cases} A \setminus \{x\} \cup \{y\} & \text{if } x \in A, y \notin A \\ A \setminus \{y\} \cup \{x\} & \text{if } x \notin A, y \in A \\ A & \text{else.} \end{cases}$$

Before we prove the correctness of the algorithm, we introduce the following two lemmas. Strictly speaking, these are special cases of statements shown within the proof of Lemma 4.2 for the deterministic setting. For the sake of illustration, we state and prove them here for the deterministic case again, independently of Lemma 4.2.

---

**Algorithm 5** NewCut

> **Input:** $R \subseteq [n]$, a pair $a, b \in R$ and $(S, T') \in \mathcal{S}^*_{a,b} \cup \mathcal{T}^*_{a,b}$
> **Output:** Partition of $R$ into $U \rhd L$ with $a \in U$ and $b \in L$
> Initialize $\mathcal{W} \leftarrow \{(S, T', a)\}, U \leftarrow \{a\}, R \leftarrow R \setminus \{a, b\}$
> **while** $\mathcal{W}$ non-empty **do**
>    Pick $(S, T, y) \in \mathcal{W}$ and remove it from $\mathcal{W}$
>    **for** $x \in R$ **do**
>      **if** $(\pi_{xy}(S), \pi_{xy}(T')) \in \mathcal{S}^*_{xb} \cup \mathcal{T}^*_{xb}$ **then**
>       add $x$ to $U$, remove $x$ from $R$
>       add $(\pi_{xy}(S), \pi_{xy}(T'), x)$ to $\mathcal{W}$
>      **else if** $|T'| = k$ and $x \in T'$ and $(S, T' \setminus \{x\}) \in \mathcal{S}^*_{xb}$ **then**
>       add $x$ to $U$ and remove it from $R$
>       add $(S, T' \setminus \{x\}, x)$ to $\mathcal{W}$
>      **end if**
>    **end for**
> **end while**
> **return** $(U, R \cup \{b\})$

---

**Lemma C.3.** *If $a \succ b \succ c$ and $(S, S') \in \mathcal{S}^*_{b,c}$, then $(\pi_{ab}(S), \pi_{ab}(S')) \in \mathcal{S}^*_{a,c}$.*

*Proof.* We distinguish two cases. First assume $a \notin S \cup S'$. Then,

$$S \cup \{a\} \succ S \cup \{b\} \succ S' \cup \{c\},$$

where the first statement follows from single-player consistency and the second statement from $(S, S') \in \mathcal{S}^*_{bc}$. Moreover,

$$S' \cup \{a\} \succ S' \cup \{b\} \succ S \cup \{c\},$$

where again the first statement follows from single-player consistency and the second one from $(S, S') \in \mathcal{S}^*_{bc}$.

If $a \in S \cup S'$, assume wlog that $a \in S$. Then, $\pi_{ab}(S) = S \setminus \{a\} \cup \{b\}$ and $\pi_{ab}(S') = S'$. We get
$$\pi_{ab}(S) \cup \{a\} = S \cup \{b\} \succ S' \cup \{c\}$$
and
$$S' \cup \{a\} \succ S' \cup \{b\} \succ S \cup \{c\} = S \setminus \{a\} \cup \{a\} \cup \{c\} \succ S \setminus \{a\} \cup \{b\} \cup \{c\} = \pi_{ab}(S) \cup \{c\},$$
where the first and last statement follow from single player consistency and the second statement from $(S, S') \in \mathcal{S}^*_{bc}$. Summarizing, $(\pi_{ab}(S), \pi_{ab}(S')) \in \mathcal{S}^*_{ac}$. $\qquad\square$

**Lemma C.4.** *If $a \succ b \succ c$ and $(S, T) \in \mathcal{T}^*_{b,c}$, then $(\pi_{ab}(S), \pi_{ab}(T)) \in \mathcal{T}^*_{a,c}$ or $(S, T \setminus \{a\}) \in \mathcal{S}^*_{a,c}$.*

*Proof.* We distinguish three cases. First, assume that $a \notin S \cup T$. Then,
$$S \cup \{a\} \succ S \cup \{b\} \succ T \succ S \cup \{c\},$$
where the first statement follows from single player consistency and the second and third from $(S, T) \in \mathcal{T}^*_{bc}$. Next, assume $a \in S$. Then, $\pi_{ab}(S) = S \setminus \{a\} \cup \{b\}$ and we get
$$\pi_{ab}(S) \cup \{a\} = S \cup \{b\} \succ T \succ S \cup \{c\} = S \setminus \{a\} \cup \{a\} \cup \{c\} \succ S \setminus \{a\} \cup \{b\} \cup \{c\} = \pi_{ab}(S) \cup \{c\}.$$
Hence, $(\pi_{ab}(S), \pi_{ab}(T)) \in \mathcal{T}^*_{ab}$. Finally, assume $a \in T$. We get,
$$S \cup \{a\} \succ S \cup \{b\} \succ T \setminus \{a\} \cup \{a\} \succ T \setminus \{a\} \cup \{c\},$$
where the first and last statement follow from single player consistency and the second statement from $(S, T) \in \mathcal{T}^*_{bc}$. Moreover,
$$T \setminus \{a\} \cup \{a\} \succ S \cup \{c\},$$
which follows from $(S, T) \in \mathcal{T}^*_{bc}$. Summarizing, $(S, T \setminus \{a\}) \in \mathcal{S}^*_{ac}$. $\qquad\square$

Having these two lemmas, we are ready to prove the correctness of the *NewCut* subroutine.

**Lemma 5.6.** *Let $R \subseteq [n]$, $a, b \in R$ and $(S, T') \in \mathcal{S}^*_{a,b} \cup \mathcal{T}^*_{a,b}$. Then, $\mathrm{NewCut}(R, (a, b), (S, T'))$ returns a partition of $R$ into $U$ and $L$ such that $U \triangleright L$, $a \in U$ and $b \in L$. The number of duels performed by $\mathrm{NewCut}$ and its running time can be bounded by $\mathcal{O}(|R|^2)$.*

*Proof.* Let $R$ be the original set of players given as input to the algorithm, and $U$ and $L$ the returned sets. We denote by $R'$ and $U'$ the corresponding sets maintained and modified by the algorithm during its execution. To see that $U$ and $L$ form a partition of $V$, observe that $U'$ and $R'$ form a partition of $R \setminus \{b\}$ during the entire execution of the algorithm.

We turn to show that $U \triangleright L$. Assume for contradiction that there exists $c \in L$ and $d \in U$ with $c \succ d$. Since $d \in U$ we know that the algorithm found a witness for $d \succ b$ which we denote by $(S, T')$ and added $(S, T', d)$ to the list $\mathcal{W}$. Moreover, as $c \in L$, the algorithm selected $x = c$ in the for loop when $(S, T', d)$ was picked from $\mathcal{W}$. Now, if $|T'| = k - 1$, we know that $(S, T') \in \mathcal{S}^*_{d,b}$ and can apply Lemma C.3 which yields $(\pi_{cd}(S), \pi_{cd}(T')) \in S^*_{c,b}$. This is a contradiction, as otherwise $c$ would have been added to $U'$ at this point. If $|T'| = k$, we can apply Lemma C.4, yielding that either $(\pi_{cd}(S), \pi_{cd}(T')) \in \mathcal{T}^*_{c,b}$ or $(S, T' \setminus \{c\}) \in \mathcal{S}^*_{c,b}$, both of which cannot be as $c \notin U'$ at the end of the algorithm. This completes the proof of correctness.

It remains to bound the number of duels performed. Since the number of duels performed in every iteration of the for loop is constant, it suffices to bound the number of iterations of the for loop. As the algorithm adds at most $|R| - 1$ elements to $\mathcal{W}$ and for each element the for loop runs at most $|R| - 2$ times, the number of duels can be bounded by $\mathcal{O}(|R|^2)$. $\qquad\square$

We now turn to formalize the subroutine *Compare* within Algorithm 6.

---

**Algorithm 6** Compare

---

**Input:** tuple $(a, b)$, witness $(S, S') \in \mathcal{S}^*_{ab}$ and $C \subseteq S$, $D \subseteq S'$ with $|C| = |D|$
**if** $S \setminus C \cup D \cup \{a\} \succ S' \setminus D \cup C \cup \{b\}$ and $S' \setminus D \cup C \cup \{a\} \succ S \setminus C \cup D \cup \{b\}$ **then**
    **return** True
**else**
    **return** False
**end if**

---

**Lemma 5.7.** *Let $a \succ b$ be two players, $(S, S') \in \mathcal{S}^*_{a,b}$ and $C \subseteq S, D \subseteq S'$ with $|C| = |D|$. If* $\mathrm{Compare}((a, b), (S, S'), (C, D))$ *returns* True*, then $v(a) - v(b) > |v(C) - v(D)|$. Otherwise, one call to* Uncover *returns $c \in C$ and $d \in D$ together with a witness for their relation.*

*Proof.* For the sake of brevity we define $\bar{S} = S \setminus C$ and $\bar{S}' = S \setminus D$. Recall that from $(S, S') \in \mathcal{S}_{a,b}$ we get that $(i)$ $\bar{S} \cup C \cup \{a\} \succ \bar{S}' \cup D \cup \{b\}$ and $(ii)$ $\bar{S}' \cup D \cup \{a\} \succ \bar{S} \cup C \cup \{b\}$ hold. Recall that we are considering additive total orders. For any set $A \subseteq [n]$ we define $v(A) = \sum_{a \in A} v(a)$. Then, we can rewrite $(i)$ and $(ii)$ to

$$(i) \ v(\bar{S}) + v(C) + v(a) > v(\bar{S}') + v(D) + v(b)$$

and

$$(ii) \ v(\bar{S}') + v(D) + v(a) > v(\bar{S}') + v(C) + v(b).$$

Then, we distinguish two cases.

**Case 1.** $(iii)$ $\bar{S} \cup D \cup \{a\} \succ \bar{S}' \cup C \cup \{b\}$ and $(iv)$ $\bar{S}' \cup C \cup \{a\} \succ \bar{S} \cup D \cup \{b\}$. Similarly to before, we can rewrite $(iii)$ and $(iv)$ to

$$(iii) \ v(\bar{S}) + v(D) + v(a) > v(\bar{S}') + v(C) + v(b)$$

and

$$(iv) \ v(\bar{S}') + v(C) + v(a) > v(\bar{S}) + v(D) + v(b).$$

Then, from adding $(ii)$ and $(iii)$ we get that

$$v(a) - v(b) > v(C) - v(D)$$

and from adding $(i)$ and $(iv)$ we get that

$$v(a) - v(b) > v(D) - v(C).$$

Summarizing, this yields $v(a) - v(b) > |v(C)| - |v(D)|$.

**Case 2.** $(v)$ $\bar{S}' \cup C \cup \{b\} \succ \bar{S} \cup D \cup \{a\}$

In that case, observe that the quartet $(C, D, \bar{S} \cup \{a\}, \bar{S}' \cup \{b\})$ satisfies the requirements for the *Uncover* subroutine due to equation $(i)$ and $(v)$. Hence, *Uncover* will return a dominance of some player in $C$ towards some player in $D$ together with a witness for this relationship.

**Case 3.** $(vi)$ $\bar{S} \cup D \cup \{b\} \succ \bar{S}' \cup C \cup \{a\}$

In that case, observe that the quartet $(D, C, \bar{S} \cup \{b\}, \bar{S}' \cup \{a\})$ satisfies the requirements for the *Uncover* subroutine due to equation $(ii)$ and $(vi)$. Hence, *Uncover* will return a dominance of some player in $D$ towards some player in $C$ together with a witness for this relationship. □

**Algorithm CondorcetWinning** Recall that the algorithm maintains a partition of the players into a weak ordering, i.e., $\mathcal{T} = \{T_1, \dots, T_\ell\}$ with $T_1 \rhd T_2 \rhd \dots \rhd T_\ell$. We introduce the short-hand notation $T_{\leq j} = \bigcup_{m \in [j]} T_m$ and $T_{<j} = \bigcup_{m \in [j-1]} T_m$. After the application of the preprocessing procedure *ReducePlayers*, this partition consists of one set, namely $\mathcal{T} = \{T_1\}$, where $|T_1| \in \mathcal{O}(k)$ and $A^*_{2k} \subseteq T_1$. At any point in the execution of the algorithm, we are especially interested in two indices, namely $i_k \in [\ell]$ such that $|T_{<i_k}| < k < |T_{\leq i_k}|$ and similarly $i_{2k} \in [\ell]$ such that $|T_{<i_{2k}}| < 2k < |T_{\leq i_{2k}}|$. In case one of these indices does not exist, this implies that we have either identified the set $A^*_k$ or $A^*_{2k}$. In the first case, we have found a Condorcet winning team and in the second case Observation 5.4 implies that we can find one by performing one additional duel. For the sake of brevity, we disregard this case from now on.

Assuming $i_k$ is defined, observe that all players from $T_{<i_k}$ are guaranteed to be among the top-$k$ players. On the other hand, among the players from $T_{i_k}$ some belong to $A^*_k$ and others do not. The main idea of the algorithm will then be to, at any given time, take some $k$-sized prefix of $\mathcal{T}$, i.e., a subset including $T_{<i_k}$ that is included in $T_{\leq i_k}$ and either proving that this prefix is a Condorcet winning team, or showing that the partition $\mathcal{T}$ can be refined.

In the following we distinguish the cases that $i_k \neq i_{2k}$ and $i_k = i_{2k}$. For the first case we give the algorithm *CondorcetWinning1* and for the latter case the algorithm *CondorcetWinning2*. Observe that, once the *CondorcetWinning1* called *CondorcetWinning2* (which implies $i_k \neq i_{2k}$) this will be true until the termination of the algorithm.

**CondorcetWinning1** The algorithm starts by partitioning the set $T_{<i_k}$ into two sets $U_1$ and $U_2$, where $U_1$ is a prefix of $T_{<i_k}$ of size $|T_{\leq i_k}| - 2k$. It partitions the set $T_{i_k}$ into five sets $X, Y, W_1, W_2$, and $Z$. In particular it is known that $(U_1 \cup U_2) \rhd (X \cup Y \cup W_1 \cup W_2 \cup Z)$ but no relation among any pair in $T_{i_k}$ is known. Regarding the sizes of the sets it holds that $|U_i| = |W_i|$ for $i \in \{1, 2\}$, $|X| = |Y| = k - |U_1| - |U_2|$ and $|U_1| = |Z|$. The main aim of the algorithm will be to define $0 < \epsilon_1 < \epsilon_2$ and prove that the following statements are true:

(i) $|v(X) - v(Y)| < \epsilon_1$

(ii) $|v(a) - v(b)| < \epsilon_2$ for all $a \in Y \cup W_1 \cup W_2$ and $b \in Z$, and

(iii) there exist $u_1, \ldots, u_{|Z|+1} \in U_1 \cup U_2$ as well as $w_1, \ldots, w_{|Z|+1} \in W_1 \cup W_2$ such that

(a) $v(u_1) - v(w_1) \geq \epsilon_1$ and

(b) $v(u_i) - v(w_i) \geq \epsilon_2$ for all $i \in \{2, \ldots, |Z| + 1\}$.

With these three statements we can show that $U_1 \cup U_2 \cup X$ is a Condorcet winning team. More precisely, one can show that $v(U_1 \cup U_2 \cup X) - v(W_1 \cup W_2 \cup Y) > |Z| \cdot \epsilon_2$ and $v(W_1 \cup W_2 \cup Y) - v(B^*) > -|Z| \cdot \epsilon_2$, where $B^*$ is the best response towards $U_1 \cup U_2 \cup X$, i.e., $B^*$ simply contains the best $k$ players from $[n] \setminus (U_1 \cup U_2 \cup X)$. See Figure 1 for an illustration of the argument.

It remains to sketch how the algorithm defines $\epsilon_1, \epsilon_2$ and proves $(i) - (iii)$. The algorithm starts by checking whether *Uncover* can be applied to the sets $A^{(1)} = U_2, A^{(2)} = X \cup Z, B^{(1)} = W_2, B^{(2)} = Y \cup W_1$. If this is not the case, a relation between a pair in $A^{(2)}$ and $B^{(2)}$ can be found and the partition can be refined by applying *NewCut*. Otherwise, let $\bar{u} \in U_2$ and $\bar{w} \in W_2$ be the returned pair from *Uncover*. For the sake of brevity we assume for now that the entire indifference class of $\bar{u}$ in $\mathcal{T}$ is included in $U_2$. Then, using *Compare*, the algorithm checks whether $|v(X) - v(Y)| < v(\bar{u}) - v(\bar{w})$ and whether $|v(a) - v(b)| < v(\bar{u}) - v(\bar{w})$ for all $a \in W_1 \cup W_2 \cup Y$ and $b \in Z$. The algorithm repeats the process by replacing $\bar{w}$ by all $w \in W_1$. If any of the calls to *Compare* returned *False*, then we show that the partition can be refined. Otherwise, we have shown that conditions $(i) - (iii)$ are satisfied for $\epsilon_1 = v(\bar{u}) - v(w_1^*)$ and $\epsilon_2 = v(\bar{u}) - v(w_2^*)$, where $w_1^*$ and $w_2^*$ are the best and second best players from $W_1 \cup \{\bar{w}\}$, respectively. For the case when not the entire indifference class of $\bar{u}$ is included in $U_2$, we still have to exchange $\bar{u}$ by other players from its indifferent class which are included in $U_1$.

**Lemma C.5.** *After performing $\mathcal{O}(k^5)$ many duels,* CondorcetWinning1 *has identified a Condorcet winning team or called* CondorcetWinning2.

*Proof.* In part I we show that the algorithm is well-defined and that, within line 13,21,24, 29, 35, and 40, a refined partition can indeed be found. In part II we show that, if the algorithm outputs a team, this team is indeed Condorcet winning. Lastly, in part III we argue about the bound on the number of duels performed.

**Part I.** We show the first two statements by going through the algorithm line by line.

We start by showing that in line 12, the two queries are feasible. First observe that by construction, the sets $U_1, U_2, X, Y, W_1, W_2$, and $Z$ are disjoint. Moreover, $|U| = |W|$, $|U_1| = |W_1|$, and hence $|U_2| = |W_2|$. Also, $|X| = |Y|$ and $|W_1| = |Z|$. In total, we get that $|W_2| + |Y| + |W_1| = |U_1| + |X| + |Z| = |U| + |X| = k$ and the same holds for the other query as well.

Next, we show that in line 13, the partition $\mathcal{T}$ can indeed be refined. Consider wlog the case when $W_2 \cup (Y \cup W_1) \succ U_2 \cup (X \cup Z)$. Then, since $U_2 \rhd W_2$ we know that $U_2 \cup (Y \cup W_1) \succ W_2 \cup (X \cup Z)$ needs to hold. Hence, $\text{Uncover}(Y \cup W_1, X \cup Z, W_2, U_2)$ returns a pair $(a, b)$ with $a \in Y \cup W_1$ and $b \in X \cup Z$ together with a witness $(S, S') \in \mathcal{S}_{a,b}$. Since $a, b \in T_{i_k}$, we can call $\text{NewCut}(\mathcal{T}, (a, b), (S, S'))$ which returns a refined partition. An analogous argument holds for the case $W_2 \cup (X \cup Z) \succ U_2 \cup (Y \cup W_2)$.

We turn to show that the input for the $\text{Uncover}$ subroutine is valid in line 15. Since the condition in line 12 is not satisfied, we know that $U_2 \cup (X \cup Z) \succ W_2 \cup (Y \cup W_1)$ and $U_2 \cup (Y \cup W_1) \succ W_2 \cup (X \cup Z)$. This suffices to show that $(U_2, W_2, (X \cup Z), (Y \cup W_1))$ is a valid input for $\text{Uncover}$. Hence, for the returned pair $(\bar{u}, \bar{w})$ is holds that $\bar{u} \in U_2$ and $\bar{w} \in W_2$. Moreover, we can assume in the following wlog that $(X \cup Z) \subseteq S$ and $(Y \cup W_1) \subseteq S'$.

We continue with the situation in line 21 and show that a refined partition can be found. We distinguish two cases.

**Case 1** $(S, S'') \in \mathcal{S}_{\bar{u},w}$. This implies $(i)$ $S \cup \{\bar{u}\} \succ S'' \cup \{w\}$ and $(ii)$ $S'' \cup \{\bar{u}\} \succ S \cup \{w\}$. Moreover, from $(S, S'') \notin \mathcal{S}_{u,w}$ we know that either $(iii)$ $S \cup \{w\} \succ S'' \cup \{u\}$ or $(iv)$ $S'' \cup \{w\} \succ S \cup \{u\}$ is true. Assume without loss of generality that $(iii)$ holds. Then, together with $(ii)$ we get that $S'' \cup \{\bar{u}\} \succ S \cup \{w\} \succ S'' \cup \{u\}$, hence $\bar{u} \succ u$ and in particular $(S \cup \{w\}, S'') \in \mathcal{T}_{\bar{u},u}$. Since $\bar{u}$ and $u$ are from the same indifference class of $\mathcal{T}$, calling NewCut2$(\mathcal{T}, (\bar{u}, u), (S \cup \{w\}, S''))$ returns a refined partition. An analogous argument holds when $(iv)$ is true.

---

**Algorithm 7** CordorcetWinning1

1: **Input:** a partition of $[n]$ into $T_1 \vartriangleright T_2 \vartriangleright \cdots \vartriangleright T_\ell$
2: **Output:** a CondorcetWinning Team

3: **if** $i_k \neq i_{2k}$ **then**
4:     **return** CondorcetWinning2$(\mathcal{T})$
5: **end if**
6: Set $U \leftarrow T_{< i_k}$
7: Set $X$ and $Y$ to be two disjoint, $(k - |U|)$-sized subsets of $T_{i_k}$
8: Set $W$ to be a $|U|$-sized subset of $T_{i_k} \setminus X \setminus Y$
9: Set $Z$ to be $T_{i_k} \setminus X \setminus Y \setminus W$
10: Set $W_1$ to be a $|Z|$-sized subset of $W$ and $W_2 \leftarrow W \setminus W_1$
11: Set $U_1$ to be a $|Z|$-sized prefix of $U$ and $U_2 \leftarrow U \setminus U_1$
12: **if** $W_2 \cup (Y \cup W_1) \succ U_2 \cup (X \cup Z)$ or $W_2 \cup (X \cup Z) \succ U_2 \cup (Y \cup W_1)$ **then**
13:     **return** CondorcetWinning(refinedPartition)
14: **end if**
15: $(\bar{u}, \bar{w}), (S, S') \leftarrow \text{Uncover}(U_2, W_2, (X \cup Z), (Y \cup W_1))$
16: Let $\bar{T}$ be indifference class of $\bar{u}$ in $\mathcal{T}$
17: **for** $u \in \bar{T} \cap U_1 \cup \{\bar{u}\}$ **do**
18:     **for** $w \in W_1 \cup \{\bar{w}\}$ **do**
19:         $S'' \leftarrow f_{\bar{w},w}(S')$
20:         **if** $(S, S'') \notin \mathcal{S}_{u,w}$ **then**
21:             **return** CondorcetWinning(refinedPartition)
22:         **end if**
23:         **if** Compare$((u, w), (S, S''), (X, Y))$ not true **then**
24:             **return** CondorcetWinning(refinedPartition)
25:         **end if**
26:         **for** $z \in Z$ **do**
27:             **for** $q \in S'' \cap (W \cup Y)$ **do**
28:                 **if** Compare$((u, w), (S, S''), (\{z\}, \{q\}))$ not true **then**
29:                     **return** CondorcetWinning(refinedPartition)
30:                 **end if**
31:             **end for**
32:         **end for**
33:         $(Q, Q') \leftarrow (S \setminus Z \cup \pi_{w^*,w}(W_1), S'' \setminus \pi_{w^*,w}(W_1) \cup Z)$
34:         **if** $(Q, Q') \notin S_{u,w}$ **then**
35:             **return** CondorcetWinning(refinedPartition)
36:         **end if**
37:         **for** $z \in Z$ **do**
38:             **for** $w' \in Q \cap W_2$ **do**
39:                 **if** Compare$((u, w), (Q, Q'), (\{w'\}, \{z\}))$ not true **then**
40:                     **return** CondorcetWinning(refinedPartition)
41:                 **end if**
42:             **end for**
43:         **end for**
44:     **end for**
45: **end for**
46: **return** $U \cup X$

---

**Case 2** $(S, S'') \notin \mathcal{S}_{\bar{u},w}$. Then, either $(i)$ $S \cup \{w\} \succ S'' \cup \{\bar{u}\}$ or $(ii)$ $S'' \cup \{w\} \succ S \cup \{\bar{u}\}$ holds while both is not possible as $\bar{u} \succ w$. First, assume $(i)$ is true. Then, from $(S, S') \in \mathcal{S}_{\bar{u},\bar{w}}$, we know

that $(iii)$ $S' \cup \{\bar{u}\} \succ S \cup \{\bar{w}\}$. Reformulating $(i)$ to $S \cup \{w\} \succ S' \setminus \{w\} \cup \{\bar{u}\} \cup \{\bar{w}\}$ and $(iii)$ to $S' \setminus \{w\} \cup \{\bar{u}\} \cup \{w\} \succ S \cup \{\bar{w}\}$ shows that $w \succ \bar{w}$ and in particular $(S, S' \setminus \{w\} \cup \{\bar{u}\}) \in \mathcal{S}_{w,\bar{w}}$. As $w$ and $\bar{w}$ are contained in the same indifference class of $\mathscr{T}$, calling $\mathrm{NewCut}(\mathscr{T}, (w, \bar{w}), (S, S' \setminus \{w\} \cup \{\bar{u}\}))$ refines the partition. Second, assume that $(ii)$ holds. However, from $(S, S') \in \mathcal{S}_{\bar{u},\bar{w}}$ we know that $(iv)$ $S \cup \{\bar{u}\} \succ S' \cup \{\bar{w}\}$ is true. As $S'' \cup \{w\} = S' \cup \{\bar{w}\}$ this yields a contradiction to $(ii)$.

We prove that we can find a refined partition within line 24. When $\mathrm{Compare}((u, w), (S, S''), (X, Y))$ is not true, then one call to $\mathrm{Uncover}(X, Y, S \setminus X, S'' \setminus Y)$ returns a pair $(x, y)$ with $x \succ y$ (or vice versa) and a witness $(P, P') \in \mathcal{S}_{x,y}$ (or $(P, P') \in \mathcal{S}_{y,x}$) (as shown within Lemma 5.7). Since $x$ and $y$ are from the same indifference class of $\mathscr{T}$, namely $T_{i_k}$, the algorithm can call $\mathrm{NewCut}(\mathscr{T}, (x, y), (P, P'))$ and obtain a refined partition.

We continue with the situation in line 29. When $\mathrm{Compare}((u, w), (S, S''), (\{z\}, \{w'\}))$ is not true, then a call to $\mathrm{Uncover}(\{z\}, \{w'\}, S \setminus \{z\}, S'' \setminus \{w'\})$ returns the pair $(z, w')$ (or $(w', z)$) and a witness $(P, P') \in \mathcal{S}_{z,w'}$ (or $(P, P') \in \mathcal{S}_{w',z}$). Since $z$ and $w'$ are from the same indifference class of $\mathscr{T}$, namely $T_{i_k}$, the algorithm can call $\mathrm{NewCut}(\mathscr{T}, (z, w'), (P, P'))$ and obtain a refined partition.

We turn to prove that we can find a refined partition within line 35. From $(Q, Q') \notin \mathcal{S}_{u,w}$ we know that either $(i)$ $Q \cup \{w\} \succ Q' \cup \{u\}$ or $(ii)$ $Q' \cup \{w\} \succ Q \cup \{u\}$ while both are not possible as $u \succ w$. First, assume that $(i)$ holds. From $(S, S'') \in \mathcal{S}_{u,w}$ we get in particular that $(iii)$ $S'' \cup \{u\} \succ S \cup \{w\}$ holds. Rewriting $(i)$ as $\pi_{\bar{w},w}(W_1) \cup S \setminus Z \cup \{w\} \succ Z \cup S'' \setminus \pi_{\bar{w},w}(W_1) \cup \{u\}$ and $(iii)$ as $\pi_{\bar{w},w}(W_1) \cup S'' \setminus \pi_{\bar{w},w}(W_1) \cup \{u\} \succ Z \cup S \setminus Z \cup \{w\}$ establishes that we can call $\mathrm{Uncover}(\pi_{\bar{w},w}(W_1), Z, S'' \setminus \pi_{\bar{w},w}(W_1) \cup \{u\}, S \setminus Z \cup \{w\})$ which returns a pair $(\hat{w}, \hat{z})$ with $\hat{w} \in \pi_{\bar{w},w}(W_1)$ and $\hat{z} \in Z$ together with a witness for their relation. As $\hat{w}$ and $\hat{z}$ are from the same indifference class of $\mathscr{T}$ we can call $\mathrm{NewCut}$ to refine the partition. The case when $(ii)$ follows by an analogous argument.

Lastly, we show that we can find a refined partition within line 40. $\mathrm{Compare}((u, w), (Q, Q'), (\{w'\}, \{z\}))$ is a valid query as, for starters, $w' \in Q$ and $z \in Q'$. Moreover, $(Q, Q') \in \mathcal{S}_{u,w}$. Hence, if $\mathrm{Compare}$ returns False, then $\mathrm{Uncover}(\{w'\}, \{z\}, Q \setminus \{w'\}, Q' \setminus \{z\})$ returns the pair $(w', z)$ (or $(z, w')$) together with a witness from $\mathcal{S}_{w',z}$ (or $\mathcal{S}_{z,w'}$). As $z$ and $w'$ are from the same equivalence class of $\mathscr{T}$, we can call the $\mathrm{NewCut}$ and obtain a refined partition.

**Part II.** We now show that the set returned by $\mathrm{CondorcetWinning}(\mathscr{T})$ is indeed a Condorcet winning team. If, at some point of the algorithm $i_k \neq i_{2k}$, then the statement follows from Lemma C.6. Otherwise, the algorithm returns $U \cup X$ which implies that within the last call of $\mathrm{CondorcetWinning}$ none of the if conditions was satisfied. We show in the following that this implies that $U \cup X$ is a Condorcet winning team.

We define

$$w_1^* = \underset{w \in W_1 \cup \{\bar{w}\}}{\arg\max} \; v(w),$$

$$w_2^* = \underset{w \in W_1 \cup \{\bar{w}\} \setminus \{w_1^*\}}{\arg\max} \; v(w), \text{ and}$$

$$u^* = \underset{u \in \bar{T} \cap U_1}{\arg\min} \; v(u).$$

Moreover, $\epsilon_1 = v(u^*) - v(w_1^*)$ and $\epsilon_2 = v(u^*) - v(w_2^*)$.

We claim that

(i) $|v(X) - v(Y)| < \epsilon_1$, and

(ii) $|v(a) - v(b)| < \epsilon_2$ for all $a \in Y \cup W$ and $b \in Z$.

For (i) observe that there was a point within the iteration of the algorithm when $u = u^*$ and $w = w_1^*$. Moreover, the algorithm called $\mathrm{Compare}((u, w), (S, S''), (X, Y))$ which returned true. As we have argued for the subroutine $\mathrm{Compare}$, this implies $\epsilon_1 = v(u^*) - v(w_1^*) > |v(X) - v(Y)|$.

To show (ii), we distinguish three cases. Let $a \in Y \cup W$ and $z \in Z$.

**Case 1.** $a = w_1^*$. Then, there was a point within the iteration of the algorithm when $u = u^*$, $w = w_2^*$, $q = w_1^* = a$ and $z = b$. As $\mathrm{Compare}((u, w), (S, S'), (\{z\}, \{q\}))$ returned true in line 28, we

know that

$$|v(a) - v(b)| < v(u^*) - v(w_2^*) = \epsilon_2.$$

**Case 2.** $a \neq w_1^*, a \in S$. Then, there was a point within the iteration of the algorithm when $u = u^*, w = w_1^*, q = a$ and $z = b$. As $\text{Compare}((u, w), (S, S'), (\{z\}, \{q\}))$ returned true in line 28, we know that

$$|v(a) - v(b)| < v(u^*) - v(w_1^*) = \epsilon_1 < \epsilon_2.$$

**Case 3.** $a \neq w_1^*, a \in S'$. Then, there was a point within the iteration of the algorithm when $u = u^*, w = w_1^*, q = a$ and $z = b$. As $\text{Compare}((u, w), (Q, Q'), (\{z\}, \{q\}))$ returned true in line 39, we know that

$$|v(a) - v(b)| < v(u^*) - v(w_1^*) = \epsilon_1 < \epsilon_2.$$

Lastly, we show that (i) and (ii) suffice to prove that $U \cup X$ is a Condorcet winning team. To this end let $B^*$ be the best response against $U \cup X$. Observe that $B^* \subseteq Y \cup W \cup Z$.

We start by showing

$$
\begin{aligned}
&v(U \cup X) - v(W \cup Y) \\
&= v(U_1 \cup \{\bar{u}\} \setminus \{u^*\}) + v(u^*) + v(U_2 \setminus \{\bar{u}\}) + v(X) \\
&\quad - v(w_1^*) - v(W_1 \cup \{\bar{w}\} \setminus \{w_1^*\}) - v(W_2 \setminus \{\bar{w}\}) - v(Y) \\
&= v(X) - v(Y) + v(u^*) - v(w_1^*) + v(U_1 \cup \{\bar{u}\} \setminus \{u^*\}) \\
&\quad - v(W_1 \cup \{\bar{w}\} \setminus \{w_1^*\}) + v(U_2 \setminus \{\bar{u}\}) - v(W_2 \setminus \{\bar{w}\}) \\
&> -\epsilon_1 + v(u^*) - v(w_1^*) + v(U_1 \cup \{\bar{u}\} \setminus \{u^*\}) \\
&\quad - v(W_1 \cup \{\bar{w}\} \setminus \{w_1^*\}) + v(U_2 \setminus \{\bar{u}\}) - v(W_2 \setminus \{\bar{w}\}) \\
&> -\epsilon_1 + \epsilon_1 + v(U_1 \cup \{\bar{u}\} \setminus \{u^*\}) - v(W_1 \cup \{\bar{w}\} \setminus \{w_1^*\}) + v(U_2 \setminus \{\bar{u}\}) - v(W_2 \setminus \{\bar{w}\}) \\
&> -\epsilon_1 + \epsilon_1 + |Z| \cdot \epsilon_2 + v(U_2 \setminus \{\bar{u}\}) - v(W_2 \setminus \{\bar{w}\}) \\
&> -\epsilon_1 + \epsilon_1 + |Z| \cdot \epsilon_2 + 0 \\
&= |Z| \cdot \epsilon_2.
\end{aligned}
$$

The first inequality follows by (i), the second by the definition of $\epsilon_1$, the third by the definition of $\epsilon_2$ and the fact that $|u(U_1 \cup \{\bar{u}\} \setminus \{u^*\})| = |v(W_2 \cup \{\bar{w}\} \setminus \{w_1^*\})| = |Z|$, and the last by the fact that $U_2 \rhd W_2$.

In addition, we get

$$
\begin{aligned}
v(W \cup Y) - v(B^*) &= v(W \cup Y \setminus B^*) - v(B^* \cap Z) \\
&> -|Z| \cdot \epsilon_2,
\end{aligned}
$$

where the inequality follows from the fact that $|v(W \cup Y)| = |v(B^* \cap Z)| < |Z|$ and (ii).

Summing up the two inequalities yields

$$v(U \cup X) - v(B^*) > 0,$$

which concludes this part of the proof.

**Part III.** It remains to argue about the number of duels performed by *CondorcetWinning1* until it calls *CondorcetWinning2* or returns a team. We first observe that the partition $\mathscr{T}$ can be refined at most $\mathcal{O}(k)$ times. Also, the number of calls to *Uncover* can be bounded by $\mathcal{O}(k)$, since, *Uncover* is either called just before a refinement (hidden within any of the lines saying "refinedPartition") or within line 15. In the following, we will therefore bound the number of duels done within one recursive call of *CondorcetWinning1*. To this end, observe that checking whether some tuple is a subsets witness as well as calling *Compare* requires $\mathcal{O}(1)$ duels. Clearly, the number of times these operations are performed within one recursive call (before the next call is initiated) can be bounded by $\mathcal{O}(k^4)$. Putting all of this together yields that the number of duels can be bounded by $\mathcal{O}(k^5)$. □

---

**Algorithm 8** CordorcetWinning2

---

1: **Input:** a partition of $[n]$ into $T_1 \rhd T_2 \rhd \cdots \rhd T_\ell$ with $i_k \neq i_{2k}$
2: **Output:** a CondorcetWinning Team

3: $j \leftarrow \min\{k - |T_{<i_k}|, |T_{\leq i_k}| - k\}$
4: Set $X$ and $Y$ to be two disjoint, $j$-sized subsets of $T_{i_k}$
5: Set $W \leftarrow T_{i_k} \setminus X \setminus Y$
6: Set $L \leftarrow \emptyset$
7: $(*)$ Set $Z$ to be a subset of $T_{i_{2k}} \setminus L$ of size $2k - |T_{<i_{2k}}|$
8: **while** $|L| < |T_{\leq i_{2k}}| - 2k + 1$ **do**
9:     **if** $|T_{\leq i_k}| - k < k - |T_{<i_k}|$ **then**
10:        $U \leftarrow T_{<i_k} \cup W, \ V \leftarrow (T_{>i_k} \cap T_{<i_{2k}}) \cup Z$
11:     **else**
12:        $U \leftarrow T_{<i_k}, \ V \leftarrow W \cup (T_{>i_k} \cap T_{<i_{2k}}) \cup Z$
13:     **end if**
14:     **if** $V \cup Y \succ U \cup X$ or $V \cup X \succ U \cup Y$ **then**
15:        **return** CondorcetWinning(refinedPartition)
16:     **end if**
17:     $(u,v), (S, S') \leftarrow$ Uncover$(U, V, X, Y)$
18:     **if** Compare$((u,v), (S, S'), (X, Y))$ not true **then**
19:        **return** CondorcetWinning(refinedPartition)
20:     **end if**
21:     **if** $v \in Z$ **then**
22:        $L \leftarrow L \cup \{v\}$, go to $(*)$
23:     **else**
24:        **return** $U \cup X$
25:     **end if**
26: **end while**
27: **return** $U \cup X$

---

**CondorcetWinning2** We continue by formalizing the second case of the algorithm, which is formalized within Algorithm 7. Since the approach is significantly easier than the one of *CondorcetWinning1*, we directly give the proof.

**Lemma C.6.** *After performing $\mathcal{O}(k^2 \cdot \log(k))$ many duels,* CondorcetWinning2 *has output a Condorcet winning team.*

*Proof.* We start by showing that the two duels in line 14 are feasible. To this end observe that $U, V, X$ and $Y$ are disjoint by construction. To argue about their cardinalities, we consider the two cases of the if condition. First, assume $|T_{\leq i_k}| - k < k - |T_{<i_k}|$. Then

$$|U| = |T_{<i_k}| + |T_{i_k}| - 2j = |T_{\leq i_k}| - (|T_{\leq i_k}| - k) - j = k - j.$$

As $|X| = |Y| = j$, we get that $|U| + |X| = |U| + |Y| = k$. Similarly, for the other case, we have

$$|V| = |T_{<i_{2k}}| - |T_{\leq i_k}| + |Z| = |T_{<i_{2k}}| - |T_{\leq i_k}| + 2k = |T_{<i_k}| = k + k - |T_{i_k}| = k - j.$$

Hence, also $|U| + |X| = |U| + |Y| = k$.

Next, we show that we can find a refined partition in line 15. Assume wlog that $V \cup Y \succ U \cup X$ holds and observe that both statements cannot be true as $U \rhd V$ by construction. Hence, we have $U \cup Y \succ V \cup X$ which implies that we can call Uncover$(Y, X, U, V)$ which returns a pair $(y, x)$ as well as a witness from $\mathcal{S}_{y,x}$ (or $\mathcal{S}_{x,y}$). Since $x$ and $y$ are from the same indifference class of $\mathscr{T}$, namely $T_{i_k}$, we can call the NewCut subroutine and obtain a refined partition.

The call to Uncover in line 17 is feasible, as the non-satisfaction of the if condition implies that $U \cup X \succ V \cup Y$ and $U \cup Y \succ V \cup X$.

In line 19 we can refine the partition $\mathscr{T}$, as, if Compare$((u,v), (S, S'), (X, Y))$ does not return true, then Uncover$(X, Y, S \setminus X, S' \setminus Y)$ returns a pair $(x, y)$ with $x \in X$ and $y \in Y$ (or $(y, x)$) together with a witness from $\mathcal{S}_{x,y}$ (or $\mathcal{S}_{y,x}$). Since $x$ and $y$ are both from the same indifference class of $\mathscr{T}$, namely $T_{i_k}$, we can refine $\mathscr{T}$ by calling the NewCut subroutine.

Lastly, we show that $U \cup X$ is a Condorcet winning team when the algorithm reaches line 24 or line 27. We first discuss line 24. First, observe that $U \triangleright V, u \in U, v \in V$ and $(S, S')$ is a witness for their relation, that is, $(S, S') \in \mathcal{S}_{uv}$. Moreover, since Compare$((u, v), (S, S'), (X, Y))$ is true, we know that

$$v(u) - v(v) > |v(X) - v(Y)|. \tag{12}$$

Additionally we know that $v \in V \setminus Z$, which implies that $v \in T_{<i_{2k}}$. Hence, $v$ is in particular contained in the best response against $U \cup X$. Since $Y$ is also guaranteed to be within the best response, we can denote the best response by $V' \cup Y$. Using eq. (12) and the fact that $U \triangleright V'$, we get

$$
\begin{aligned}
v(U \cup X) - v(V' \cup Y) &= v(U \setminus \{u\}) + v(u) + v(X) - v(V' \setminus \{v\}) - v(v) - v(Y) \\
&> v(U \setminus \{u\}) - v(V' \setminus \{v\}) > 0,
\end{aligned}
$$

showing that $U \cup X \succ V' \cup Y$.

Now, consider the situation in line 27. This implies that the list $L$ is of length $|T_{\leq i_{2k}}| - 2k + 1$ and for each $v \in L$ there exists $u \in U$ such that

$$v(u) - v(v) > |v(X) - v(Y)|. \tag{13}$$

Again, the best response against $U \cup X$ contains $Y$. Denote the best response by $V' \cup Y$. By the size of $L$ we know that $V' \cap L \neq \emptyset$. Let $v$ be a node in the intersection and $u$ be the node for which the algorithm has proven eq. (13). Due to the same argumentation as before, $U \triangleright V'$ and $v(u') - v(v') > v(X) - v(Y)$ implies $U \cup X \succ V' \cup Y$.

It remains to argue about the number of duels performed by *CondorcetWinning2*. Again, it is clear that the partition $\mathcal{T}$ can be refined at most $\mathcal{O}(k)$ times. Per refinement, the is one additional call to *Uncover* which is bounded by $\mathcal{O}(\log(k))$ duels. Moreover, the iterations of the while loop can be bounded by $\mathcal{O}(k)$. Within one iteration the algorithm performs *Compare* (requiring $\mathcal{O}(1)$ duels) and *Uncover* (requiring $\mathcal{O}(\log(k))$ duels). Putting everything together, the number of duels can hence be bounded by $\mathcal{O}(k^2 \log(k))$. □

Putting Lemma C.5 and Lemma C.6 together clearly yields the proof of Lemma 5.8.

**Lemma 5.8.** *For every instance with $\mathcal{O}(k)$ players, after performing $\mathcal{O}(k^5)$ many duels,* CondorcetWinning1 *has identified a Condorcet winning team.* CondorcetWinning2 *identifies a Condorcet winning team after $\mathcal{O}(k^2 \log(k))$ duels.*

**Extension to a stochastic environment**    In the following we sketch how we can reduce any stochastic instance satisfying $|P_{A,B} - 1/2| \in [1/2 + \theta, 1]$ to our deterministic setting. To achieve such a reduction, simulate each deterministic duel by $\mathcal{O}(\frac{\ln m/\delta}{\theta^2})$ stochastic duels to determine the duel's winner with probability at least $1 - \delta/m$, where $\mathcal{O}(m)$ is the sample complexity of an algorithm that finds a Condorcet winning team in the deterministic case. An invocation of Chernoff-Hoeffding concentration bound yields that each duel's winner is correctly determined by this simulation with probability at least $1 - \delta/m$, and applying union bound over the total number of duels results in an algorithm that requires $\mathcal{O}(m\frac{\ln m/\delta}{\theta^2})$ team duels to identify a Condorcet winning team with probability at least $1 - \delta$.

# D    Algorithms and Proofs of Section 6

**Regret Bound** We will now show a regret bound of $\mathcal{O}(n(\Delta^{-2}(\log(T) + \log \log \Delta^{-1})))$ for team duels. Using the second part of Theorem 4.5, we can choose $\delta = 1/(Tn)$ and derive a regret bound of

$$R_T = (1 - (Tn)^{-1}) \cdot n(\Delta^{-2}(\log(T) + \log \log \Delta^{-1}) + T\frac{1}{Tn} = \mathcal{O}(n(\Delta^{-2}(\log(T) + \log \log \Delta^{-1})).$$

This follows from the SST of the distinguishabilities (Lemma 4.2) implies $\Delta_i \geq \Delta$ for all $i \in [n]$.

**Lower Bounds** We show that in the deterministic setting with additive total orders, an algorithm needs to perform at least $n - 2k$ duels in order to identify a Condorcet winning team. Clearly, this result carries over to the more general stochastic setting.

**Theorem D.1.** *In the deterministic setting with additive total orders, any algorithm identifying a Condorcet winning team performs at least $n - 2k$ duels.*

*Proof.* Consider an adversary that fixes, over time, a reverse lexicographical order, i.e., a duel is decided against the worst player participating. When the algorithm performs its first duel, the adversary picks an arbitrary player from the duel, makes him player $n$ and answer the query accordingly. Then, whenever the algorithm performs a duel containing a player which has already been fixed, the adversary decides the duel against the worst fixed player participating. Otherwise, he picks an arbitrary player from the duel and fixes him to become player $n - t$, where $t$ is the number of so far fixed players. As long as $t < n - 2k$, the algorithm cannot identify a Condorcet winning team.

Observe that the described order is additive total, as it can be realized by assigning strongly decreasing values to the players. $\square$

Observe that, when $k$ is small (i.e. constant), our upper and lower bounds match. Deriving stronger lower bounds for our setting, especially in dependency on the team size, is an interesting question for future work.

## E   Additive Total Orders

In the following we provide a sufficient condition for assigning values to players in a way that complies with a total order on teams, assuming that each team has value of the cumulative values of it's players and that team $A$ is better than team $B$ if and only if the value of $A$ is larger than the value of $B$. Formally:

**Given:** A set of players $[n]$ and a total order $\succ$ on the subsets of size $k$.

**Question:** Do there exist values for the players representing this order? Or more precisely, does the following system of linear inequalities have a feasible solution?

We denote define $\mathcal{D} = \{(A, B) \mid A, B \text{ are teams}, A \succ B\}$.

$$\sum_{b \in B} x_b - \sum_{a \in A} x_a \leq -1 \text{ for all } (A, B) \in \mathcal{D}$$
$$x_a \geq 0 \text{ for all } a \in [n]$$

We remark that, alternatively to $-1$ on the right hand side, we could have chosen any other negative number.

The following is a variant of Farkas Lemma:

**Lemma E.1** (Farkas' Lemma (Farkas, 1902)). *Let $n, m \in \mathbb{N}$, $A \in \mathbb{R}^{n \times m}$ and $b \in \mathbb{R}^m$. Then, exactly one of the following is true.*

1. $\exists x \in \mathbb{R}^n, Ax \leq b, x \geq 0$

2. $\exists y \in \mathbb{R}^m, y^T A \geq 0, y \geq 0$ *and* $y^T b < 0$.

Imagine the system above in matrix form $Ax$, then the system $y^T A \geq 0, y^T b < 0, y \geq 0$ looks as follows:

$$\sum_{(A,B) \in \mathcal{D}: i \in B} y_{AB} - \sum_{(A,B) \in \mathcal{D}: i \in A} y_{AB} \geq 0 \text{ for all players } i \in [n]$$
$$y_{AB} \geq 0 \text{ for all } (A, B) \in \mathcal{D}$$
$$\sum_{(A,B) \in \mathcal{D}} y_{AB} > 0$$

Assume the second system does have a feasible solution $y \geq 0$. In particular, there exists one pair $A \succ B$ for which $y_{AB} > 0$. We can assume wlog that this solution is rational and by scaling it up that it is integer.

We define the following condition:

**Condition (*)** There exist $\mathcal{A} = \{A_1, \ldots, A_m\}$ and $\mathcal{B} = \{B_1, \ldots, B_m\}$ satisfying the following two conditions:

   (i) $A_j \succ B_j$ for all $j \in [m]$
   (ii) Let $n_i^{\mathcal{A}}$ be the number of times that player $i$ is included in some element of $\mathcal{A}$. Define $n_i^{\mathcal{B}}$ analogously. Then, $n_i^{\mathcal{A}} = n_i^{\mathcal{B}}$ for all players $i \in [n]$.

**Claim E.2.** *The second system of linear inequalities has a feasible solution if and only if $(*)$ is satisfied.*

*Proof.* " $\Rightarrow$ " Assume the second system has a feasible (and wlog integral) solution $y$. We construct $\mathcal{A}$ and $\mathcal{B}$ as follows: For each pair $A \succ B$ for which $y_{AB} > 0$, add exactly $y_{AB}$ copies of A and B to $\mathcal{A}$ and $\mathcal{B}$, respectively. The first constraints for condition $(*)$ is clearly satisfied. Now, assume for contradiction that there exists a player $i \in [n]$ for which $n_i^{\mathcal{A}} > n_i^{\mathcal{B}}$ holds. Then, we get

$$\sum_{(A,B)\in\mathcal{D}:i\in B} y_{AB} - \sum_{(A,B)\in\mathcal{D}:i\in A} y_{AB} = n_i^{\mathcal{B}} - n_i^{\mathcal{A}} < 0,$$

a contradiction to the feasibility of $y$. On the other hand, assume that there exists a player $i \in [n]$ for which $n_i^{\mathcal{A}} < n_i^{\mathcal{B}}$ holds. Observe that

$$\sum_{j\in[n]} n_j^{\mathcal{A}} = \sum_{j\in[n]} n_j^{\mathcal{B}} = |\mathcal{A}|k$$

and hence

$$\sum_{j\in[n]\setminus\{i\}} n_j^{\mathcal{A}} > \sum_{j\in[n]\setminus\{i\}} n_j^{\mathcal{B}},$$

which implies that there exists some $i' \in [n] \setminus \{i\}$ with $n_{i'}^{\mathcal{A}} > n_{i'}^{\mathcal{B}}$, a contradiction.

" $\Leftarrow$ " Assume that there exist $\mathcal{A}$ and $\mathcal{B}$ satisfying condition $(*)$. Then, set $y_{A_j,B_j} = |\{q \in [m] : (A_q, B_q) = (A_j, B_j)\}|$ for all $j \in [m]$ and $y_{A,B} = 0$ for all other duels. This is a feasible solution to the second system of inequalities. $\square$

This directly yields the sufficient condition for a total order to be representable by values.

**Corollary E.3.** *There exists a solution to the first system of inequalities if and only if condition $(*)$ does not hold.*