# OpenReview forum: "Dueling Bandits with Team Comparisons"
_NeurIPS.cc/2021/Conference — NeurIPS 2021 Poster_

### Official Review · Reviewer_HAJh · 2021-07-11

**Rating:** 6
**Confidence:** 3

**Summary:**

A new dueling bandit framework with team comparisons.

**Main Review:**

This paper presents an interesting and non-trival generalization of dueling bandits by considering the duel between teams rather than single player. I can see this is a solid work with new framework and insights but the current presentation needs improvement and makes me feel hard to read.

The intro and Section 2 are generally good. The description of the problem is clear to me. One question is about the consistency assumption. Is that kind of an implicit gap assumption on the performance of a single player? That means for play 1 should be much better than other players such that no matter what teammates it has, this team is always better. This should be not required for standard dueling bandits? One minor is Line 64, there is no definition of set S.

I can understand what is included in Section 3 but it's better to clearly state why we need this. I think a standard flow might be stating the problem, defining performance metrices, having the algorithm and proving the theorem.

Section 4 starts to confuse me. Do you propose new algorithm eventually? Becuase you write any dueling bandit algorithm for top k identification can solve your problem. In orignial dueling bandits paper, they properly define the measure of regret. But in this paper, there is no such measure. I think it's better to comment on that.

Section 5 is not well-organized. It's pretty long and dense. And it's hard to get the key point. Will not determinstic setting be a special case of stochastic setting? There are too many newly defined concepts in this section. If you have new algorithms, it's better to have an algorithm box here. And it looks like the algorithm and theorem always mix together.

No conclusion is included.

**Time Spent Reviewing:**

3

---

> ### Author Response · Authors · 2021-08-10
> **Response to reviewer #HAJh**
>
> Consistency: Our assumption is significantly weaker. For example, our setting allows the instance where $\\{2,3,n-k-2,..,n\\} \succ \\{1,4,5,...,k+2\\}$.
> In the standard dueling bandit setting the goal is to identify the best arm or to minimize regret, thus the gap is usually defined w.r.t the distinguishability of first and second best players.
>
> In our work, the defined gap (which appears in Section 4) is aligned with the dueling bandit setting where the goal is to identify the top$-k$ players, and is defined w.r.t. The distinguishability of the $k-$th and the $(k+1)-$th top players.
>
> Algorithms in Section 4: The novelty of our work for the stochastic case relies heavily on the result obtained in Thm 3.2 in Section 3, namely a necessary and sufficient condition for any deducible pairwise players relations via team duels (as they cannot be observed directly).
>
> Building on that, in Section 4 we introduced a random variable, $X_{a,b}$, that could get positive values iff it is possible to deduce that player $a$ is better than $b$ by team duels alone.
>
> We then explained how to get an unbiased sample of this r.v. by performing $4$ team duels. Then, we showed that the probabilities instance $P_{a,b}=E[X_{a,b}]+1/2$ for every $a\ne b\in [n]$ hold all the common dueling assumptions of dueling bandits (Namely, that for every $a\ne b\in [n]$ it holds that $P_{a,b}=1-P_{b,a}$,  $P_{a,b}>1/2$ yields $a\succ b$, linear order over players and SST w.r.t. this linear order.).
>
> This allows us to reduce an instance of our problem to a dueling bandit instance and derive a bound over the number of queries needed to find the top $k$ players w.h.p.
>
> That being said, in the last paragraph of Appendix C we explain how to use some of the novel algorithms of the deterministic in a stochastic environment.
>
> Regret: We refer the reviewer to 'Extensions and discussion' (Section D in Appendix), where we define and derive a regret bound for this problem.
>
> Section 5: It is correct that the deterministic setting is a special case of the stochastic setting. However, by focusing on this important special case, we were able to drop the assumption that the gap-parameter $\Delta$ is strictly larger than $0$.
>
> In particular, we presented two new algorithms which are combinatorial in nature and independent of $\Delta$. Due to their complexity and to improve readability, we divided the algorithms into 5 subroutines.
> All subroutines are written down formally and in algorithm environments within appendix section C.
>
> Due to space constraints, we only gave sketches of the subroutines together with their formal theorem statements (which are all fully-proven in the appendix) within the main text.
>
> Taking your feedback into account and using the extra-content page provided in the camera-ready version, we will revise this section to clarify these points and improve readability.
>
>
> Conclusion: Due to space constraints, we summarized our contributions within the introduction section and referred the readers into the 'Extensions and discussion' (Section D in the appendix).
>
>
> We will improve the presentation and flow of the paper as the reviewer advises: (i) We will provide a detailed explanation about the consistency assumption and the definition of the gap for this setting, and supply examples. (ii) We will add an overview of why we need Section 3 and how we are going to use the results obtained in this section. Specifically, how do the results in Section 3 allow the reduction of the problem into the framework of dueling bandits. (iii) Use the extra page in the proceeding to add a conclusion section.

---

### Official Review · Reviewer_6N8R · 2021-07-16

**Rating:** 7
**Confidence:** 2

**Summary:**

The paper introduces a new problem in which a learner can choose k disjoint teams from a set of n players and observe a winner. The objective is to pick a team that would win against any other team with probability at least 0.5 (Condorcet winner) and to do with a minimal number of duels. Deterministic and Stochastic settings are considered and upper bounds on the number of duels are obtained.

**Limitations And Societal Impact:**

I think limitations/societal impacts were adequately addressed.

**Main Review:**

-The problem seems relevant and an interesting generalization of the duelling bandits setting. Further, various settings are considered and the solutions seem well-thought out.

-Can the notion of the Condorcet winner be generalized? i.e. winning against any other team with probability 3/4 or in general p>= 1/2? How would that look like and affect the algorithms?

-Isn't the problem statement restrictive, specifically the learner is choosing teams of k players. Suppose instead that the learner can choose teams of up to k, would this change things significantly? This would enable pairwise relations between single players.

-Is the consistency assumption realistic? Since it ignores the interaction between different players. E.g., we might have a>b, but if c complements b much better than a, we could have {b,c} > {a,c}.

-Although there are significant differences, I believe the problem of communications complexity in stable marriage could be relevant, see e.g. Gonczarowski et al "A Stable Marriage Requires Communication". I wonder if the authors see any connections or applications of their techniques especially from the deterministic setting, etc.

-Lack of experiments: Even for work that is heavily theoretical, there is usually an experiments section.

**Time Spent Reviewing:**

2

---

> ### Author Response · Authors · 2021-08-10
> **Response to reviewer #6N8R**
>
> Condorcet winner notion generalization: That's an interesting idea for a generalization. From the assumptions of our model, if there exists a Condorcet winning team that wins against any other disjoint team with probability $p>1/2$, then from the linear order on teams and the SST the top $k$ players also form such a team (i.e., team $\{1,..,k\}$ is a Condorcet winning team that wins against any other disjoint team with a probability of at least $p$.).
>
> Here is how to extend our setting for the reviewer's generalization idea: run the suggested algorithm for the stochastic case on the entire set of players [n], and denote the returned team (of $k$ players) by $\hat{A}_1$.
> Run it again on $[n]-\hat{A}_1$ and denote the output by $\hat{A}_2$.
>
> Notice that with high probability, $\hat{A}_1$ is the team of top $k$ players and $\hat{A}_2$ is the best team that can compete in a duel against $\hat{A}_1$. Then, repeat the duel ($\hat{A}_1$, $\hat{A}_2$) for $\Theta(\frac{\ln(1/\delta)}{(p-1/2)^2})$ times.
>
> If the empirical probability that $\hat{A}_1$ wins against $\hat{A}_2$ is at least $p$, then from SST w.h.p. (of at least $1-3\delta$), $\hat{A}_1$ wins against any other disjoint team with probability at least $p$. If not, then w.h.p. no such team exists.
>
> Choose teams of up to $k$: It seems likely that estimating pairwise relations between single players would indeed be the best we can do. Rather than having a different value for the gap between the $k-$th and the $(k+1)-$th best players, the upper bound will be the same.
>
> However, as assumed in some of the Combinatorial bandits works, such flexibility is not always available to the learner, and having the constraint of teams of exact size $k>1$ as we deal with here makes the problem significantly harder. This occurs naturally in tennis doubles and Relay races (as in the 2021 Olympics).
>
> Consistency assumption: Here is how we think about the assumptions of the model: there exists a linear order on teams which is consistent with some linear order over the players. While this assumption does not take into account complements skills, it is much stronger than additivity.
> For example, the duel probabilities could be a function of the best (or worst) player in each team.
>
> Dropping the consistency assumption would lead to an intractable model, as the order among the teams could be arbitrary and independent of the players' identities. Namely, there would be no linkage to the composition of the team.
>
> For example, one could create an instance in which players $n-k,...,n$ are the $k$ 'intuitively worst players' (i.e., they lose in almost all duels) but together they form the best team (and the unique Condorcet winning team), and players $1,...,k$ form the second best team in the linear order among teams.
> Clearly, as there is no connection between the team and it's players, in such a model any algorithm would need to iterate over all teams, which could be exponentially many (The problem becomes a dueling bandits problem with $\Omega((n/k)^{k})$ number of arms. Whenever $k=\Omega(n)$ this number is exponential in $n$.). Lower bounds from the dueling bandit literature do apply here and translate to $\Omega((n/k)^{k})$.
>
> In order to make the problem tractable for this first paper on our model, we introduced the consistency assumption, which links the performance of a team towards the performance of single players by inducing a combinatorial structure on the problem. This, in a way, is parallel to combinatorial bandits models (e.g. Rejwan \& Mansour 2019), namely we assumed that the performance of each team depends on the aggregated independent preformance of it's players.
>
> Having said this, a generalization of our results to other team valuation functions which can capture notions such as substitutes and/or complements is definitely an intriguing direction for future work.
>
> We are not sure what the reviewer meant by 'compare arms a and b directly based on skills'. The linear orders among teams is unknown to the learner (otherwise the learner would have simply selected the top team). In case by 'skills' the reviewer means evaluations that the learner can map into a score (reward), then similarly to the fundamental assumption in dueling bandits we assume that sampling based on individual arms' signals are not readily available.
>
> Connection to communications complexity Stable Marriage: This is an interesting connection. Gonczarowski et al. employed communication complexity to derive a lower bound on the query complexity of the stable marriage problem. This approach could also be a promising direction for deriving a tighter lower bound for our problem. Having said this, it is less obvious how to partition the information in our case. This is in contrast to the stable marriage problem where the underlying bipartite graph leads to a natural partition of the information.
>
> Experiments: This is a theoretical paper and our main goal is a theoretical analysis of the sampling and regret complexity.
>
> - Rejwan, I., and Mansour, Y. (2020). Top-$k$ combinatorial bandits with full-bandit feedback, ALT 2020.

---

### Official Review · Reviewer_Bd1b · 2021-07-21

**Rating:** 5
**Confidence:** 4

**Summary:**

This paper studies a variant of dueling bandits with teams. There are n players, and your goal is to select the “best” team of k of these players (more specifically, you want to select a team of k of these players that doesn’t lose to any team of size k that you can form from the remainder of the n-k players). Each round you can pick two teams of size k, watch them compete, and see who wins. The probabilities that one team beats another are chosen in such a way to be consistent with an underlying ordering of the players, in addition to satisfying some other assumptions (e.g. strong stochastic transitivity).

The authors study this problem in both the stochastic setting and the deterministic setting (where all the probabilities are guaranteed to be 0/1). In the stochastic setting, they can identify an optimal team whp with O((n + klog(k))/Delta^2), where Delta is some notion of “gap” between the performance of the kth player and the (k+1)th player. They accomplish this by first showing how to compare two players (roughly, you do something like pick a random S of size (k-1) and compare S + {a} to S + {b}, but something a little more complex than this). They then use this to reduce the problem to the well-studied problem of top K identification under pairwise comparisons.

In the deterministic case (where Delta is usually 0, so the previous bounds don’t apply), they give an algorithm which identifies a “best” team with O(kn log (k) + 2^{O(k)}) comparisons. Furthermore, when the order is an “additive total order” (i.e. generated by just adding values corresponding to specific members), they give an algorithm that only uses O(kn log(k) + k^5). These algorithms are complex combinatorial algorithms -- the main idea is showing that it is possible to identify useful comparisons somewhat quickly (especially under additive orders).


**Limitations And Societal Impact:**

No concerns here.

**Main Review:**


I think the problem of team selection in this sort of dueling bandits framework is an interesting practical problem (as the authors mention, it shows up in many different settings, especially in sports / games), and I think interesting progress on it would be welcome at NeurIPS. I also think that this paper is quite technically impressive (mainly the algorithms for the deterministic case). I do have some qualms about this paper, however.

- One issue I have is that the Delta gap in the bound of the algorithm in the stochastic case is very “tailored” to the specific algorithm they present. What I mean by this is that, usually when gaps appear in regret bounds, they correspond to some semantically meaningful notion about the test data (e.g. the gap in expected reward between the best and second-best solutions). This Delta is somewhat connected to the gap between the kth and (k+1)th players, but the definition feels not at all natural. Is there a good reason to care about this specific choice of Delta?

- The analysis of the deterministic case is definitely a combinatorial tour-de-force, but I think 1. it is unlikely to be applicable in practice (even the stochastic case has pretty strong assumptions that prevent it from being applicable in practice) and 2. I don’t really see the result or techniques in the analysis proving useful in other areas / for other problems (they are very tailored to this setting). These facts combined make me concerned how interesting the deterministic results will be to the NeurIPS audience.

The paper was well-written and easy to read.


**Time Spent Reviewing:**

3

---

> ### Author Response · Authors · 2021-08-10
> **Response to reviewer #Bd1b**
>
> Delta gap: This is a very good question and we certainly intend to incorporate the explanation we provide here within the paper.
>
> Intuitively, in our setting, the best team and the second best are not comparable by a team duel. E.g. with $k=2$ with player $1$ being at least as good as player $2$ which is at least as good as the rest of the players, the learner cannot select $(12,13)$ for a duel, though they are the first and second best solutions.
>
> The classical dueling bandits setting often uses the gap-parameter $\epsilon_{k,k+1}=P_{k,k+1}-1/2$. As in our setting two players cannot be compared directly, we needed to define a quantity that captures the distinguishability between $k$ and $k+1$ by only using team duels. We carefully constructed $\Delta_{k,k+1}$ such that it generalizes $\epsilon_{k,k+1}$ in the following sense:
>
> An alternative interpretation for our gap-parameter is $$\Delta_{k,k+1}=E[Y_{k,k+1}]+E[Z_{k,k+1}]-1,$$ Where $Y_{k,k+1}$ and $Z_{k,k+1}$ are the random variables defined in Definition B.1. in the Appendix. We have that $E[Z_{k,k+1}]\in[0,1]$ is the probability that $k$ wins against $k+1$, when both players are placed with disjoint random subsets of $k-1$ players. Hence, when $k=1$, $E[Z_{k,k+1}]=\epsilon_{k,k+1}$ as ${k}={1}$ and ${k+1}={2}$  form the only feasible teams. Moreover, we can show that $\Delta_{k,k+1}\geq \epsilon_{k,k+1}$ is satisfied in this case. For details of the proof see the response for reviewer bjRF.
>
> The second quantity, $E[Y_{k,k+1}]\in[0,1]$ captures the difference in performance of $k$ and $k+1$ within duels containing exactly one of the two players. We decided to extend $E[Z_{k,k+1}]$ by $E[Y_{k,k+1}]$ since our characterization in Section 3 has shown that these duels can help to distinguish $k$ and $k+1$, even if $E[Z_{k,k+1}]=1/2$ (which translates to $\epsilon_{k,k+1}=0)$.
>
> This extension is usually not done in dueling bandits, even though it would be reasonable and our entire characterization also holds for the case of $k=1$.
>
> Assumptions in the deterministic case:
> Within the deterministic section, we relaxed the assumption we had in the stochastic section regarding a gap between the $k-$th and the $(k+1)-$th best players. Within the last paragraph of Appendix C we explain how to extend all the results in the deterministic section up to Theorem 5.1 to the stochastic setting with the assumption that team duels winning probabilities are bounded away from 1/2.
>
> We hope that we have addressed the issues and that the reviewer will consider increasing the paper's score.

---

### Official Review · Reviewer_bjRF · 2021-07-21

**Rating:** 5
**Confidence:** 4

**Summary:**

This paper considers a variation of the dueling bandits problem where teams of arms compete against each other rather than individual arms. The paper considers two different models for team-wise comparisons: (1) stochastic where each competing team wins with certain probability and the better team wins with probability >= 1/2; (2) deterministic where the better team always wins. This paper works in the pure exploration setting as the goal is to find the Condorcet winner rather than minimising regret. For the deterministic setting the paper shows that any solution for the classic dueling bandits problem can be used to solve the team-wise dueling bandits problem with bounded number of duels depending on a \emph{gap} parameter. For the deterministic setting the paper gives an algorithm to find the Condorcet winner with a bounded number of duels with no dependence on gap.





**Limitations And Societal Impact:**

The paper addresses societal impact in the broader impact section.

**Main Review:**

1. Real world applications: The paper should expand a bit on the real-world applications where only team comparisons are possible. In my opinion the applications that suit this problem are based on team sports/gaming as one can only observe team comparisons here. In many other cases such as advertising one can even compare individual arms to each other which might be better from a sample complexity point of view.

2. Consistency Assumptions: This assumptions roughly states that there is an ordering among arms, and if a is better than b then adding a to  any team S results in a better team than adding b to S. In my opinion this assumption destroys the essence of team comparisons as one can expect that player a can be more compatible with players in S, and b can be more compatible with players in S'. Hence, adding a to S will result in a better team, whereas adding b to S' will result in a better team. If there is such a linear ordering amongst teams, then why can't we just compare arms a and b directly based on skills etc? It would be easier to make this assumption if there is a real-world validation based on experiments.

3. Dependence on \Delta and computation complexity: In the stochastic setting the algorithm is essentially searching for a witness among O(n^k) potential witnesses and if there are a few witnesses (which may likely be the case) then it can take O(n^k) time and samples. However, this detail is hidden away in the definition of \Delta (addressed briefly in Section 5). The paper mentions that the \Delta in this setting is similar to the \Delta in dulling bandits. I would argue that this is not the case as \Delta in dueling bandits is a natural parameter and can be even constant, whereas \Delta here is a result of the reduction to dueling bandits and will likely have a polynomial dependence on n.

4. Experiments: The paper would benefit greatly through experimental validation of the theoretical results on synthetic/real data and comparisons to a few baselines. Understanding the dependence of \Delta on n through experiments would also be helpful.

**Time Spent Reviewing:**

5

---

> ### Author Response · Authors · 2021-08-10
> **Response to reviewer #bjRF**
>
> Consistency assumption: Dropping the consistency assumption would lead to an intractable model, as the order among the teams could be arbitrary and independent of the players' identities. Namely, there would be no linkage to the composition of the team.
>
> For example, one could create an instance in which players $n-k,...,n$ are the $k$ 'intuitively worst players' (i.e., they lose in almost all duels) but together they form the best team (and the unique Condorcet winning team), and players $1,...,k$ form the second best team in the linear order among teams.
> Clearly, as there is no connection between the team and it's players, in such a model any algorithm would need to iterate over all teams, which could be exponentially many (The problem becomes a dueling bandits problem with $\Omega((n/k)^{k})$ number of arms. Whenever $k=\Omega(n)$ this number is exponential in $n$.). Lower bounds from the dueling bandit literature do apply here and translate to $\Omega((n/k)^{k})$.
>
> In order to make the problem tractable for this first paper on our model, we introduced the consistency assumption, which links the performance of a team towards the performance of single players by inducing a combinatorial structure on the problem. This, in a way, is parallel to combinatorial bandits models (e.g. Rejwan \& Mansour 2019), namely we assumed that the performance of each team depends on the aggregated independent preformance of it's players.
>
> Having said this, a generalization of our results to other team valuation functions which can capture notions such as substitutes and/or complements is definitely an intriguing direction for future work.
>
> We are not sure what the reviewer meant by 'compare arms a and b directly based on skills'. The linear orders among teams is unknown to the learner (otherwise the learner would have simply selected the top team). In case by 'skills' the reviewer means evaluations that the learner can map into a score (reward), then similarly to the fundamental assumption in dueling bandits we assume that sampling based on individual arms' signals are not readily available.
>
> Dependence on $\Delta$: We appreciate the interesting question and plan to incorporate this explanation into the paper! In what follows, we will show a connection between our gap ($\Delta$) to the usual gap parameter of dueling bandit setting.
> Consider $k=1$ (where each team contains a single player). In this case, an instance of our problem becomes a dueling bandit instance. In addition, consider the goal of identifying the best arm and denote $\Delta^{bandit}=P^{bandit}_{1,2}-1/2$.
>
> Notice that $\Delta^{bandit}$ is the common gap definition in dueling bandit.
> The characterization for deducible relations we have in the paper holds in particular for $k=1$. Since in this case the potential subsets witnesses set contains only $\{\emptyset\}$ and the potential subset-team witnesses set contains all the pairs $\{(\emptyset,i)\}$ where $i\notin{1,2}$. we have that
> $$\Delta=E[X_{1,2}(S,S',T)]=1/2(E[P_{1,2}]-E[P_{2,1}]+E_{i\notin {1,2}}[P_{1,i}]-E_{i\notin {1,2}}[P_{2,i}])=1/2(2P_{1,2}-1+E_{i\notin {1,2}}[P_{1,i}+P_{i,2}]-1)\geq P_{1,2}-1/2 =\Delta^{bandit}.$$ The inequality is due to $1>2>i$ for all $i$, thus $P_{1,i}+P_{i,2}\geq 1$.
>
> Technically, our gap could be strictly larger than $\Delta^{bandit}$ as it also take into account the average added winning probability of having player $1$ dueling instead of player $2$ when competing against a random player $i\notin {1,2}$.
>
> For example, if $P_{1,i\ne2}=5/6, P_{2,i\ne1}=2/3$ and $P_{1,2}=1/2+10^{-10}$ then the usual gap is $\Delta^{bandit}=P_{1,2}-1/2=10^{-10}$ while ours is $\Delta=\frac{1}{2}(10^{-10}+\frac{n}{6(n-2)})\approx 1/12$.).
>
>
> Finally, within the last paragraph of Appendix C we explain how to extend all the results in the deterministic section up to Theorem 5.1 to the stochastic setting with the assumption that team duels winning probabilities are bounded away from $1/2$. This provides a solution for instances where our gap is tiny (or even $0$).
>
> Experiments: This is a theoretical paper and our main goal is a theoretical analysis of the sampling and regret complexity.
>
> We hope that we have addressed the issues and that the reviewer will consider increasing the paper's score.
>
> - Rejwan, I., and Mansour, Y. (2020). Top-$k$ combinatorial bandits with full-bandit feedback, ALT 2020.

---

### Author Response · Authors · 2021-08-10
**General response**

We thank the reviewers for all the helpful feedback on our submission! We will incorporate it to improve the presentation of our results. We address the specific comments below.

---

### Decision · Program_Chairs · 2021-09-27

**Decision:**

Accept (Poster)

**Comment:**

The reviewers generally appreciate the analysis while questioning the assumptions and raising also some interesting questions about the Delta terms that I found well addressed by the authors in the response.

I also read the paper, and while I share some of the reviewers concerns, I also found the setup and analysis interesting and also just fun. I am not bothered that this is purely theoretical work. NeurIPS has in my time always accepted good theoretical papers, even when not accompanied by experiments.

I do agree with the reviewers that question the assumptions, which are strongly violated in all of the proposed applications. Players on sports teams have roles, and in League of Legends as well. R&D teams are almost always greater than the sum of their parts. Bundles in advertising will surely benefit from (an absence of) diversity.

The paper is quite dense, which is a problem with short page limits. I hope the authors can use the extra pages to introduce more intuition and clarity as suggested by the reviewers.